# Unique sparse decomposition of low rank matrices

**Dian Jin**
Rutgers University
dj370@scarletmail.rutgers.edu

**Xin Bing**
Cornell University
xb43@cornell.edu

**Yuqian Zhang**
Rutgers University
yqz.zhang@rutgers.edu

## Abstract

The problem of finding the unique low dimensional decomposition of a given matrix has been a fundamental and recurrent problem in many areas. In this paper, we study the problem of seeking a unique decomposition of a low rank matrix $Y \in \mathbb{R}^{p \times n}$ that admits a sparse representation. Specifically, we consider $Y = AX \in \mathbb{R}^{p \times n}$ where the matrix $A \in \mathbb{R}^{p \times r}$ has full column rank, with $r < \min\{n, p\}$, and the matrix $X \in \mathbb{R}^{r \times n}$ is element-wise sparse. We prove that this sparse decomposition of $Y$ can be uniquely identified by recovering ground-truth $A$ column by column, up to some intrinsic signed permutation. Our approach relies on solving a nonconvex optimization problem constrained over the unit sphere. Our geometric analysis for the nonconvex optimization landscape shows that any *strict* local solution is close to the ground truth solution, and can be recovered by a simple data-driven initialization followed with any second order descent algorithm. At last, we corroborate these theoretical results with numerical experiments.

## 1 Introduction

The problem of matrix decomposition has been a popular and fundamental topic under extensive investigations across several disciplines, including signal processing, machine learning, natural language processing [10, 11, 31, 46, 32, 8]. From the decomposition, one can construct efficient representation of the original data matrix. However, for any matrix $Y \in \mathbb{R}^{p \times n}$ that can be factorized as a product of two matrices $A \in \mathbb{R}^{p \times r}$ and $X \in \mathbb{R}^{r \times n}$, there exist infinitely many decompositions, simply because one can use any $r \times r$ invertible matrix $Q$ to construct $A' = AQ$ and $X' = Q^{-1}X$ such that $Y = AX = A'X'$, while $A' \neq A$ and $X' \neq X$. Thus, in various applications, additional structures and priors are being exploited to find a preferred representation [22, 15]. For example, principal component analysis (PCA) aims to find orthogonal representations which retain as much variations in $Y$ as possible [17, 23], whereas independent component analysis (ICA) targets the representations of statistically independent non-Gaussian signals [26].

In this paper, we are interested in finding a unique *sparse* low-dimensional representation of $Y$. To this end, we study the decomposition of a low rank matrix $Y \in \mathbb{R}^{p \times n}$ that satisfies

$$Y = AX, \tag{1.1}$$

where $A \in \mathbb{R}^{p \times r}$ is an unknown deterministic matrix, with $r < \min\{n, p\}$, and $X \in \mathbb{R}^{r \times n}$ is an unknown sparse matrix.

Formulation (1.1) is an important model problem in many applications. As columns of $Y$ are viewed as linear combinations of columns of $A$ with $X$ being the sparse coefficient, (1.1) can be used to form overlapping clusters of the $n$ columns of $Y$ via the support of $X$ with columns of $A$ being viewed as $r$ cluster centers [12, 7]. When we form a $p \times r$ low-dimensional representation of $Y$ via sparse combinations, this greatly enhance the interpretability of the resulting representations [28, 21, 4], in the same spirit as the sparse PCA, but (1.1) generalizes to the factorization of non-orthogonal matrices.

35th Conference on Neural Information Processing Systems (NeurIPS 2021).

To motivate our approach, we first consider the simple case that $\boldsymbol{A}$ has orthonormal columns, namely, $\boldsymbol{A}^T\boldsymbol{A} = \boldsymbol{I}_r$[1]. Then it is easy to see that the sparse coefficient matrix $\boldsymbol{X}$ is recovered by multiplying $\boldsymbol{Y}$ on the left by $\boldsymbol{A}^T$,

$$\boldsymbol{A}^T\boldsymbol{Y} = \boldsymbol{A}^T\boldsymbol{A}\boldsymbol{X} = \boldsymbol{X}. \tag{1.2}$$

The problem of finding such orthonormal matrix $\boldsymbol{A}$ boils down to successively finding a unit-norm direction $\boldsymbol{q}$ that renders $\boldsymbol{q}^T\boldsymbol{Y}$ as sparse as possible [34, 39, 35],

$$\min_{\boldsymbol{q}} \quad \left\|\boldsymbol{q}^T\boldsymbol{Y}\right\|_{\text{sparsity}} \qquad \text{s.t.} \quad \|\boldsymbol{q}\|_2 = 1. \tag{1.3}$$

However, the natural choice of sparsity penalty, either $\ell_0$ or $\ell_1$, leads to trivial and meaningless solutions, as there always exists $\boldsymbol{q}$ in the null space of $\boldsymbol{A}^T$ such that $\boldsymbol{q}^T\boldsymbol{Y} = \boldsymbol{0}$.

To avoid the null space of $\boldsymbol{A}^T$, we instead choose to find the unit direction $\boldsymbol{q}$ that maximizes the $\ell_4$ norm of $\boldsymbol{q}^T\boldsymbol{Y}$ as

$$\max_{\boldsymbol{q}} \quad \left\|\boldsymbol{q}^T\boldsymbol{Y}\right\|_4 \qquad \text{s.t.} \quad \|\boldsymbol{q}\|_2 = 1. \tag{1.4}$$

The above formulation is based on the key observation that the objective value is maximized when $\boldsymbol{q}$ coincides with one column of $\boldsymbol{A}$ (see, Section 2, for details) while the objective value is zero when $\boldsymbol{q}$ lies in the null space of $\boldsymbol{A}^T$. The $\ell_4$ norm objective function and its variants have been adopted as a sparsity regularizer in a line of recent works [30, 44, 43, 35, 42]. However, even with this new objective function, the null space of $\boldsymbol{A}^T$ persists as a challenge for solving the optimization problem: they form a flat region of saddle points.

This paper characterizes the nonconvex optimization landscape of (1.4) and proposes a guaranteed procedure that avoids the flat region and provably recovers the global solution to (1.4), which corresponds to one column of $\boldsymbol{A}$. More specifically, we demonstrate that, despite the non-convexity, (1.4) still possesses benign geometric property in the sense that any *strict* local solution with *large* objective value is globally optimal and recovers one column of $\boldsymbol{A}$, up to its sign. See, Theorem 3.1 in Section 3.1 for the population level result and Theorem 3.4 for the finite sample result.

We further extend these results to the general case when $\boldsymbol{A}$ only has full column rank in Theorem 3.6 of Section 3.2. To recover a general $\boldsymbol{A}$ with full column rank, our procedure first resorts to a preconditioning procedure of $\boldsymbol{Y}$ proposed in Section 2.3 and then solves a optimization problem similar to (1.4). From our analysis of the optimization landscape, the intriguing problem boils down to developing algorithms to recover the nontrivial local solutions by avoiding regions with small objective values. We thus propose a simple initialization scheme in Section 4.1 and prove in Theorem 4.3 that such initialization, proceeded with any second order descent algorithm [20, 27], suffices to find the global solution, up to some statistical error. Our theoretical analysis provides the explicit convergence rate of the statistical error and characterizes its dependence on various dimensions, such as $p$, $r$ and $n$, as well as the sparsity of $\boldsymbol{X}$.

Numerical simulation results are provided in Section 5. Due to the space limitation, we defer all the proof along with our conclusions and discussion of several future directions of our work to Appendix.

**Notations** Throughout this paper, we use bold lowercase letters, like $\boldsymbol{a}$, to represent vectors and bold uppercase letters, like $\boldsymbol{A}$, to represent matrices. For matrix $\boldsymbol{X}$, $\boldsymbol{X}_{ij}$ denotes the entry at the $i$-th row and $j$-th column of $\boldsymbol{X}$, with $\boldsymbol{X}_{i\cdot}$ and $\boldsymbol{X}_{\cdot j}$ denoting the $i$-th row and $j$-th column of $\boldsymbol{X}$, respectively. Oftentimes, we write $\boldsymbol{X}_{\cdot j} = \boldsymbol{X}_j$ for simplicity. We use grad and Hess to represent the Riemannian gradient and Hessian. For any vector $\boldsymbol{v} \in \mathbb{R}^d$, we use $\|\boldsymbol{v}\|_q$ to denote its $\ell_q$ norm, for $1 \le q \le \infty$. The notation $\boldsymbol{v}^{\circ q}$ stands for $\{\boldsymbol{v}_i^q\}_i$. For matrices, we use $\|\cdot\|_F$ and $\|\cdot\|_{\text{op}}$ to denote the Frobenius norm and the operator norm, respectively. For any positive integer $d$, we write $[d] = \{1, 2, \ldots, d\}$. The unit sphere in $d$-dimensional real space $\mathbb{R}^d$ is written as $\mathbb{S}^{d-1}$. For two sequences $a_n$ and $b_n$, we write $a_n \lesssim b_n$ if there exists some constant $C > 0$ such that $a_n \le Cb_n$ for all $n$. Both uppercase $C$ and lowercase $c$ are reserved to represent numerical constants, whose values may vary line by line.

---

[1] $\boldsymbol{I}_r$ is the identity matrix of size $r \times r$.

## 1.1 Related work

Finding the unique factorization of a matrix is an ill-posed problem in general due to infinitely many solutions. There exist several strands of studies from different contexts on finding the unique decomposition of $Y$ by imposing additional structures on $A$ and $X$. We start by reviewing the literature which targets the sparse decomposition of $Y$.

**Dictionary learning** The problems of dictionary learning (DL) [2, 38, 19, 39] and sparse blind deconvolution or convolutional dictionary learning [13, 29] study the unique decomposition of $Y = AX$ where $X$ is sparse and $A$ has full row rank. In this case, the row space of $Y$ lies in the row space of $X$, suggesting to recover the sparse rows of $X$ via solving the following problem,

$$\min_{q} \quad \left\| q^T Y \right\|_1 \qquad \text{s.t.} \quad q \neq 0. \tag{1.5}$$

Under certain scaling and incoherence conditions on $A$, the objective achieves the minimum value when $q$ is equal to one column of $A$, at the same time $q^T Y$ recovers one sparse row of $X$. This idea has been studied and modified in a strand of papers when $A$ has full row rank [38, 39, 45, 30, 44, 35, 42, 37, 47]. In our context, the major difference rises in the matrix $A$, which has full *column* rank rather than *row* rank, therefore minimizing $\left\| q^T Y \right\|_1$ as before only leads to some vector in the null space of $A^T$, yielding the trivial zero objective value.

We would love to note that [35] uses the same objective function in (1.4) to study the problem of overcomplete dictionary learning (where $A$ has full row rank), however the optimization landscape when $A$ has full column rank is significantly different from that in the overcomplete setting. The more complicated optimization landscape in our setting brings additional difficulty of the analysis and requires a proper initialization in our proposed algorithm. We refer to Appendix B for detailed technical comparison with [35].

**Sparse PCA** Sparse principal component analysis (SPCA) is a popular method that recovers a unique decomposition of a low-rank matrix $Y$ by utilizing the sparsity of its singular vectors. However, as being said, under $Y = AX$, SPCA is only applicable when $X$ coincides with the right singular vectors of $Y$. Indeed, one formulation of SPCA is to solve

$$\max_{U \in \mathbb{R}^{n \times r}} \quad \text{tr}\left(U^T Y^T Y U\right) - \lambda \|U\|_1, \qquad \text{s.t.} \quad U^T U = I_r, \tag{1.6}$$

which is promising only if $X$ corresponds to the right singular vectors of $Y$. It is worth mentioning that among the various approaches of SPCA, the following one might be used to recover one sparse row of $X$,

$$\min_{u,v} \quad \left\| Y - u v^T \right\|_2^2 + \lambda \|v\|_1 \qquad \text{s.t.} \quad \|u\|_2 = 1. \tag{1.7}$$

This procedure was originally proposed by [49] and [36] together with an efficient algorithm by alternating the minimization between $u$ and $v$. However, there is no guarantee that the resulting solution recovers the ground truth.

**Factor analysis** Factor analysis is a popular statistical tool for constructing low-rank representations of $Y$ by postulating $Y = AX + E$ where $A \in \mathbb{R}^{p \times r}$ is the so-called loading matrix with $r = \text{rank}(A) < \min\{n, p\}$, $X \in \mathbb{R}^{r \times n}$ contains $n$ realizations of a $r$-dimensional factor and $E$ is some additive noise. Here only $Y$ is observable. Factor analysis is mainly used to recover the low-dimension column space of $A$ or the row space of $X$, rather than to identify and recover the unique decomposition. Recently, [7] studied the unique decomposition of $Y$ when the columns of $X$ are i.i.d. realizations of a $r$-dimensional latent random factor. The unique decomposition is further used for (overlapping) clustering the rows of $Y$ via the assignment matrix $A$. To uniquely identify $A$, [7] assumes that $A$ contains at least one $r \times r$ identity matrix, coupled with other scaling conditions on $A$ (we refer to [7] for detailed discussions of other existing conditions in the literature of factor models that ensure the unique decomposition of $Y$ but require strong prior information on either $A$ or $X$). By contrast, we rely on the sparsity of $X$ instead of $A$ which is more general than requiring the existence of a $r \times r$ identity matrix.

**NMF and topic models** Such existence condition of identity matrix in either $A$ or $X$ has a variant in non-negative matrix factorization (NMF) [14] and topic models [3, 8, 9], also

see the references therein, where $\boldsymbol{Y}$, $\boldsymbol{A}$ and $\boldsymbol{X}$ have non-negative entries. Since all $\boldsymbol{Y}$, $\boldsymbol{A}$ and $\boldsymbol{X}$ from model (1.1) are allowed to have arbitrary signs in our context, the approaches designed for NMF and topic models are inapplicable.

## 2 Formulation and Assumptions

The decomposition of $\boldsymbol{Y} = \boldsymbol{AX}$ is not unique without further assumptions. To ensure the uniqueness of such decomposition, we rely on two assumptions on the matrices $\boldsymbol{A}$ and $\boldsymbol{X}$, stated in the following Section 2.1.

Our goal is to uniquely recover $\boldsymbol{A}$ from $\boldsymbol{Y}$, up to a some signed permutation. More precisely, we aim to recover columns of $\boldsymbol{AP}$ for some signed permutation matrix $\boldsymbol{P} \in \mathbb{R}^{r \times r}$. To facilitate the understanding and motivate our approach, in Section 2.2 we first state our procedure for the unique recovery of $\boldsymbol{A}$ when $\boldsymbol{A}$ has orthonormal columns. Its theoretical analysis is presented in Section 3. Later in Section 2.3, we discuss how to extend our results to the case when $\boldsymbol{A}$ is a more general full column rank matrix under Assumption 2.2.

For now, we only focus on the recovery of one column of $\boldsymbol{A}$ as the remaining columns can be recovered via the same procedure after projecting $\boldsymbol{Y}$ onto the complement space spanned by the recovered columns of $\boldsymbol{A}$ (see Section D for detailed discussion).

### 2.1 Assumptions

We first resort to the matrix $\boldsymbol{X} \in \mathbb{R}^{r \times n}$ being element-wise sparse. The sparsity of $\boldsymbol{X}$ is modeled via the Bernoulli-Gaussian distribution, stated in the following assumption.

**Assumption 2.1** *Assume $\boldsymbol{X}_{ij} = \boldsymbol{B}_{ij}\boldsymbol{Z}_{ij}$ for $i \in [r]$ and $j \in [n]$, where*

$$\boldsymbol{B}_{ij} \overset{i.i.d.}{\sim} Ber(\theta), \quad \boldsymbol{Z}_{ij} \overset{i.i.d.}{\sim} \mathcal{N}(0, \sigma^2). \tag{2.1}$$

The Bernoulli-Gaussian distribution is popular for modeling sparse random matrices [38, 2, 1, 39]. The overall sparsity level of $\boldsymbol{X}$ is controlled by $\theta$, the parameter of the Bernoulli distribution. We remark that the Gaussianity is assumed only to simplify the proof and to obtain more transparent deviation inequalities between quantities related with $\boldsymbol{X}$ and their population counterparts. Both our approach and analysis can be generalized to cases where $\boldsymbol{Z}_{ij}$ are centered i.i.d. sub-Gaussian random variables.

We also need another condition on the matrix $\boldsymbol{A}$. To see this, note that even when $\boldsymbol{A}$ were known, recovering $\boldsymbol{X}$ from $\boldsymbol{Y} = \boldsymbol{AX}$ requires $\boldsymbol{A}$ to have full column rank. We state this in the following assumption.

**Assumption 2.2** *Assume the matrix $\boldsymbol{A} \in \mathbb{R}^{p \times r}$ has $\mathrm{rank}(\boldsymbol{A}) = r$ with $\|\boldsymbol{A}\|_{op} = 1$.*

The unit operator norm of $\boldsymbol{A}$ is assumed without loss of generality as one can always re-scale $\sigma^2$, the variance of $\boldsymbol{X}$, by $\|\boldsymbol{A}\|_{\mathrm{op}}$.

### 2.2 Recovery of the orthonormal columns of $A$

In this section, we consider the recovery of one column of $\boldsymbol{A}$ when $\boldsymbol{A}$ is a semi-orthogonal matrix satisfying the following assumption.

**Assumption 2.3** *Assume $\boldsymbol{A}^T \boldsymbol{A} = \boldsymbol{I}_r$.*

Our approach recovers columns of $\boldsymbol{A}$ one at a time by adopting the $\ell_4$ maximization to penalize the sparsity of rows of matrix $\boldsymbol{X}$. Its rationale is based on the following lemma, assuming the orthogonality among columns of $\boldsymbol{A}$.

**Lemma 2.4** *Under Assumption 2.3, solving the following problem*

$$\max_{\boldsymbol{q}} \quad \left\|\boldsymbol{A}^T \boldsymbol{q}\right\|_4^4 \quad \text{s.t.} \quad \|\boldsymbol{q}\|_2 = 1 \tag{2.2}$$

*recovers one column of $\boldsymbol{A}$, up to its sign.*

Intuitively, under Assumption 2.3, we have $\|\boldsymbol{A}^T \boldsymbol{q}\|_2 \leq 1$ for any unit vector $\boldsymbol{q}$. Therefore, criterion (2.2) seeks a vector $\boldsymbol{A}^T \boldsymbol{q}$ within the unit ball to maximize its $\ell_4$ norm. When $\boldsymbol{q}$ corresponds to one column of $\boldsymbol{A}$, that is, $\boldsymbol{q} = \boldsymbol{a}_i$ for any $i \in [r]$, we have the largest objective $\|\boldsymbol{A}^T \boldsymbol{a}_i\|_4^4 = 1$. This $\ell_4$ norm maximization approach has been used in several related literature, for instance, sparse blind deconvolution [44, 30], complete and over-complete dictionary learning [43, 42, 35], independent component analysis [25, 24] and tensor decomposition [18].

The appealing property of maximizing the $\ell_4$ norm is its benign geometry landscape under the unit sphere constraint. Indeed, despite of the non-convexity of (2.2), our result in Theorem 3.1 implies that any strict location solution to (2.2) is globally optimal. This enables us to use any second order gradient ascent method to solve (2.2).

Motivated by Lemma 2.4, since we only have access to $\boldsymbol{Y} \in \mathbb{R}^{p \times n}$, we propose to solve the following problem to recover one column of $\boldsymbol{A}$,

$$\min_{\boldsymbol{q}} \quad F(\boldsymbol{q}) \doteq -\frac{1}{12\theta\sigma^4 n} \left\| \boldsymbol{Y}^T \boldsymbol{q} \right\|_4^4 \qquad \text{s.t.} \quad \|\boldsymbol{q}\|_2 = 1. \tag{2.3}$$

The scalar $(12\theta\sigma^4 n)^{-1}$ is a normalization constant. The following lemma justifies the usage of (2.3) and also highlights the role of the sparsity of $\boldsymbol{X}$.

**Lemma 2.5** *Under model (1.1) and Assumption 2.1, we have*

$$\mathbb{E}\left[F(\boldsymbol{q})\right] = -\frac{1}{4} \left[ (1-\theta) \left\| \boldsymbol{A}^T \boldsymbol{q} \right\|_4^4 + \theta \left\| \boldsymbol{A}^T \boldsymbol{q} \right\|_2^4 \right] \tag{2.4}$$

*where the expectation is taken over the randomness of $\boldsymbol{X}$.*

**Remark 2.6 (Role of the sparsity parameter $\theta$)** *Lemma 2.5 implies that, for large $n$, solving (2.3) approximately finds the solution to*

$$\min_{\boldsymbol{q}} \quad f\left(\boldsymbol{q}\right) \doteq -\frac{1}{4} \left[ (1-\theta) \left\| \boldsymbol{A}^T \boldsymbol{q} \right\|_4^4 + \theta \left\| \boldsymbol{A}^T \boldsymbol{q} \right\|_2^4 \right] \qquad \text{s.t.} \quad \|\boldsymbol{q}\|_2 = 1 \tag{2.5}$$

*The objective function is a convex combination of $\|\boldsymbol{A}^T \boldsymbol{q}\|_4^4$ and $\|\boldsymbol{A}^T \boldsymbol{q}\|_2^2$ with coefficients depending on the magnitude of $\theta$. In view of Lemma 2.4, it is easy to see that solving (2.5) recovers one column of $\boldsymbol{A}$, up to the sign, as long as $\theta < 1$. However, the magnitude of $\theta$ controls the benignness of the geometry landscape of (2.5). When $\theta$ is small, or $\boldsymbol{X}$ is sufficiently sparse, we essentially solve (2.2) which has the most benign landscape. On the other hand, when $\theta \to 1$, the landscape of (2.5) is mostly determined by the eigenvalue problem[2] which maximizes $\|\boldsymbol{A}^T \boldsymbol{q}\|_2$ subject to $\|\boldsymbol{q}\|_2 = 1$. We will demonstrate that when $\boldsymbol{X}$ is sufficiently sparse, second order descent algorithm with a simple initialization finds the globally optimal solution to (2.3) in Section 3.*

### 2.3 Recovery of the non-orthogonal columns of $A$

In this section, we discuss how to extend our procedure to recover $\boldsymbol{A}$ from $\boldsymbol{Y} = \boldsymbol{AX}$ when $\boldsymbol{A}$ is a general full column rank matrix satisfying Assumption 2.2. The main idea is to first resort to a preconditioning procedure of $\boldsymbol{Y}$ such that the preconditioned $\boldsymbol{Y}$ has the decomposition $\bar{\boldsymbol{A}}\bar{\boldsymbol{X}}$, up to some small perturbation, where $\bar{\boldsymbol{A}}$ satisfies Assumption 2.3 and $\bar{\boldsymbol{X}}$ satisfies Assumption 2.1 with $\sigma^2 = 1$. Then we apply our procedure in Section 2.2 to recover $\bar{\boldsymbol{A}}$. The recovered $\bar{\boldsymbol{A}}$ is further used to recover the original $\boldsymbol{A}$.

To precondition $\boldsymbol{Y}$, we propose to left multiply $\boldsymbol{Y}$ by the following matrix

$$\boldsymbol{D} \doteq \left[ \left( \boldsymbol{Y}\boldsymbol{Y}^T \right)^+ \right]^{1/2} \in \mathbb{R}^{p \times p} \tag{2.6}$$

where $\boldsymbol{M}^+$ denotes the Moore-Penrose inverse of any matrix $\boldsymbol{M}$. The resulting preconditioned $\boldsymbol{Y}$ satisfies

$$\bar{\boldsymbol{Y}} \doteq \boldsymbol{DY} = \bar{\boldsymbol{A}}\bar{\boldsymbol{X}} + \boldsymbol{E} \tag{2.7}$$

with $\bar{\boldsymbol{A}}$ satisfying Assumption 2.3, $\bar{\boldsymbol{X}} = \boldsymbol{X}/\sqrt{\theta n \sigma^2}$ and $\boldsymbol{E}$ being a perturbation matrix with small entries. We refer to Proposition 3.5 below for its precise statements.

Analogous to (2.3), we propose to recover one column of $\bar{\boldsymbol{A}}$ by solving the following problem

$$\min_{\|\boldsymbol{q}\|_2=1} \quad F_{\mathrm{g}}(\boldsymbol{q}) \doteq -\frac{\theta n}{12} \left\| \bar{\boldsymbol{Y}}^T \boldsymbol{q} \right\|_4^4 \tag{2.8}$$

Theoretical guarantees of this procedure are provided in Section 3.2. After recovering one column of $\bar{\boldsymbol{A}}$, the remaining columns of $\bar{\boldsymbol{A}}$ can be successively recovered via the procedure in Section D. In the end, $\boldsymbol{A}$ can be recovered by first inverting the preconditioning matrix $\boldsymbol{D}$ as $\boldsymbol{D}^{-1}\bar{\boldsymbol{A}}$ and then re-scaling its largest singular value to 1.

---

[2]When $\boldsymbol{A}$ is orthonormal, this eigenvalue problem processes the worst landscape as there are infinitely many solutions obtaining the same eigenvalue.

# 3 Theoretical Guarantees

We provide theoretical guarantees for our procedure (2.3) in Section 3.1 when $\boldsymbol{A}$ has orthonormal columns. The theoretical guarantees of (2.8) for recovering a general full column rank $\boldsymbol{A}$ are stated in Section 3.2.

## 3.1 Theoretical guarantees for semi-orthonormal $A$

In this section, we provide guarantees for our procedure by characterizing the solution to (2.3) when $\boldsymbol{A}$ satisfies Assumption 2.3.

As the objective function $F(\boldsymbol{q})$ in (2.3) concentrates around $f(\boldsymbol{q})$ in (2.5), it is informative to first analyze the solution to (2.5). Although (2.5) is a nonconvex problem and has multiple local solutions, Theorem 3.1 below guarantees that any strict local solution to (2.5) is globally optimal, in the sense that, it recovers one column of $\boldsymbol{A}$, up to its sign. We introduce the null region $R_0$ of our objective in (2.5),

$$R_0 = \left\{ \boldsymbol{q} \in \mathbb{S}^{p-1} : \|\boldsymbol{A}^T\boldsymbol{q}\|_\infty = 0 \right\}. \tag{3.1}$$

**Theorem 3.1 (Population case)** *Under Assumption 2.3, assume $\theta \leq 1/6$. Any local solution $\bar{\boldsymbol{q}}$ to (2.5), that is not in $R_0$, satisfies*

$$\bar{\boldsymbol{a}} = \boldsymbol{A}\boldsymbol{P}\boldsymbol{e}_1 \tag{3.2}$$

*for some signed permutation matrix $\boldsymbol{P} \in \mathbb{R}^{r \times r}$.*

The detailed proof of Theorem 3.1 is deferred to Appendix F.3. We only offer an outline of our analysis below.

The proof of Theorem 3.1 relies on analysis of the optimization landscape of (2.5) on disjoint partitions of $\mathbb{S}^{p-1} = \{\boldsymbol{q} \in \mathbb{R}^p : \|\boldsymbol{q}\|_2 = 1\}$[3], defined as

$$R_1 \doteq R_1(C_\star) = \left\{ \boldsymbol{q} \in \mathbb{S}^{p-1} : \|\boldsymbol{A}^T\boldsymbol{q}\|_\infty^2 \geq C_\star \right\}, \tag{3.3}$$
$$R_2 = \mathbb{S}^{p-1} \setminus (R_0 \cup R_1).$$

Here $C_\star$ is any fixed constant between $0$ and $1$. The upper bound follows from the inequality that $\|\boldsymbol{A}^T\boldsymbol{q}\|_\infty = \max_k |\boldsymbol{a}_k^T\boldsymbol{q}| \leq \|\boldsymbol{a}_k\|_2 \|\boldsymbol{q}\|_2 = 1$ for any $\boldsymbol{q} \in \mathbb{S}^{p-1}$. The region $R_0$ can be easily avoided by choosing the initialization such that the objective function $f(\boldsymbol{q})$ is not equal to zero. For $R_1$ and $R_2$, we are able to show the following results. Let $\operatorname{Hess} f(\boldsymbol{q})$ be the Riemannian Hessian matrix of (2.5) at any point $\boldsymbol{q} \in \mathbb{S}^{p-1}$.

(1) Optimization landscape for $R_1$:

> **Lemma 3.2** *Assume $\theta < 1$. Any local solution $\bar{\boldsymbol{q}} \in R_1(C_\star)$ to (2.5) with $C_\star > \frac{1}{2}\sqrt{\frac{\theta}{1-\theta}}$ recovers one column of $\boldsymbol{A}$, that is, for some signed permutation matrix $\boldsymbol{P}$*
>
> $$\bar{\boldsymbol{q}} = \boldsymbol{A}\boldsymbol{P}_{\cdot 1}.$$
>
> Lemma 3.2 shows that any critical point $\boldsymbol{q} \in R_1$ is either a strict saddle point that there exists a direction along which the Hessian is negative, or the desired local solution $\bar{\boldsymbol{q}}$ that satisfies the second order optimality condition and is equal to one column of $\boldsymbol{A}$, up to its sign.

(2) Optimization landscape for $R_2$:

> **Lemma 3.3** *Assume $\theta < 1/3$. For any point $\boldsymbol{q} \in R_2(C_\star)$ with $C_\star \leq \frac{1-3\theta}{2}$, there exists $\boldsymbol{v}$ such that*
>
> $$\boldsymbol{v}^T \operatorname{Hess} f(\boldsymbol{q}) \boldsymbol{v} < 0. \tag{3.4}$$
>
> Lemma 3.3 implies that any critical point in $R_2$ is a saddle point that can be escaped by negative curvature. Hence there is no local solution to (2.5) in the region $R_2$.

---

[3]Visualization of the partitions in $\mathbb{S}^2$ is available in section A.

Theorem 3.1 thus follows from Lemma 3.2 and Lemma 3.3, provided that

$$\sqrt{\frac{\theta}{1-\theta}} < 1 - 3\theta. \tag{3.5}$$

Condition (3.5) puts restrictions on the upper bound of $\theta$. It is easy to see that (3.5) holds for any $\theta \leq 1/6$. As discussed in Remark 2.6, a smaller $\theta$ leads to a more benign optimization landscape.

In light of Theorem 3.1, we now provide guarantees for the solution to the finite sample problem (2.3) in the following theorem. Define the sample analogue of the null region $R_0$ in (3.1) as

$$R_0'(c_\star) \doteq \left\{ q \in \mathbb{S}^{p-1} : \|A^T q\|_\infty^2 \leq c_\star \right\} \tag{3.6}$$

for any given value $c_\star \in [0, 1)$.

**Theorem 3.4 (Finite sample case)** *Under Assumptions 2.1 and 2.3, assume $\theta \in (0, 1/9]$ and*

$$n \geq C \max\left\{ \frac{r^2}{c_\star}, \log^2 n \right\} \frac{r \log n}{\theta c_\star} \tag{3.7}$$

*for some sufficiently large constant $C > 0$ and any $c_\star \in (0, 1/4]$. Then with probability at least $1 - cn^{-c'}$, any local solution $\bar{q}$ to (2.3) that is not in $R_0'(c_\star)$ satisfies*

$$\|\bar{q} - AP_{\cdot 1}\|_2^2 \lesssim \sqrt{\frac{r^2 \log n}{\theta n}} + \left( \theta r^2 + \frac{\log^2 n}{\theta} \right) \frac{r \log n}{n} \tag{3.8}$$

*for some signed permutation matrix $P$.*

Here we defer our discussion of technical details and full proof in section C.

## 3.2 Theoretical guarantees for general full column rank $A$

In this section, we provide theoretical guarantees for our procedure of recovering a general full column rank matrix $A$ under Assumption 2.2.

Recall from Section 2.3 that our approach first preconditions $Y$ by using $D$ from (2.6). The following proposition provides guarantees for the preconditioned $Y$, denoted as $\bar{Y} = DY$. The proof is deferred to Appendix F.5. Write the SVD of $A = U_A D_A V_A^T$ with $U_A \in \mathbb{R}^{p \times r}$ and $V_A \in \mathbb{R}^{r \times r}$ being, respectively, the left and right singular vectors.

**Proposition 3.5** *Under Assumptions 2.1 and 2.2, assume $n \geq Cr/\theta^2$ for some sufficiently large constant $C > 0$. With probability greater than $1 - 2e^{-c'r}$, one has*

$$\bar{Y} = \bar{A}\bar{X} + E \tag{3.9}$$

*where $\bar{A} = U_A V_A^T$, $\bar{X} = X/\sqrt{\theta n \sigma^2}$ and $E = \bar{A}\Delta\bar{X}$ with*

$$\|\Delta\|_{op} \leq c'' \frac{1}{\theta}\sqrt{\frac{r}{n}}. \tag{3.10}$$

*Here $c'$ and $c''$ are positive constants.*

Proposition 3.5 implies that, when $n \geq Cr/\theta^2$, the preconditioned $Y$ satisfies

$$\bar{Y} = \bar{A}(I_r + \Delta)\bar{X} \approx \bar{A}\bar{X} \tag{3.11}$$

with $\bar{A}^T \bar{A} = I_r$. This naturally leads us to apply our procedure in Section 2.2 to recover columns of $\bar{A}$ via (2.8). We formally show in Theorem 3.6 below that any local solution to (2.8) approximately recover one column of $\bar{A}$ up to a signed permutation matrix. Similar to (3.6), define

$$R_0''(c_\star) \doteq \left\{ q \in \mathbb{S}^{p-1} : \|\bar{A}^T q\|_\infty^2 \leq c_\star \right\} \tag{3.12}$$

for some given value $c_\star \in [0, 1)$.

**Theorem 3.6** *Under Assumption 2.1 and 2.2, assume $\theta \in (0, 1/9]$ and*

$$n \geq C \frac{r}{c_\star \theta} \max\left\{ \log^3 n, \frac{\log n}{c_\star \theta \sqrt{\theta}}, \frac{\log^2 n}{c_\star \theta}, \frac{r}{c_\star \sqrt{\theta}}, \frac{r^2 \log n}{c_\star} \right\}. \tag{3.13}$$

*Then with probability at least $1 - cn^{-c'} - 4e^{-c''r}$, any solution $\bar{q}$ to (2.8) that is not in Region $R_0''(c_\star)$ satisfies*

$$\|\bar{q} - \bar{A}P_{\cdot 1}\|_2^2 \lesssim \sqrt{\frac{r \log n}{\theta^2 n}} + \sqrt{\frac{r^2 \log n}{\theta n}} + \left( \theta r^2 + \frac{\log^2 n}{\theta} \right) \frac{r \log n}{n} \tag{3.14}$$

*for some signed permutation matrix $P$.*

The proof of Theorem 3.6 can be found in Appendix F.6. Due to the preconditioning step, the requirement of the sample size in (3.13) is slightly stronger than (3.7), whereas the estimation error of $\bar{q}$ only has an additional $\sqrt{r \log n / (\theta^2 n)}$ term comparing to (3.8).

Theorem 3.6 requires to avoid the null region $R_0''(c_\star)$ in (3.12). We provide a simple initialization in the next section that provably avoids $R_0''$. Furthermore, every iterate of any descent algorithm based on such initialization is provably not in $R_0''$ either.

# 4   Complete Algorithm and Provable Recovery

In this section, we present a complete pipeline for recovering $A$ from $Y$. So far we have established that every local solution to (2.8), that is not in $R_0''(c_\star)$, approximately recovers one column of $\bar{A} = U_A V_A^T$. To our end, we will discuss: (1) a data-driven initialization in Section 4.1 which, together with Theorem 3.6, provably recovers one column of $\bar{A}$; (2) a deflation procedure [38, 39, 35] in Section D that sequentially recovers all remaining columns of $\bar{A}$. Due to the limitation of space we defer our discussion of deflation procedure in appendix.

## 4.1   Initialization

Our goal is to provide a simple initialization such that solving (2.8) via any second order descent algorithm provably recovers one column of $\bar{A}$. According to Theorem 3.6, such an initialization needs to guarantee the following conditions.

- **Condition I:** The initial point $q^{(0)}$ does not fall into region $R_0''(c_\star)$ for some $c_\star$ satisfying (3.13) in Theorem 3.6.

- **Condition II:** The updated iterates $q^{(k)}$, for all $k \geq 1$, stay away from $R_0''(c_\star)$ as well.

We propose the following initialization
$$q^{(0)} = \frac{\bar{Y} \mathbf{1}_n}{\|\bar{Y} \mathbf{1}_n\|_2} \in \mathbb{S}^{p-1}. \tag{4.1}$$

The following two lemmas guarantee that both **Condition I** and **Condition II** are met for this choice. Their proofs can be found in Appendices F.7 and F.8.

**Lemma 4.1** *Under Assumption 2.1 and 2.2, assume $\theta \in (0, 1/9]$ and*
$$n \geq C \frac{r^2}{\theta} \max \left\{ \log^3 n, \ \frac{r \log n}{\theta \sqrt{\theta}}, \ \frac{r \log^2 n}{\theta}, \ \frac{r^2}{\sqrt{\theta}}, \ r^3 \log n \right\}. \tag{4.2}$$
*holds, then, with probability at least $1 - 2e^{-cr}$, the initialization $q^{(0)}$ in (4.1) is not in region $R_0''(c_\star)$ with $c_\star = 1/(2r)$.*

**Lemma 4.2** *Let $q^{(k)}$, for $k \geq 1$, be any updated iterate from solving (2.3) by using any monotonic decreasing algorithm with the initial point $q^{(0)}$ chosen as (4.1). If*
$$n \geq C \frac{r^2}{\theta} \max \left\{ \log^3 n, \ \frac{r \log n}{\theta \sqrt{\theta}}, \ \frac{r^2}{\sqrt{\theta}}, \ \theta^2 r^2 \log n \right\} \tag{4.3}$$
*holds, then, with probability at least $1 - cn^{-c'} - 2e^{-c''r}$, one has*
$$q^{(k)} \notin R_0''(c_\star), \qquad \text{for all } k \geq 1,$$
*with $c_\star = 1/(2r)$.*

Combining Lemmas 4.1 and 4.2 together with Theorem 3.6 readily yields the following theorem.

**Theorem 4.3** *Under Assumptions 2.1 and 2.2, assume $\theta \in (0, 1/9]$ and (4.2) holds. Let $\bar{q}$ be any local solution to (2.8) from any monotonic decreasing second order algorithms with the initial point chosen as (4.1). With probability at least $1 - cn^{-c'} - 4e^{-c''r}$, one has*
$$\|\bar{q} - \bar{A} P_{\cdot 1}\|_2^2 \lesssim \sqrt{\frac{r \log n}{\theta^2 n}} + \sqrt{\frac{r^2 \log n}{\theta n}} + \frac{r \log^3 n}{\theta n}$$
*for some signed permutation matrix $P$.*

Theorem 4.3 provides the guarantees for using any monotonic decreasing second order algorithms [33, 5] to solve (2.8) with the initialization chosen in (4.1).

# 5 Experiments

In this section we verify the empirical performance of our proposed procedure for recovering $A$ under model (1.1) in different scenarios. Due to the space limit, we defer more experiments to the Appendix of this paper.

**Experiment setup**

To generate the data $Y = AX$, we generate the columns of $A$ by using the normalized left singular vectors of $R \in \mathbb{R}^{p \times r}$ where $R_{ij} \overset{i.i.d.}{\sim} \mathcal{N}(0,1)$. The sparse coefficient matrix $X \in \mathbb{R}^{r \times n}$ are generated as $X_{ij} \overset{i.i.d.}{\sim} \text{BG}(\theta)$. To evaluate the success of recovering one column vector of $A$ for any estimate $q \in \mathbb{S}^{p-1}$, we use the following criterion,

$$\text{Err}(q) = \min_{1 \leq i \leq r} (1 - |\langle q, a_i \rangle|) \tag{5.1}$$

If $\text{Err}(q) \leq \rho_e$, we say the vector $q$ recovers the ground-truth column vector of $A$. We choose $\rho_e = 1 \times 10^{-2}$ in our simulation settings. To evaluate the recovery of the whole matrix $A$, we use the following normalized Frobenius norm between any estimate $A_{est}$ and the true $A$:

$$\min_{P} \quad \frac{1}{\sqrt{r}} \|A_{est} - AP\|_F \quad \text{s.t.} \quad P \text{ is a signed permutation matrix.} \tag{5.2}$$

We first evaluate the probability of successfully recovering one column of $A$ in two scenarios. In the first case, we vary simultaneously $\theta$ and $r$ while in the second case we change $n$ and $r$. We then evaluate the performance of our procedure, Algorithm 1 in Section D, for recovering the full matrix $A$.

**Recovery probability with varying $\theta$ and $r$**

We fix $p = 100$ and $n = 5 \times 10^3$ while vary $\theta \in \{0.01, 0.04, \ldots, 0.58\}$ and $r \in \{10, 30, \ldots, 70\}$. For each pair of $(\theta, r)$, we repeatedly generate 200 data sets and apply our procedure in (2.8). The averaged recovery probability of our procedure over the 200 replicates is shown in Figure 1a. The recovery probability gets larger as $r$ decreases, in line with Theorem 3.6. We also note that the recovery increases for smaller $\theta$. This is because smaller $\theta$ renders a nicer geometric landscape of the proposed non-convex problem, as detailed in Remark 2.6. On the other hand, the recovery probability decreases when $\theta$ is approaching to $0$. As suggested by Theorem 3.4, the statistical error of estimating $A$ gets inflated as $\theta$ gets too small.

**Recovery probability with varying $n$ and $r$**

Here we fix $p = 100$ and the sparsity parameter $\theta = 0.1$. We vary $r \in \{10, 30, \ldots, 70\}$ and $n \in \{2000, 3000, \ldots, 12000\}$. Figure 1b shows the averaged recovery probability of our procedure over 200 replicates in each setting. Our procedure performs increasingly better as $n$ increases, as expected from Theorem 3.4.

# 6 Conclusion and Future Work

In this paper, we have studied the unique decomposition of a low rank matrix $Y$ that admits a sparse low-dimensional representation. Under model $Y = AX$ where $X$ has i.i.d. Bernoulli-Gaussian entries and $A$ has full column rank, we propose a nonconvex procedure that provably recovers $A$, a quantity that can be further used to recover $X$. We provide a complete analysis for recovering one column of $A$, up to the sign, by showing that any second order descent algorithm provably attains the global solution with a simple and data-driven initialization, despite the nonconvex nature of the proposed procedure.

There are several directions that are certainly worth further pursuing. For instance, a complete analysis of the deflation procedure for recovering the full matrix $A$ is certainly of great interest. It is also worth studying this decomposition problem in presence of some additive errors, that is, $Y = AX + E$. Our current procedure only tolerates $E$ that has small entries. How to modify our procedure to accommodate a moderate / large $E$ is an interesting and challenging problem that we leave to future research.

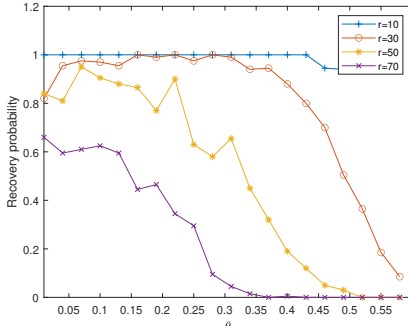

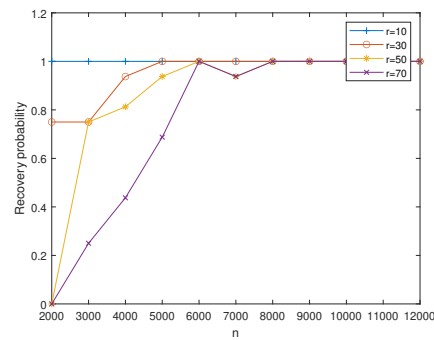

**(a) Recovery probability versus $\theta$:** the averaged probability of successful recovery for different $\theta$ and $r$ with $p = 100$ and $n = 1.2 \times 10^4$.

**(b) Recovery probability versus $n$:** the averaged probability of successful recovery for different $n$ and $r$ with $p = 100$ and $\theta = 0.1$.

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
