

**Figure 2:** Landscape of object function (2.8) over $\mathbb{S}^2$.

## A Landscape Visualization

Figure 2 visualize the landscape of our object function (2.8) on $\mathbb{S}^2$ with region defined as in (2.3). Here the ground truth $\boldsymbol{A}$ is set to $[\boldsymbol{e}_1, \boldsymbol{e}_2]$ while sparsity $\theta$ and constant $C_\star$ are set to $0.5$, $0.65$ respectively. As we can see in the figure, Region $R_1$ contains saddle points and local solutions that recover ground truth up to sign ambiguity which is line with lemma 3.2.

## B Techinical comparison with [35]

[35] studies the unique recovery of $\boldsymbol{Y} = \boldsymbol{A}\boldsymbol{X}$ under the setting of over-complete dictionary learning, that is, $\boldsymbol{A}$ has full **row** rank with $p \leq r$. Although our objective in (2.3) is similar to that used in [35], the low-rank structure of $\boldsymbol{Y}$ in our setting leads to fundamental differences in both the rationale of using (2.3) and the subsequent analysis. To be specific, under the setting in [35], the matrix $\boldsymbol{A} = (\boldsymbol{a}_1, \ldots, \boldsymbol{a}_r) \in \mathbb{R}^{p \times r}$, with $r \geq p$, is assumed to be unit norm tight frame (UNTF), in the sense that

$$\boldsymbol{A}\boldsymbol{A}^T = \frac{r}{p}\boldsymbol{I}_p, \qquad \|\boldsymbol{a}_i\|_2 = 1, \qquad \mu = \max_{i \neq j}|\langle \boldsymbol{a}_i, \boldsymbol{a}_j \rangle| \ll 1.$$

Under this condition and Assumption 2.1, the objective $F(\boldsymbol{q})$ in (2.3) satisfies (see, display (2.4) in [35])

$$\mathbb{E}[F(\boldsymbol{q})] = -\frac{1}{4}(1-\theta)\|\boldsymbol{A}^T\boldsymbol{q}\|_4^4 - C \tag{B.1}$$

where $C$ is some numerical value that does not depend on $\boldsymbol{q}$. Therefore, solving (2.3), for large $n$, approximately maximizes $\|\boldsymbol{A}^T\boldsymbol{q}\|_4^4$ over the unit sphere in the context of [35].

There are at least three major differences to be noted. First, as $\boldsymbol{A}$ is UNTF in the setting of [35], columns of $\boldsymbol{A}$ are not orthonormal, or equivalently, $\mu > 0$. As a result, Lemma 2.4 does not hold for their setting. In another word, even one can directly solve (2.2), the solution does not exactly recover one column of $\boldsymbol{A}$. Indeed, Proposition B.1 in [35] shows that the difference between the solution to (2.2) and one column of $\boldsymbol{A}$ is small when $\mu \ll 1$ but does to not exactly equal to zero unless $\mu = 0$. By contrast, when $\boldsymbol{A}$ satisfies $\boldsymbol{A}^T\boldsymbol{A} = \boldsymbol{I}_r$ in our setting, the exact recovery of columns of $\boldsymbol{A}$ is achievable via solving (2.2) as shown in Lemma 2.4.

Second, due to $\mathrm{rank}(\boldsymbol{A}) = r$, solving (2.3) in our setting approximates maximizes (2.5), the objective of which is a convex combination of $\|\boldsymbol{A}^T\boldsymbol{q}\|_4^4$ and $\|\boldsymbol{A}^T\boldsymbol{q}\|_2^4$ with coefficients depending on the sparsity parameter $\theta$. Thus, the expected objective in our setting no longer coincides with that in [35] and in fact is more complicated due to the extra term $\|\boldsymbol{A}^T\boldsymbol{q}\|_2^4$. Basically, this paper provide an answer to the composition of two nonconvex optimization problems: a sparisity problem and an eigenvalue problem. This additional term brings

more complications in our analysis of the geometry landscape of (2.3) and requires more delicate arguments.

Third, the low-rank structure of $\boldsymbol{Y}$ leads to a null region of solving (2.3), that is, the region of $\boldsymbol{q}$ such that $\boldsymbol{q}^T\boldsymbol{Y} = \boldsymbol{q}^T\boldsymbol{A}\boldsymbol{X} = \boldsymbol{0}$ (see, (3.1) for the population-level analysis and (3.6) – (3.12) for the finite sample results). This null region does not appear when the matrix $\boldsymbol{A}$ is UNTF. In presence of such a region, to provide recovery guarantees for the desired solution, we need a proper initialization outside of this region and to further demonstrate that every iterate does not fall back into the null region. Such analysis brings more technical challenges, for instance, Lemmas 4.1 and 4.2, to our analysis than [35].

## C   Technical Details of Theorem 3.4

The proof of Theorem 3.4 can be found in Appendix F.4. The geometric analysis of the landscape of the optimization problem (2.3) is in spirit similar to that of Theorem 3.1, but has additional technical difficulty of taking into account the deviations between the finite sample objective $F(\boldsymbol{q})$ in (2.3) and the population objective $f(\boldsymbol{q})$ in (2.5), as well as the deviations of both their gradients and hessian matrices. Such deviations also affect both the size of $R_0'(c_\star)$ in (3.6), a enlarged region of $R_0$ in (3.1), via condition (3.7), and the estimation error of the local solution $\bar{\boldsymbol{q}}$.

In Lemmas F.13, F.14 and F.15 of Appendix F.9, we provide finite sample deviation inequalities of various quantities between $F(\boldsymbol{q})$ and $f(\boldsymbol{q})$. Our analysis characterizes the explicit dependency on dimensions $n$, $p$ and $r$, as well as on the sparsity parameter $\theta$. In particular, our analysis is valid for fixed $p$, $r$ and $\theta$, as well as growing $p = p(n)$, $r = r(n)$ and $\theta = \theta(n)$.

The estimation error of our estimator in (3.8) depends on both the rank $r$ and the sparsity parameter $\theta$, but is independent of the higher dimension $p$. The smaller $\theta$ is, the larger estimation error (or the stronger requirement on the sample size $n$) we have. This is as expected since one needs to observe enough information to accurately estimate the population-level objective in (2.5) by using (2.3). On the other hand, recalling from Remark 2.6 that a larger $\theta$ could lead to worse geometry landscape. Therefore, we observe an interesting trade-off of the magnitude of $\theta$ between the optimization landscape and the statistical error.

## D   Recovering the Full Matrix $A$

Theorem 4.3 provides the guarantees for recovering one column of $\bar{\boldsymbol{A}}$. In this section, we discuss how to recover the remaining columns of $\bar{\boldsymbol{A}}$ by using the deflation method.

Suppose solving (2.8) recovers $\bar{\boldsymbol{a}}_1$, the first column of $\bar{\boldsymbol{A}}$. For $2 \leq k \leq r$, write $\mathcal{A}_k = \mathrm{span}(\bar{\boldsymbol{a}}_1, \ldots, \bar{\boldsymbol{a}}_{k-1})$, the space spanned by all previously recovered columns of $\boldsymbol{A}$ at step $k$. Further define $P_{\mathcal{A}_k}$ as the projection matrix onto $\mathcal{A}_k$ and write $P_{\mathcal{A}_k}^\perp = \boldsymbol{I}_p - P_{\mathcal{A}_k}$. We propose to solve the following problem to recover a new column of $\bar{\boldsymbol{A}}$,

$$\min_{\boldsymbol{q}} \quad -\frac{\theta n}{12}\left\|\boldsymbol{q}^T P_{\mathcal{A}_k}^\perp \bar{\boldsymbol{Y}}\right\|_4^4, \tag{D.1}$$
$$\text{s.t.} \quad \|\boldsymbol{q}\|_2 = 1.$$

To facilitate the understanding, consider $k = 2$ and $P_{\mathcal{A}_k}^\perp = P_{\bar{\boldsymbol{a}}_1}^\perp$. Then (D.1) becomes

$$\min_{\boldsymbol{q}} \quad -\frac{\theta n}{12}\left\|\boldsymbol{q}^T P_{\bar{\boldsymbol{a}}_1}^\perp \bar{\boldsymbol{Y}}\right\|_4^4, \tag{D.2}$$
$$\text{s.t.} \quad \|\boldsymbol{q}\|_2 = 1.$$

From Proposition 3.5, we observe that

$$P_{\bar{\boldsymbol{a}}_1}^\perp \bar{\boldsymbol{Y}} \approx P_{\bar{\boldsymbol{a}}_1}^\perp \bar{\boldsymbol{A}}\bar{\boldsymbol{X}} = \bar{\boldsymbol{A}}_{(-1)}\bar{\boldsymbol{X}}_{(-1)},$$

where we write $\bar{\boldsymbol{A}}_{(-1)} \in \mathbb{R}^{p \times (r-1)}$ and $\bar{\boldsymbol{X}}_{(-1)} \in \mathbb{R}^{(r-1) \times n}$ for $\bar{\boldsymbol{A}}$ and $\bar{\boldsymbol{X}}$ with the 1th column and the 1th row removed, respectively. Then it is easy to see that recovering one column

of $\bar{A}_{(-1)}$ from $P_{\bar{a}_1}^{\perp} \bar{Y}$ is the same problem as recovering one column of $\bar{A}$ from $\bar{Y}$ with $r$ replaced by $r - 1$, hence can be done via solving (D.2). Similar reasoning holds for any $2 \leq k \leq r$.

For the reader's convenience, we summarize our whole procedure of recovering $\bar{A}$ in Algorithm 1. The original $A$ is recovered by $D^{-1}\bar{A}$ in the end.

---

**Algorithm 1:** Sparse Low Rank Decomposition

---

**Data:** a low rank matrix $Y \in \mathbb{R}^{p \times n}$ with rank $r$
**Result:** matrix $A$
Set $\mathcal{A}_j = \emptyset$ and initialize $q^{(0)}$ as (4.1);
**for** $j = \{1, 2, \ldots, r\}$ **do**
  Solve $a_j^\star$ from (D.1) by using $q^{(0)}$ and any second order descent algorithm;
  Update $A_{\cdot j} = a_j^\star$;
  Set $\mathcal{A}_j = \text{span}(A_{\cdot 1}, \ldots, A_{\cdot j})$;
**end**

---

# E   Additional Experiment

In this section, we provide additional experiments to corroborate the main paper. In Section E.1 we demonstrate that the algorithm introduced in section D could be extended to recover general full dictionary $A$ under assumption 2.2; in Section E.2, we compare our methods with algorithms for solving sparse principal component analysis (SPCA) under assumptions 2.1 and 2.3. Here the recovery error is defined as normalized Frobenius norm of difference between ground-truth $A_{gt}$ and our estimated $A$. Please refer to (5.2) for detailed definition of recovery error in equation 5.2, section 5.

## E.1   Extension to general full column rank $A$

This section we demonstrate our algorithm proposed in section D can be applied to recover a general full column rank matrix $A$(Theorem 3.6). Here $A$ is generated as $A_{ij} \overset{i.i.d.}{\sim} \mathcal{N}(0, 1)$ and $X$ follows the same setting as in section 5. Figure 3 shows the performance of our method under different $\theta$ and $r$. The error for estimating $A$ gets smaller when $\theta$ and $r$ decrease. When $r$ is small enough, our method successfully recover whole general full column rank matrix $A$.

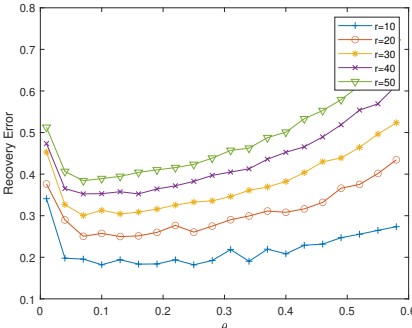

**Figure 3: Recovery of full general Dictionary** $A$: for different $\theta$ and $r$ with $n = 1.2 \times 10^4$ and $p = 100$ fixed.

---

[3] $P_{\mathcal{A}_j}^{\perp}$ is the projection matrix onto the orthogonal space of $\mathcal{A}_j$.

### E.2 Comparison with the SPCA

In this section we compare the performance of our method with two SPCA procedures. Among the various algorithms for solving the SPCA problem, we compare with the LARS algorithm [49, 48] and the alternating direction method (ADM) [6, 40]. The LARS solves

$$\min_{\boldsymbol{v}_i} \quad \|\boldsymbol{Z}_i - \boldsymbol{Y}\boldsymbol{v}_i\|_2 + \lambda\|\boldsymbol{v}_i\|_1 + \lambda'\|\boldsymbol{v}_i\|_2,$$

for each $1 \leq i \leq r$, to recover rows of $\boldsymbol{X}$. Here, $\boldsymbol{Z}_i$ is the $i$th principle component of $\boldsymbol{Y}$. Denote $\hat{\boldsymbol{X}}$ by the estimated $\boldsymbol{X}$ from LARS. The matrix $\boldsymbol{A}$ is then recovered by $\boldsymbol{Y}\hat{\boldsymbol{X}}^T(\hat{\boldsymbol{X}}\hat{\boldsymbol{X}}^T)^{-1}$ with its operator norm re-scaled to 1. On the other hand, the ADM algorithm solves

$$\min_{\boldsymbol{u},\boldsymbol{v}} \quad \left\|\boldsymbol{Y} - \boldsymbol{u}\boldsymbol{v}^T\right\|_F^2 + \lambda\|\boldsymbol{v}\|_1 \quad \text{s.t.} \quad \|\boldsymbol{u}\|_2 = 1,$$

to recover one column of $\boldsymbol{A}$ (from the optimal $\boldsymbol{u}$) by alternating minimization between $\boldsymbol{u}$ and $\boldsymbol{v}$. The above procedure is successively used to recover the rest columns of $\boldsymbol{A}$ by projecting $\boldsymbol{Y}$ onto the complement space spanned by the recovered columns of $\boldsymbol{A}$ [36].

Since both SPCA procedures aim to recover $\boldsymbol{A}$ with orthonormal columns, to make a fair comparison, we generate $\boldsymbol{A}$ by the normalized left singular vectors of $\boldsymbol{R} \in \mathbb{R}^{p \times r}$ where $\boldsymbol{R}_{ij} \overset{i.i.d.}{\sim} \mathcal{N}(0,1)$. Figure 4a and 4b depict the estimation error of three methods in terms of (5.2) for varying $r$ and $\theta$, respectively. The estimation errors get larger when either $r$ and $\theta$ increases for all three methods. LARS has the worse performance in all scenarios. Compared to ADM method, our method has the similar performance for relatively small $\theta$ ($\theta < 0.4$) but has significantly better performance for moderate $\theta$ ($\theta > 0.4$). It is also worth mentioning that, in contrast to the established guarantees of our method, there is no theoretical guarantee that the ADM method recovers the ground truth.

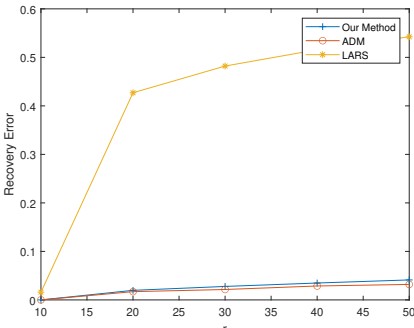
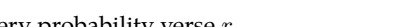
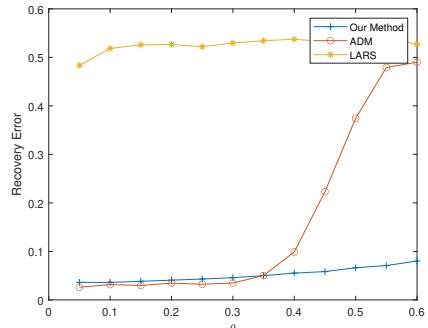

(a) Recovery probability verse $r$    (b) Recovery probability verse $\theta$

**Figure 4: Experiment result for two different methods comparing and full general dictionary recovery:** Figure 4a and 4b shows the relation of recovery error between different $\theta$ and atoms number $r$ for two methods with $p = 100$ and $n = 1.2 \times 10^4$. In Figure 4a, $\theta$ is fixed to 0.1 and in Figure 4b $r = 30$ for all data points.

## F  Main Proofs

### F.1  Proof of Lemma 2.4

**Proof.** First, note that, for any $\boldsymbol{q} \in \mathbb{S}^{p-1}$,

$$\|\boldsymbol{A}^T\boldsymbol{q}\|_4^4 = \sum_{j=1}^r (\boldsymbol{a}_j^T\boldsymbol{q})^4 \leq \max_{1 \leq j \leq r}(\boldsymbol{a}_j^T\boldsymbol{q})^2 \sum_{j=1}^r (\boldsymbol{a}_j^T\boldsymbol{q})^2 \leq \max_{1 \leq j \leq r}\|\boldsymbol{a}_j\|_2^2\,\lambda_1(\boldsymbol{A}\boldsymbol{A}^T) = 1. \quad \text{(F.1)}$$

Here $\lambda_1(\boldsymbol{A}\boldsymbol{A}^T)$ denotes the largest eigenvalue of $\boldsymbol{A}\boldsymbol{A}^T$ and is equal to $\lambda_1(\boldsymbol{A}^T\boldsymbol{A}) = 1$. Also note that the maximal value one is achieved by $\boldsymbol{q} = \pm\boldsymbol{a}_i$ for all $1 \leq i \leq r$ as

$$\|\boldsymbol{A}^T\boldsymbol{a}_i\|_4^4 = 1.$$

To prove there is no other maximizer than columns of $\boldsymbol{A}$, we observe that the first inequality of (F.1) holds with equality if and only if

$$(\boldsymbol{a}_j^T \boldsymbol{q})^2 = \frac{1}{s}, \qquad \forall j \in S$$

where

$$S = \{j \in [r] : \ \boldsymbol{a}_j^T \boldsymbol{q} \neq 0\}$$

and $s = |S|$. We thus have

$$\boldsymbol{q} = \frac{1}{\sqrt{s}} \sum_{j \in S} \boldsymbol{a}_j.$$

This choice of $\boldsymbol{q}$ leads to $\|\boldsymbol{A}^T \boldsymbol{q}\|_4^4 = 1/s$ which is equal to one if and only if $s = 1$. ∎

## F.2  Proof of Lemma 2.5

**Proof.** Pick any $\boldsymbol{q}$ and $C$. One has

$$\mathbb{E}\left[F(\boldsymbol{q})\right] = -\frac{1}{12\theta\sigma^4 n}\mathbb{E}\left[\|\boldsymbol{q}^T \boldsymbol{A}\boldsymbol{X}\|_4^4\right] = \frac{1}{12\theta\sigma^4}\mathbb{E}\left[|\boldsymbol{q}^T \boldsymbol{A}\boldsymbol{X}_{\cdot i}|^4\right] \tag{F.2}$$

by the i.i.d. assumption of columns of $\boldsymbol{X}$. Write $\boldsymbol{\zeta} = \boldsymbol{A}^T \boldsymbol{q}$ and use Assumption 2.1 to obtain

$$\mathbb{E}\left[F(\boldsymbol{q})\right] = -\frac{1}{12\theta\sigma^4}\mathbb{E}\left[\left(\sum_{j=1}^{r} \zeta_j \boldsymbol{B}_{ji} \boldsymbol{Z}_{ji}\right)^4\right].$$

Since $\boldsymbol{B}_{ji}$ is independent of $\boldsymbol{Z}_{ji}$ and

$$\sum_{j=1}^{r} \zeta_j \boldsymbol{B}_{ji} \boldsymbol{Z}_{ji}\Big|\boldsymbol{B}_{\cdot i} \sim N\left(0, \sigma^2 \sum_{j=1}^{r} \zeta_j^2 \boldsymbol{B}_{ji}^2\right)$$

from Assumption 2.1, we obtain

$$\mathbb{E}\left[\left(\sum_{j=1}^{r} \zeta_j \boldsymbol{B}_{ji} \boldsymbol{Z}_{ji}\right)^4\right] = 3\sigma^4 \mathbb{E}\left[\left(\sum_{j=1}^{r} \zeta_j^2 \boldsymbol{B}_{ji}^2\right)^2\right] \tag{F.3}$$

$$= 3\sigma^4 \mathbb{E}\left[\sum_{j=1}^{r} \zeta_j^4 \boldsymbol{B}_{ji}^4\right] + 3\sigma^4 \mathbb{E}\left[\sum_{j \neq \ell} \zeta_j^2 \zeta_\ell^2 \boldsymbol{B}_{ji}^2 \boldsymbol{B}_{\ell i}^2\right] \tag{F.4}$$

$$= 3\sigma^4 \theta \sum_{j=1}^{r} \zeta_j^4 + 3\sigma^4 \theta^2 \sum_{j \neq \ell} \zeta_j^2 \zeta_\ell^2 \tag{F.5}$$

$$= 3\sigma^4 \theta \left[(1-\theta) \sum_{j=1}^{r} \zeta_j^4 + \theta \left(\sum_{j=1}^{r} \zeta_j^2\right)^2\right] \tag{F.6}$$

$$= 3\sigma^4 \theta \left[(1-\theta)\|\boldsymbol{\zeta}\|_4^4 + \theta\|\boldsymbol{\zeta}\|_2^4\right]. \tag{F.7}$$

The result then follows. ∎

## F.3  Proof of Theorem 3.1

We prove Theorem 3.1 by proving Lemmas 3.2 and 3.3 in Sections F.3.2 and F.3.1, respectively.

To analyze the solution to (2.5), we need the following Riemannian gradient and Hessian matrix of $f(\boldsymbol{q})$ constrained on the sphere $\|\boldsymbol{q}\|_2 = 1$

$$\operatorname{grad} f(\boldsymbol{q}) = -P_{\boldsymbol{q}^\perp} \left[ (1-\theta) \sum_{j=1}^r \boldsymbol{a}_j (\boldsymbol{q}^T \boldsymbol{a}_j)^3 + \theta \|\boldsymbol{q}^T \boldsymbol{A}\|_2^2 \boldsymbol{A}\boldsymbol{A}^T \boldsymbol{q} \right] \tag{F.8}$$

$$\operatorname{Hess} f(\boldsymbol{q}) = \operatorname{Hess}_{\ell_4} f(\boldsymbol{q}) + \operatorname{Hess}_{\ell_2} f(\boldsymbol{q}) \tag{F.9}$$

where

$$\operatorname{Hess}_{\ell_4} f(\boldsymbol{q}) = -(1-\theta)P_{\boldsymbol{q}^\perp} \left[ 3\sum_{j=1}^r \boldsymbol{a}_j \boldsymbol{a}_j^T (\boldsymbol{q}^T \boldsymbol{a}_j)^2 - \|\boldsymbol{q}^T \boldsymbol{A}\|_4^4 \boldsymbol{I} \right] P_{\boldsymbol{q}^\perp}, \tag{F.10}$$

$$\operatorname{Hess}_{\ell_2} f(\boldsymbol{q}) = -\theta P_{\boldsymbol{q}^\perp} \left[ \|\boldsymbol{q}^T \boldsymbol{A}\|_2^2 \boldsymbol{A}\boldsymbol{A}^T + 2\boldsymbol{A}\boldsymbol{A}^T \boldsymbol{q}\boldsymbol{q}^T \boldsymbol{A}\boldsymbol{A}^T - \|\boldsymbol{q}^T \boldsymbol{A}\|_2^4 \boldsymbol{I} \right] P_{\boldsymbol{q}^\perp}. \tag{F.11}$$

Recall that, for any $C_\star \in (0,1)$, we partition $\mathbb{S}^{p-1}$ into

$$R_1(C_\star) = \left\{ \boldsymbol{q} \in \mathbb{S}^{p-1} : \left\| \boldsymbol{A}^T \boldsymbol{q} \right\|_\infty^2 \geq C_\star \right\}, \qquad R_2(C_\star) = \mathbb{S}^{p-1} \setminus \Big( R_1(C_\star) \cup R_0 \Big).$$

### F.3.1 Geometric Analysis for $q \in R_2$

We prove the following lemma which shows the existence of negative curvature for any $\boldsymbol{q} \in R_2$.

**Lemma F.1** *Assume $\theta < 1/3$. For any point $\boldsymbol{q} \in R_2(C_\star)$ with*

$$C_\star \leq \frac{1-3\theta}{2},$$

*there exists $\boldsymbol{v}$ such that*

$$\boldsymbol{v}^T \operatorname{Hess} f(\boldsymbol{q}) \boldsymbol{v} < 0. \tag{F.12}$$

*In particular, if $\theta \leq 1/6$, for any point $\boldsymbol{q} \in R_2(C_\star)$ with*

$$C_\star \leq \frac{1}{3\sqrt{2}},$$

*there exists $\boldsymbol{v}$ such that*

$$\boldsymbol{v}^T \operatorname{Hess} f(\boldsymbol{q}) \boldsymbol{v} < -\frac{11 - 5\sqrt{2}}{9} \|\boldsymbol{\zeta}\|_\infty^2. \tag{F.13}$$

**Proof.** Fix $C_\star$. Pick any $\boldsymbol{q} \in R_2(C_\star)$ and write $\boldsymbol{\zeta} = \boldsymbol{A}^T \boldsymbol{q}$ for simplicity. Assume $|\zeta_i| = \|\boldsymbol{\zeta}\|_\infty$ for some $i \in [r]$. Recall that $D_{\boldsymbol{\zeta}}^{\circ 2} = \operatorname{diag}\left(\boldsymbol{\zeta}^{\circ 2}\right)$ with $\boldsymbol{\zeta}^{\circ 2} = \{\zeta_j^2\}_{j \in [r]}$. From (F.10), we have

$$
\begin{aligned}
&\boldsymbol{a}_i^T \operatorname{Hess}_{\ell_4} f(\boldsymbol{q}) \boldsymbol{a}_i \\
&= (1-\theta) \left[ -3\boldsymbol{a}_i^T \boldsymbol{A} D_{\boldsymbol{\zeta}}^{\circ 2} \boldsymbol{A}^T \boldsymbol{a}_i + 6\zeta_i \boldsymbol{\zeta}^T D_{\boldsymbol{\zeta}}^{\circ 2} \boldsymbol{A}^T \boldsymbol{a}_i - 3\zeta_i^2 \|\boldsymbol{\zeta}\|_4^4 - \|\boldsymbol{\zeta}\|_4^4 (\zeta_i^2 - \|\boldsymbol{a}_i\|_2^2) \right] \\
&= (1-\theta) \left[ -3\zeta_i^2 + 6\zeta_i^4 - 3\zeta_i^2 \|\boldsymbol{\zeta}\|_4^4 - \|\boldsymbol{\zeta}\|_4^4 (\zeta_i^2 - 1) \right] \\
&= (1-\theta) \left[ -3\|\boldsymbol{\zeta}\|_\infty^2 + 6\|\boldsymbol{\zeta}\|_\infty^4 - 4\|\boldsymbol{\zeta}\|_\infty^2 \|\boldsymbol{\zeta}\|_4^4 + \|\boldsymbol{\zeta}\|_4^4 \right] \\
&\leq (1-\theta) \left[ -2\|\boldsymbol{\zeta}\|_\infty^2 + 6\|\boldsymbol{\zeta}\|_\infty^4 - 4\|\boldsymbol{\zeta}\|_\infty^6 \right] \tag{F.14}
\end{aligned}
$$

where in the last line we used $\|\boldsymbol{\zeta}\|_4^4 \leq \|\boldsymbol{\zeta}\|_2^2 \|\boldsymbol{\zeta}\|_\infty^2 \leq \|\boldsymbol{\zeta}\|_\infty^2$ and $\|\boldsymbol{\zeta}\|_4^4 \geq \|\boldsymbol{\zeta}\|_\infty^4$. On the other hand, we obtain

$$
\begin{aligned}
\boldsymbol{a}_i^T \operatorname{Hess}_{\ell_2} f(\boldsymbol{q}) \boldsymbol{a}_i &= \theta \left[ -2\zeta_i^2 + 6 \|\boldsymbol{\zeta}\|_\infty \|\boldsymbol{\zeta}\|_2^2 \zeta_i - 4 \|\boldsymbol{\zeta}\|_\infty^2 \|\boldsymbol{\zeta}\|_2^4 - \|\boldsymbol{\zeta}\|_2^2 \left\| \boldsymbol{a}_i^T \boldsymbol{A} \right\|_2^2 + \|\boldsymbol{\zeta}\|_2^4 \right] \\
&\leq \theta \left[ -2 \|\boldsymbol{\zeta}\|_\infty^2 + 6 \|\boldsymbol{\zeta}\|_\infty^2 - 4 \|\boldsymbol{\zeta}\|_\infty^6 + \|\boldsymbol{\zeta}\|_2^2 \left( \|\boldsymbol{\zeta}\|_2^2 - 1 \right) \right]
\end{aligned}
$$

$$\leq \theta \left[ 4 \left\| \boldsymbol{\zeta} \right\|_\infty^2 - 4 \left\| \boldsymbol{\zeta} \right\|_\infty^6 \right] \tag{F.15}$$

where in the second and third lines we used $\left\| \boldsymbol{\zeta} \right\|_2^2 \leq 1$. Combine (F.14) and (F.15) to obtain

$$\boldsymbol{a}_i^T \operatorname{Hess} f(\boldsymbol{q}) \boldsymbol{a}_i \leq -4 \left\| \boldsymbol{\zeta} \right\|_\infty^6 + 6 \left( 1 - \theta \right) \left\| \boldsymbol{\zeta} \right\|_\infty^4 - 2(1 - 3\theta) \left\| \boldsymbol{\zeta} \right\|_\infty^2$$

$$= -4 \left\| \boldsymbol{\zeta} \right\|_\infty^2 \left\{ \left\| \boldsymbol{\zeta} \right\|_\infty^4 - \frac{3 \left( 1 - \theta \right)}{2} \left\| \boldsymbol{\zeta} \right\|_\infty^2 + \frac{1 - 3\theta}{2} \right\} \tag{F.16}$$

Define

$$g\left( x \right) = x^2 - \phi x + \omega, \qquad \text{with} \quad \phi = \frac{3 \left( 1 - \theta \right)}{2}, \quad \omega = \frac{1 - 3\theta}{2}. \tag{F.17}$$

It remains to prove $\boldsymbol{a}_i^T \operatorname{Hess} f(\boldsymbol{q}) \boldsymbol{a}_i \leq -4 \left\| \boldsymbol{\zeta} \right\|_\infty^2 g \left( \left\| \boldsymbol{\zeta} \right\|_\infty^2 \right) < 0$. To this end, note that $\omega > 0$ under $\theta < 1/3$. Since

$$\phi^2 - 4\omega = \frac{9 \left( 1 - \theta \right)^2}{4} - 2 + 6\theta = \left( \frac{3\theta + 1}{2} \right)^2 > 0, \tag{F.18}$$

we know that, for all

$$\left\| \boldsymbol{\zeta} \right\|_\infty^2 \leq \frac{\phi - \sqrt{\phi^2 - 4\omega}}{2} = \frac{1 - 3\theta}{2}, \tag{F.19}$$

$g(\left\| \boldsymbol{\zeta} \right\|_\infty^2) \geq 0$ and $g(\left\| \boldsymbol{\zeta} \right\|_\infty^2)$ increases as $\left\| \boldsymbol{\zeta} \right\|_\infty^2$ gets smaller. Recall that $\boldsymbol{q} \in R_2(C_\star)$ implies $\left\| \boldsymbol{\zeta} \right\|_\infty^2 < C_\star$. Thus, as long as

$$C_\star \leq \frac{1 - 3\theta}{2},$$

we conclude $g(\left\| \boldsymbol{\zeta} \right\|_\infty^2) > g(C_\star) \geq 0$ hence

$$\boldsymbol{a}_i^T \operatorname{Hess} f(\boldsymbol{q}) \boldsymbol{a}_i \leq -4 \left\| \boldsymbol{\zeta} \right\|_\infty^2 g \left( \left\| \boldsymbol{\zeta} \right\|_\infty^2 \right) < 0. \tag{F.20}$$

This completes the proof of the first statement. The second one follows by taking $C_\star \leq 1/(3\sqrt{2})$. ∎

### F.3.2 Geometric analysis for $q \in R_1$

In this section we prove that any local solution to (2.5) in $R_1$ recovers one column of $\boldsymbol{A}$, as stated in the following lemma.

**Lemma F.2** *Assume $\theta < 1$. Any local solution $\bar{\boldsymbol{q}} \in R_1(C_\star)$ to (2.5) with*

$$C_\star > \frac{1}{2} \sqrt{\frac{\theta}{1 - \theta}}$$

*recovers one column of $\boldsymbol{A}$, that is,*

$$\bar{\boldsymbol{q}} = \pm \boldsymbol{A} \boldsymbol{e}_i$$

*for some standard basis vector $\boldsymbol{e}_i$.*

**Proof.** We prove the result by showing that any critical point of (2.5) in $R_1(C_\star)$ is either a saddle point, or it satisfies the second order optimality condition and is equal to one column of $\boldsymbol{A}$.

Our proof starts by characterizing all critical points of (2.5). For any critical point $\boldsymbol{q}$ of (2.5), by writing $\boldsymbol{\zeta} = \boldsymbol{A}^T \boldsymbol{q}$, letting the gradient (F.8) equal to zero gives

$$(1 - \theta) \boldsymbol{A} \boldsymbol{\zeta}^{\circ 3} - (1 - \theta) \boldsymbol{q} \left\| \boldsymbol{\zeta} \right\|_4^4 + \theta \left\| \boldsymbol{\zeta} \right\|_2^2 \boldsymbol{A} \boldsymbol{\zeta} - \theta \boldsymbol{q} \left\| \boldsymbol{\zeta} \right\|_2^4 = 0. \tag{F.21}$$

Pick any $1 \leq i \leq r$. Multiply both sides by $\boldsymbol{a}_i^T$ to obtain

$$(1 - \theta) \boldsymbol{a}_i^T \boldsymbol{A} \boldsymbol{\zeta}^{\circ 3} - (1 - \theta) \zeta_i \left\| \boldsymbol{\zeta} \right\|_4^4 + \theta \left\| \boldsymbol{\zeta} \right\|_2^2 \boldsymbol{a}_i^T \boldsymbol{A} \boldsymbol{\zeta} - \theta \left\| \boldsymbol{\zeta} \right\|_2^4 \zeta_i = 0 \tag{F.22}$$

with $\boldsymbol{\zeta}^{\circ 3}$ means $\{\zeta_j^3\}_{j \in [r]}$. By using

$$\boldsymbol{a}_i^T \boldsymbol{A} \boldsymbol{\zeta}^{\circ 3} = \|\boldsymbol{a}_i\|_2^2 \, \zeta_i^3 + \sum_{j \neq i} \langle \boldsymbol{a}_i, \boldsymbol{a}_j \rangle \, \zeta_j^3 = \zeta_i^3 \tag{F.23}$$

$$\boldsymbol{a}_i^T \boldsymbol{A} \boldsymbol{\zeta} = \|\boldsymbol{a}_i\|_2^2 \, \zeta_i + \sum_{j \neq i} \langle \boldsymbol{a}_i, \boldsymbol{a}_j \rangle \, \zeta_j = \zeta_i, \tag{F.24}$$

under Assumption 2.3, after a bit algebra and rearrangement, we obtain

$$\zeta_i^3 - \alpha \zeta_i = 0 \tag{F.25}$$

where

$$\alpha = \|\boldsymbol{\zeta}\|_4^4 + \frac{\theta}{1-\theta} \left( \|\boldsymbol{\zeta}\|_2^2 - 1 \right) \|\boldsymbol{\zeta}\|_2^2. \tag{F.26}$$

We then have that, for any critical point $\boldsymbol{q} \in R_1$, $\boldsymbol{\zeta} = \boldsymbol{A}^T \boldsymbol{q}$ satisfies (F.25) for all $1 \leq i \leq r$. Furthermore, since Lemma F.5, stated and proved in Section F.3.3, shows that $\alpha > 0$, we conclude that $\boldsymbol{\zeta}$ belongs to one of the following three cases:

1. **Case 1**: $\|\boldsymbol{\zeta}\|_\infty = 0$;
2. **Case 2**: There exists $i \in [r]$ such that

$$|\zeta_i| = \sqrt{\alpha}, \qquad \zeta_j = 0, \quad \forall j \in [r] \setminus \{i\};$$

3. **Case 3**: There exists at least $i, j \in [r]$ with $i \neq j$ such that

$$|\zeta_i| = |\zeta_j| = \sqrt{\alpha}.$$

Note that the definition of $R_1$ excludes $R_0$ defined in (3.3), hence rules out **Case 1**. We then provide analysis for the other two cases separately. Specifically, for any $\boldsymbol{\zeta}$ belonging to **Case 2**, Lemma F.3 below proves that $\boldsymbol{\zeta}$ satisfies the second order optimality condition, hence is a local solution. Furthermore, $\boldsymbol{\zeta}$ is equal to one column of $\boldsymbol{A}$ up to the sign.

**Lemma F.3** *Let $\boldsymbol{q}$ be any critical point in $R_1(C_\star)$ and let $\boldsymbol{\zeta} = \boldsymbol{A}^T \boldsymbol{q}$. If there exists $i \in [r]$ such that*

$$|\zeta_i| = \sqrt{\alpha}, \qquad |\zeta_j| = 0, \quad \forall j \in [r] \setminus \{i\},$$

*with $\alpha$ defined in (F.26), then there exists some signed permutation $\boldsymbol{P}$ such that*

$$\boldsymbol{q} = \boldsymbol{A} \boldsymbol{P}_{\cdot 1} \tag{F.27}$$

*Furthermore,*

$$\boldsymbol{v}^T \operatorname{Hess} f(\boldsymbol{q}) \boldsymbol{v} \geq (1-\theta) \|P_{\boldsymbol{q}}^\perp \boldsymbol{v}\|_2^2, \qquad \forall \boldsymbol{v} \text{ such that } P_{\boldsymbol{q}}^\perp \boldsymbol{v} \neq 0. \tag{F.28}$$

**Proof.** Lemma F.3 is proved in Section F.3.4. ∎

Finally, we show in Lemma F.4 below that any $\boldsymbol{\zeta}$ belonging to **Case 3** is a saddle point, hence is not a local solution.

**Lemma F.4** *For any critical point $\boldsymbol{q} \in R_1(C_\star)$ with $\boldsymbol{\zeta} = \boldsymbol{A}^T \boldsymbol{q}$ and $\alpha$ as defined in (F.26), if there exists $k$ $(k \geq 2)$ non-zero elements such that*

$$|\zeta_{\pi(1)}| = |\zeta_{\pi(2)}| = \cdots = |\zeta_{\pi(k)}| = \sqrt{\alpha}$$

*for some permutation $\pi : [r] \to [r]$, then there exists $\boldsymbol{v}$ with $P_{\boldsymbol{q}}^\perp \boldsymbol{v} \neq 0$ such that*

$$\boldsymbol{v}^T \operatorname{Hess} f(\boldsymbol{q}) \boldsymbol{v} \leq -\frac{2(1-\theta)}{k} \|P_{\boldsymbol{q}}^\perp \boldsymbol{v}\|_2^2 < 0. \tag{F.29}$$

**Proof.** Lemma F.4 is proved in Section F.3.5. ∎

Summarizing the above two lemmas conclude that all local solutions in $R_1$ lie in **Case 2**, hence completes the proof of Lemma 3.2. ∎

### F.3.3 Additional lemmas used in Section F.3.2

The following lemma gives the upper and low bounds for $\alpha$ defined in equation (F.26).

**Lemma F.5** *For any $\boldsymbol{q} \in R_1(C_\star)$, let $\boldsymbol{\zeta} = \boldsymbol{A}^T \boldsymbol{q}$ and $\alpha$ be defined in (F.26). We have*

$$\|\boldsymbol{\zeta}\|_4^4 \left[ 1 - \frac{\theta}{4(1-\theta)C_\star^2} \right] \le \alpha \le \|\boldsymbol{\zeta}\|_4^4. \tag{F.30}$$

*As a result, when*

$$C_\star^2 > \frac{\theta}{4(1-\theta)},$$

*we have $\alpha > 0$.*

**Proof.** The upper bound of $\alpha$ follows from

$$\alpha = \|\boldsymbol{\zeta}\|_4^4 \left[ 1 + \frac{\theta}{(1-\theta)\|\boldsymbol{\zeta}\|_4^4} \left( \|\boldsymbol{\zeta}\|_2^4 - \|\boldsymbol{\zeta}\|_2^2 \right) \right] \le \|\boldsymbol{\zeta}\|_4^4$$

by using $\|\boldsymbol{\zeta}\|_2^2 \le 1$ and $\|\boldsymbol{\zeta}\|_2^4 \le \|\boldsymbol{\zeta}\|_2^2$. To prove the lower bound, we have

$$\alpha = \|\boldsymbol{\zeta}\|_4^4 \left[ 1 + \frac{\theta}{(1-\theta)\|\boldsymbol{\zeta}\|_4^4} \|\boldsymbol{\zeta}\|_2^2 \left( \|\boldsymbol{\zeta}\|_2^2 - 1 \right) \right] \tag{F.31}$$

$$= \|\boldsymbol{\zeta}\|_4^4 \left[ 1 - \frac{\theta}{(1-\theta)\|\boldsymbol{\zeta}\|_\infty^4} \|\boldsymbol{\zeta}\|_2^2 \left( 1 - \|\boldsymbol{\zeta}\|_2^2 \right) \right] \qquad \text{by } \|\boldsymbol{\zeta}\|_2^2 \le 1 \tag{F.32}$$

$$\ge \|\boldsymbol{\zeta}\|_4^4 \left[ 1 - \frac{\theta}{4(1-\theta)\|\boldsymbol{\zeta}\|_\infty^4} \right] \qquad \text{by } \|\boldsymbol{\zeta}\|_2^2(1 - \|\boldsymbol{\zeta}\|_2^2) \le 1/4 \tag{F.33}$$

$$\ge \|\boldsymbol{\zeta}\|_4^4 \left[ 1 - \frac{\theta}{4(1-\theta)C_\star^2} \right] \tag{F.34}$$

The last inequality uses $\|\boldsymbol{\zeta}\|_\infty^2 \ge C_\star$ from the definition of $R_1(C_\star)$.

To prove the lower bound, we have

$$\alpha = \|\boldsymbol{\zeta}\|_4^4 \left[ 1 + \frac{\theta}{(1-\theta)\|\boldsymbol{\zeta}\|_4^4} \left( \|\boldsymbol{\zeta}\|_2^4 - \|\boldsymbol{\zeta}\|_2^2 \right) \right] \tag{F.35}$$

$$= \|\boldsymbol{\zeta}\|_4^4 \left[ 1 + \frac{\theta}{(1-\theta)} \left( \frac{\|\boldsymbol{\zeta}\|_2^4}{\|\boldsymbol{\zeta}\|_4^4} - \frac{\|\boldsymbol{\zeta}\|_2^2}{\|\boldsymbol{\zeta}\|_4^4} \right) \right] \tag{F.36}$$

$$\ge \|\boldsymbol{\zeta}\|_4^4 \left[ 1 + \frac{\theta}{1-\theta} \left( 1 - \frac{1}{C_\star^2} \right) \right] \tag{F.37}$$

The last inequality uses

$$\frac{\|\boldsymbol{\zeta}\|_2^4}{\|\boldsymbol{\zeta}\|_4^4} \ge \frac{\|\boldsymbol{\zeta}\|_2^4}{\|\boldsymbol{\zeta}\|_\infty^2 \|\boldsymbol{\zeta}\|_2^2} = \frac{\|\boldsymbol{\zeta}\|_2^2}{\|\boldsymbol{\zeta}\|_\infty^2} \ge 1 \tag{F.38}$$

and

$$\frac{\|\boldsymbol{\zeta}\|_2^2}{\|\boldsymbol{\zeta}\|_4^4} \le \frac{1}{\|\boldsymbol{\zeta}\|_\infty^4} \le \frac{1}{C_\star^2}. \tag{F.39}$$

by the definition of $R_1(C_\star)$. ∎

### F.3.4 Proof of Lemma F.3

**Proof.** Let $\boldsymbol{q}$ be any critical point in $R_1(C_\star)$ with $C_\star > 0$. Write $\boldsymbol{\zeta} = \boldsymbol{A}^T \boldsymbol{q}$ and suppose

$$|\zeta_\ell| = \sqrt{\alpha}, \qquad |\zeta_j| = 0, \quad \forall j \in [r] \setminus \{\ell\},$$

with $\alpha$ defined in (F.26). Our proof contains two parts. We first show that $\boldsymbol{q} = \boldsymbol{a}_l$ (we assume $\boldsymbol{P}$ is identity for simplicity) and then show that $\boldsymbol{q}$ satisfies the second order optimality condition.

**Recovery of $a_\ell$:** First notice that

$$\zeta_l^2 = \alpha = \|\boldsymbol{\zeta}\|_4^4 \left[ 1 + \frac{\theta}{\|\boldsymbol{\zeta}\|_4^4 (1-\theta)} \left( \|\boldsymbol{\zeta}\|_2^4 - \|\boldsymbol{\zeta}\|_2^2 \right) \right] \tag{F.40}$$

Since $\|\boldsymbol{\zeta}\|_2^2 = \zeta_l^2$ and $\|\boldsymbol{\zeta}\|_4^4 = \zeta_l^4$, we immediately have

$$\alpha = \alpha^2 \left[ 1 + \frac{\theta}{\alpha^2 (1-\theta)} \left( \alpha^2 - \alpha \right) \right]. \tag{F.41}$$

Solving it gives $\alpha = 1$, which implies $\zeta_\ell^2 = |\langle a_l, \boldsymbol{q} \rangle|^2 = 1$, as desired.

**Second order optimality:** We prove

$$\boldsymbol{v}^T \operatorname{Hess} f(\boldsymbol{q}) \boldsymbol{v} = \boldsymbol{v}^T \left[ \operatorname{Hess}_{\ell_2} f(\boldsymbol{q}) + \operatorname{Hess}_{\ell_4} f(\boldsymbol{q}) \right] \boldsymbol{v} > 0$$

for all $\boldsymbol{v}$ such that $P_{\boldsymbol{q}}^\perp \boldsymbol{v} \neq 0$.

Recall from (F.10) that

$$\operatorname{Hess}_{\ell_4} f(\boldsymbol{q}) = -(1-\theta) P_{\boldsymbol{q}^\perp} \left[ 3 \sum_{j=1}^r a_j a_j^T (\boldsymbol{q}^T a_j)^2 - \|\boldsymbol{q}^T \boldsymbol{A}\|_4^4 \boldsymbol{I} \right] P_{\boldsymbol{q}^\perp}.$$

Without loss of generality, let $\boldsymbol{v} \in \mathbb{S}^{p-1}$ be any vector such that $\boldsymbol{v} \perp \boldsymbol{q}$. Recall that $\boldsymbol{\zeta} = \boldsymbol{A}^T \boldsymbol{q}$. Then

$$\boldsymbol{v}^T \operatorname{Hess}_{\ell_4} f(\boldsymbol{q}) \boldsymbol{v} = (1-\theta) \left[ -3 \sum_{j=1}^r \left( a_j^T \boldsymbol{v} \right)^2 \zeta_j^2 + \|\boldsymbol{\zeta}\|_4^4 \right] \tag{F.42}$$

$$= (1-\theta) \left[ -3 \left( a_\ell^T \boldsymbol{v} \right)^2 + 1 \right]. \tag{F.43}$$

where we used $\zeta_\ell^2 = 1$ and $\zeta_j = 0$ for all $j \neq \ell$ together with $\|\boldsymbol{\zeta}\|_4^4 = 1$ in the second line. In addition, we find

$$\left( a_\ell^T \boldsymbol{v} \right)^2 = |\langle a_l, \boldsymbol{v} \rangle|^2 = |\langle \boldsymbol{q}, \boldsymbol{v} \rangle|^2 = 0 \tag{F.44}$$

so that

$$\boldsymbol{v}^T \operatorname{Hess}_{\ell_4} f(\boldsymbol{q}) \boldsymbol{v} = 1 - \theta. \tag{F.45}$$

On the other hand,

$$\begin{aligned}
\boldsymbol{v}^T \operatorname{Hess}_{\ell_2}(\boldsymbol{q}) \boldsymbol{v} &= \theta \left[ -2 \left( \boldsymbol{v}^T \boldsymbol{A} \boldsymbol{\zeta} \right)^2 - \|\boldsymbol{\zeta}\|_2^2 \|\boldsymbol{A} \boldsymbol{v}\|_2^2 + \|\boldsymbol{\zeta}\|_2^4 \right] \\
&= \theta \left[ -2 \left( a_l^T \boldsymbol{v} \zeta_l \right)^2 - \|\boldsymbol{A} \boldsymbol{v}\|_2^2 + 1 \right] \\
&= \theta \left[ 1 - \|\boldsymbol{A} \boldsymbol{v}\|_2^2 \right] \\
&\geq 0
\end{aligned} \tag{F.46}$$

where we used $\lambda_1(\boldsymbol{A} \boldsymbol{A}^T) \leq 1$ in the last line. Combine equation (F.45) and inequality (F.124) to obtain

$$\boldsymbol{v}^T \operatorname{Hess} f(\boldsymbol{q}) \boldsymbol{v} \geq 1 - \theta > 0, \tag{F.47}$$

completing the proof. ∎

### F.3.5 Proof of Lemma F.4

**Proof.** Let $q$ be any critical point $q \in R_1(C_\star)$ with $C_\star > 0$ and $\zeta = A^T q$ having at least $k$ non-zero entries for $2 \leq k \leq r$. Without loss of generality, we assume

$$|\zeta_j| = \sqrt{\alpha} \qquad \forall j \leq k, \qquad \zeta_j = 0 \qquad \forall j > k. \tag{F.48}$$

We show there exists $v$ such that

$$v^T \operatorname{Hess} f(q) v = v^T \left( \operatorname{Hess}_{\ell_2} f(q) + \operatorname{Hess}_{\ell_4} f(q) \right) v = -\frac{2(1-\theta)}{k} \| P_q^\perp v \|_2^2 < 0.$$

Without loss of generality, pick any vector $v \in \mathbb{S}^{p-1}$ satisfying $v \perp q$ and $v$ lies in the span of $\{a_1, a_2, \cdots, a_k\}$. Write $v = \sum_{j=1}^k c_j a_j$. From (F.9), we have

$$v^T \operatorname{Hess}_{\ell_4} f(q) v = (1-\theta) \left[ -3 v^T A D_\zeta^{\circ 2} A v + \| \zeta \|_4^4 \right] = (1-\theta) \left[ -3 \sum_{j=1}^k \left( a_j^T v \right)^2 \zeta_j^2 + \| \zeta \|_4^4 \right]. \tag{F.49}$$

Recall from the definition of $\alpha$ in (F.26) that

$$\alpha = \| \zeta \|_4^4 \left[ 1 + \frac{\theta}{\| \zeta \|_4^4 (1-\theta)} \left( \| \zeta \|_2^4 - \| \zeta \|_2^2 \right) \right], \tag{F.50}$$

using $\| \zeta \|_4^4 = \sum_{j=1}^r \zeta_j^4 = \sum_{j=1}^k \zeta_j^4 = k\alpha^2$ and $\| \zeta \|_2^2 = \sum_{j=1}^r \zeta_j^2 = \sum_{j=1}^k \zeta_j^2 = k\alpha$ yields

$$\alpha = k\alpha^2 \left[ 1 + \frac{\theta}{k\alpha^2 (1-\theta)} \left( k\alpha^2 - k\alpha \right) \right]. \tag{F.51}$$

Solve the equation above to obtain $\alpha = 1/k$, hence

$$\| \zeta \|_4^4 = \frac{1}{k}, \qquad |\zeta_j|^2 = \frac{1}{k}, \quad \forall j \leq k.$$

Plugging this into (F.49) gives

$$\begin{aligned}
v^T \operatorname{Hess}_{\ell_4} f(q) v &= (1-\theta) \left[ -\frac{3}{k} \sum_{j=1}^k \left( a_j^T v \right)^2 + \frac{1}{k^2} \right] \\
&= (1-\theta) \left[ -\frac{3}{k} \sum_{j=1}^k c_j^2 + \frac{1}{k} \right] \\
&= -\frac{2(1-\theta)}{k}
\end{aligned} \tag{F.52}$$

where the second equality used $\sum_{j=1}^k (a_j^T v)^2 = \sum_{j=1}^k c_j^2 = 1$.

On the other hand, we have

$$\begin{aligned}
v^T \operatorname{Hess}_{\ell_2} f(q) v &= \theta \left[ -2 \left( v^T A \zeta \right)^2 - \| \zeta \|_2^2 \| A^T v \|_2^2 + \| \zeta \|_2^4 \right] \\
&\leq \theta \left[ -\| \zeta \|_2^2 \| A^T v \|_2^2 + \| \zeta \|_2^4 \right] \\
&= \theta \left[ -\| \zeta \|_2^2 \sum_{j=1}^k c_j^2 + \| \zeta \|_2^4 \right] \\
&= 0.
\end{aligned} \tag{F.53}$$

The second equation follows from $\| A^T v \|_2^2 = \sum_{j=1}^k (v^T a_j)^2 = \sum_{j=1}^k c_j^2 = 1$ and the last step uses $\| \zeta \|_2^2 = 1$. Combining equation (F.52) and (F.94) gives

$$v^T \operatorname{Hess} f(q) v \leq -\frac{2(1-\theta)}{k} \tag{F.54}$$

and completes the proof. ∎

## F.4    Proof of Theorem 3.4

To prove Theorem 3.4, analogous to (3.3), we give a new partition of $\mathbb{S}^{p-1}$ as

$$R_0' \doteq R_0'(c_\star) = \left\{ \boldsymbol{q} \in \mathbb{S}^{p-1} : \left\| \boldsymbol{A}^T \boldsymbol{q} \right\|_\infty^2 \leq c_\star \right\}, \tag{F.55}$$

$$R_1' \doteq R_1(C_\star) = \left\{ \boldsymbol{q} \in \mathbb{S}^{p-1} : \left\| \boldsymbol{A}^T \boldsymbol{q} \right\|_\infty^2 \geq C_\star \right\},$$

$$R_2' = \mathbb{S}^{p-1} \setminus (R_0' \cup R_1').$$

Here $c_\star$ and $C_\star$ are positive constants satisfying $0 \leq c_\star \leq C_\star < 1$.

Let $\delta_1$ and $\delta_2$ be some positive sequences to be determined later. Define the random event

$$\mathcal{E} = \left\{ \sup_{\boldsymbol{q} \in \mathbb{S}^{p-1}} \left\| \operatorname{grad} f\left( \boldsymbol{q} \right) - \operatorname{grad} F\left( \boldsymbol{q} \right) \right\|_2 \lesssim \delta_1, \quad \sup_{\boldsymbol{q} \in \mathbb{S}^{p-1}} \left\| \operatorname{Hess} f\left( \boldsymbol{q} \right) - \operatorname{Hess} F\left( \boldsymbol{q} \right) \right\|_{\mathrm{op}} \lesssim \delta_2 \right\}. \tag{F.56}$$

Here $\operatorname{grad} f(\boldsymbol{q})$ and $\operatorname{grad} F(\boldsymbol{q})$ are the gradients of (2.5) and (2.3), respectively, at any point $\boldsymbol{q} \in \mathbb{S}^{p-1}$. Similarly, $\operatorname{Hess} f(\boldsymbol{q})$ and $\operatorname{Hess} F(\boldsymbol{q})$ are the corresponding Hessian matrices.

On the event $\mathcal{E}$, the results of Theorem 3.4 immediately follow from the two lemmas below. Lemma F.6 shows that the objective $F\left(\boldsymbol{q}\right)$ in (2.3) exhibits negative curvature at any point $\boldsymbol{q} \in R_2'$. Meanwhile, Lemma F.7 proves that any critical point in region $R_1'$ is either a solution that is close to the ground-truth, or a saddle point with negative curvature that is easy to escape by any second-order descent algorithm. Lemmas F.6 and F.7 are proved in Section F.4.1 and F.4.2, respectively.

**Lemma F.6 (Optimization landscape for $R_2'$)**  *Assume $\theta \leq 6/25$. For any point $\boldsymbol{q} \in R_2'(C_\star)$ with*

$$C_\star \leq \frac{12}{25} - 2\theta,$$

*if $\delta_2 \leq c_\star/25$ for some constant $c_\star \in (0, C_\star)$, then there exists $\boldsymbol{v}$ such that*

$$\boldsymbol{v}^T \operatorname{Hess} F\left( \boldsymbol{q} \right) \boldsymbol{v} < 0. \tag{F.57}$$

*In particular, if $\theta \leq 1/9$, for any point $\boldsymbol{q} \in R_2(C_\star)$ with $C_\star \leq 1/4$, there exists $\boldsymbol{v}$ such that*

$$\boldsymbol{v}^T \operatorname{Hess} F\left( \boldsymbol{q} \right) \boldsymbol{v} < -\frac{21}{100} \left\| \boldsymbol{\zeta} \right\|_\infty^2 < -\frac{1}{5} c_\star. \tag{F.58}$$

**Lemma F.7 (Optimization landscape for $R_1'$)**  *Assume*

$$\theta \leq 1/9, \quad \delta_1 \leq 5 \times 10^{-5}, \quad \delta_2 \leq 10^{-3}. \tag{F.59}$$

*Any local solution $\bar{\boldsymbol{q}} \in R_1'(C_\star)$ with $C_\star \geq 1/5$ satisfies*

$$\left\| \bar{\boldsymbol{q}} - \boldsymbol{A} \boldsymbol{P}_{\cdot 1} \right\|_2^2 \leq C \delta_1 \tag{F.60}$$

*for some signed permutation matrix $\boldsymbol{P}$ and some constant $C > 0$.*

Finally, the proof of Theorem 3.4 is completed by invoking Lemmas F.14 and F.15 and using condition (3.7) to establish that $\delta_2 \leq c_\star/25$, $\delta_2 \leq 10^{-3}$ and $\delta_1 \leq 5 \times 10^{-5}$. Indeed, we have

$$\delta_1 = \sqrt{\frac{r^2 \log(M_n)}{\theta n}} + \frac{M_n}{n} \frac{r \log(M_n)}{n}, \tag{F.61}$$

$$\delta_2 = \sqrt{\frac{r^3 \log(M_n)}{\theta n}} + \frac{M_n}{n} \frac{r \log(M_n)}{n} \tag{F.62}$$

where $M_n = C(n + r) \left( \theta r^2 + \log^2 n/\theta \right)$. Under (3.7), we have $\log(M_n) \lesssim \log n$ whence $\delta_2 \leq \min\{c_\star/25, 10^{-3}\}$ requires

$$n \geq C \max \left\{ \frac{r^3 \log n}{\theta c_\star^2}, \left( \theta r^2 + \frac{\log^2 n}{\theta} \right) \frac{r \log n}{c_\star} \right\}$$

for sufficiently large $C > 0$, which holds under (3.7).

### F.4.1 Proof of Lemma F.6

**Proof.** Fix $C_\star$. Pick any $q \in R_2'(C_\star)$ and write $\zeta = A^T q$ for simplicity. Assume $|\zeta_i| = \|\zeta\|_\infty$ for some $i \in [r]$. Note that, on the event $\mathcal{E}$,

$$a_i^T \operatorname{Hess} F(q) a_i \leq a_i^T \operatorname{Hess} f(q) a_i + a_i^T \left[ \operatorname{Hess} F(q) - \operatorname{Hess} f(q) \right] a_i \qquad (\text{F.63})$$

$$\leq a_i^T \operatorname{Hess} f(q) a_i + \delta_2 \qquad (\text{F.64})$$

$$\leq a_i^T \operatorname{Hess} f(q) a_i + \frac{1}{25} \|\zeta\|_\infty^2 \qquad (\text{F.65})$$

where in the last inequality we used $\delta_2 \leq c_\star/25 \leq \|\zeta\|_\infty^2/25$ as $q \in R_2'(C_\star)$. By inequality (F.16), we obtain

$$a_i^T \operatorname{Hess} f(q) a_i \leq -4 \|\zeta\|_\infty^2 \left\{ \|\zeta\|_\infty^4 - \frac{3(1-\theta)}{2} \|\zeta\|_\infty^2 + \frac{1-3\theta}{2} \right\}, \qquad (\text{F.66})$$

hence

$$a_i^T \operatorname{Hess} F(q) a_i \leq -4 \|\zeta\|_\infty^2 \left\{ \|\zeta\|_\infty^4 - \frac{3(1-\theta)}{2} \|\zeta\|_\infty^2 + \frac{1-3\theta}{2} - \frac{1}{100} \right\} \qquad (\text{F.67})$$

Define

$$g(x) = x^2 - \phi x + \omega, \qquad \text{with} \quad \phi = \frac{3(1-\theta)}{2}, \quad \omega = \frac{1-3\theta}{2} - \frac{1}{100}. \qquad (\text{F.68})$$

It remains to prove $a_i^T \operatorname{Hess} F(q) a_i \leq -4 \|\zeta\|_\infty^2 g\left(\|\zeta\|_\infty^2\right) < 0$. To this end, note that $\omega > 0$ under $\theta < 1/4$. Since

$$\phi^2 - 4\omega = \frac{9(1-\theta)^2}{4} - 2 + 6\theta + \frac{1}{25} = \left(\frac{3\theta+1}{2}\right)^2 + \frac{1}{25} > 0, \qquad (\text{F.69})$$

we know that, for all

$$\|\zeta\|_\infty^2 \leq \frac{\phi - \sqrt{\phi^2 - 4\omega}}{2} = \frac{\frac{3-3\theta}{2} - \left(\frac{3\theta+1}{2}\right)\sqrt{1 + \frac{1}{25\left(\frac{3\theta+1}{2}\right)^2}}}{2} \doteq r_- \qquad (\text{F.70})$$

$g(\|\zeta\|_\infty^2) \geq 0$ and $g(\|\zeta\|_\infty^2)$ increases as $\|\zeta\|_\infty^2$ gets smaller. Recall that $q \in R_2(C_\star)$ implies $\|\zeta\|_\infty^2 < C_\star$. We then have

$$g(\|\zeta\|_\infty^2) > g(C_\star).$$

We proceed to show $C_\star \leq r_-$ by noticing that

$$r_- \geq \frac{\frac{3-3\theta}{2} - \left(\frac{3\theta+1}{2}\right)\left[1 + \frac{1}{50\left(\frac{3\theta+1}{2}\right)^2}\right]}{2}$$

$$\geq \frac{\frac{3-3\theta}{2} - \left(\frac{3\theta+1}{2}\right)\left[1 + \frac{4}{50}\right]}{2}$$

$$= \frac{1 - 3\theta - \frac{12}{100}\theta - \frac{1}{25}}{2}$$

$$> \frac{12}{25} - 2\theta. \qquad (\text{F.71})$$

Thus, provided that

$$C_\star \leq \frac{12}{25} - 2\theta,$$

we conclude $g(\|\zeta\|_\infty^2) > g(C_\star) \geq 0$ hence

$$a_i^T \operatorname{Hess} F(q) a_i \leq -4 \|\zeta\|_\infty^2 g\left(\|\zeta\|_\infty^2\right) < 0. \qquad (\text{F.72})$$

In particular, taking $\theta \leq 1/9$ and $C_\star \leq 1/4$ yields

$$a_i^T \operatorname{Hess} F(q) a_i \leq -\frac{21}{100} \|\zeta\|_\infty^2. \qquad (\text{F.73})$$

This completes the proof. ∎

### F.4.2  Proof of Lemma F.7

**Proof.**  The proof of this lemma is similar in spirit to that of Lemma F.3.2. Follows the notations there, any critical point $q \in R_1'(C_\star)$ satisfies

$$\operatorname{grad} f(q) + \operatorname{grad} F(q) - \operatorname{grad} f(q) = 0. \tag{F.74}$$

Following the same procedure of proving Lemma F.3.2, analogous to (F.25), we obtain

$$\zeta_i^3 - \alpha\zeta_i + \beta = 0 \tag{F.75}$$

for any $i \in [r]$, where $\zeta = A^T q$,

$$\alpha = \|\zeta\|_4^4 + \frac{\theta}{1-\theta}\left(\|\zeta\|_2^4 - \|\zeta\|_2^2\right), \qquad \beta \doteq \beta_i = \langle \operatorname{grad} f(q) - \operatorname{grad} F(q), a_i \rangle. \tag{F.76}$$

To further characterize $\zeta$ satisfying (F.75), note that $\alpha > 0$ from Lemma F.5 and we also prove in Lemma F.11, stated and proved in Section F.4.3, that $4|\beta| < \alpha^{3/2}$. In conjunction with Lemma F.12 in Section F.4.3, we conclude that $\zeta$ belongs to one of the following three cases:

- **Case 1**:
$$|\zeta_i| \le \frac{2|\beta|}{\alpha}, \quad \forall 1 \le i \le r;$$

- **Case 2**: There exists $i \in [r]$ such that
$$|\zeta_i| \ge \sqrt{\alpha} - \frac{2|\beta|}{\alpha}, \qquad |\zeta_j| < \sqrt{\alpha} - \frac{2|\beta|}{\alpha}, \quad \forall j \in [r] \setminus \{i\};$$

- **Case 3**: There exists at least $i, j \in [r]$ with $i \ne j$ such that
$$|\zeta_i| \ge \sqrt{\alpha} - \frac{2|\beta|}{\alpha}, \qquad |\zeta_j| \ge \sqrt{\alpha} - \frac{2|\beta|}{\alpha}.$$

We provide analysis case by case. **Case 1** is ruled out by Lemma F.8 below. For any $\zeta$ belonging to **Case 2**, Lemma F.9 below proves that $\zeta$ satisfies the second order optimality condition, hence is a local solution. Furthermore, $q$ is close to one column of $A$. Finally, Lemma F.10 shows that any $\zeta$ belonging to **Case 3** is a saddle point, hence is not a local solution. Summarizing the Lemmas F.8 – F.10 concludes that all local solutions in $R_1$ lie in **Case 2**, hence concludes the proof of lemma F.7. Lemmas F.8 – F.10 are proved in Sections F.4.4, F.4.6 and F.4.5, respectively. ∎

**Lemma F.8** *Assume*
$$\theta \le 1/9, \qquad \delta_1 \le 10^{-4}.$$
*For any critical point $q \in R_1'(C_\star)$ with $C_\star \ge 1/5$, there exists at least one $i \in [r]$ such that*
$$|\zeta_i| > \frac{2|\beta|}{\alpha}$$
*where $\zeta = A^T q$ and $\alpha$ and $\beta$ are defined in (F.76).*

**Lemma F.9** *Assume*
$$\theta \le 1/9, \quad \delta_1 \le 5 \times 10^{-5}, \quad \delta_2 \le 10^{-3}. \tag{F.77}$$
*Let $q$ be any critical point in $R_1'(C_\star)$ with $C_\star \ge 1/5$. If there exists $i \in [r]$ such that*
$$|\zeta_i| \ge \sqrt{\alpha} - \frac{2|\beta|}{\alpha}, \qquad |\zeta_j| \le \frac{2|\beta|}{\alpha}, \quad \forall j \in [r] \setminus \{i\},$$
*with $\zeta = A^T q$ and $\alpha$ and $\beta$ defined in (F.76), then*
$$\|q - AP_{\cdot 1}\|_2^2 \le C\delta_1 \tag{F.78}$$
*for some signed permutation matrix $P$ and some constant $C > 0$. Furthermore,*
$$v^T \operatorname{Hess} f(q) v > 0, \qquad \forall v \text{ such that } P_q^\perp v \ne 0. \tag{F.79}$$

**Lemma F.10** *Assume*

$$\theta \leq 1/9, \qquad \delta_1 \leq 10^{-4}, \qquad \delta_2 \leq 10^{-3}. \qquad (F.80)$$

*For any critical point $\boldsymbol{q} \in R_1'(C_\star)$ with $C_\star \geq 1/5$, if there exists $i, j \in [r]$ with $i \neq j$ such that*

$$|\zeta_i| \geq \sqrt{\alpha} - \frac{2|\beta|}{\alpha}, \qquad |\zeta_j| \geq \sqrt{\alpha} - \frac{2|\beta|}{\alpha},$$

*where $\boldsymbol{\zeta} = \boldsymbol{A}^T \boldsymbol{q}$ and $\alpha$ and $\beta$ are defined in (F.76), then there exists $\boldsymbol{v}$ with $P_{\boldsymbol{q}}^\perp \boldsymbol{v} \neq 0$ such that*

$$\boldsymbol{v}^T \operatorname{Hess} f(\boldsymbol{q}) \boldsymbol{v} \leq -0.00315 \| P_{\boldsymbol{q}}^\perp \boldsymbol{v} \|_2^2 < 0. \qquad (F.81)$$

### F.4.3 Lemmas used in Section F.4.2

**Lemma F.11** *Assume $\theta \leq 1/9$. For any critical point $\boldsymbol{q} \in R_1'(C_\star)$ with $C_\star \geq 1/5$, on the event $\mathcal{E}$ in (F.56), we have*

$$4|\beta| < \alpha^{3/2}$$

*where $\beta$ and $\alpha$ are defined in (F.76).*

**Proof.** By definition and the event $\mathcal{E}$,

$$|\beta| = |\langle \operatorname{grad} f(\boldsymbol{q}) - \operatorname{grad} F(\boldsymbol{q}), \boldsymbol{a}_i \rangle| \leq \delta_1 \|\boldsymbol{a}_i\|_2 = \delta_1. \qquad (F.82)$$

Then by Lemma F.5,

$$\frac{|\beta|}{\alpha^{3/2}} \leq \frac{\delta_1}{\alpha^{3/2}} \leq \frac{\delta_1}{\|\boldsymbol{\zeta}\|_4^6 \left[1 - \frac{\theta}{4(1-\theta)C_\star^2}\right]^{\frac{3}{2}}}.$$

Since $\boldsymbol{q} \in R_1(C_\star)$ implies $\|\boldsymbol{\zeta}\|_\infty^2 \geq C_\star$, using $\|\boldsymbol{\zeta}\|_4^6 \geq \|\boldsymbol{\zeta}\|_\infty^6 \geq C_\star^3$ together with $C_\star \geq 1/5$ and $\theta \leq 1/9$ gives

$$\|\boldsymbol{\zeta}\|_4^6 \left[1 - \frac{\theta}{4(1-\theta)C_\star^2}\right]^{\frac{3}{2}} \geq \left[C_\star^2 - \frac{\theta}{4(1-\theta)}\right]^{\frac{3}{2}} \geq \left(\frac{7}{800}\right)^{3/2} \qquad (F.83)$$

The result follows from $\delta_1 < 2 \times 10^{-4}$. ∎

**Lemma F.12 (Lemma $B.3$, [35])** *Considering the cubic function*

$$f(x) = x^3 - \alpha x + \beta \qquad (F.84)$$

*When $\alpha \geq 0$ and $4|\beta| \leq \alpha^{3/2}$, the roots of the function $f(\cdot)$ are contained in the following union of the intervals.*

$$\left\{|x| \leq \frac{2|\beta|}{\alpha}\right\} \bigcup \left\{|x - \sqrt{\alpha}| \leq \frac{2|\beta|}{\alpha}\right\} \bigcup \left\{|x + \sqrt{\alpha}| \leq \frac{2|\beta|}{\alpha}\right\}. \qquad (F.85)$$

### F.4.4 Proof of Lemma F.8

**Proof.** We prove that for any critical point $\boldsymbol{q} \in R_1'$, there exists at least one $i \in [r]$ such that $|\zeta_i| > 2|\beta|/\alpha$ with $\alpha$ and $\beta$ being defined in (F.76). Suppose

$$|\zeta_i| \leq \frac{2|\beta|}{\alpha}, \quad \forall i \in [r].$$

Assume $|\zeta_k| = \|\boldsymbol{\zeta}\|_\infty$ for some $k \in [r]$. We obtain

$$\|\boldsymbol{\zeta}\|_\infty \leq \frac{2|\beta|}{\alpha}$$

hence, by also using $\|\boldsymbol{\zeta}\|_2 \leq 1$,

$$\|\boldsymbol{\zeta}\|_4^4 \leq \|\boldsymbol{\zeta}\|_\infty^2 \|\boldsymbol{\zeta}\|_2^2 \leq \frac{4\beta^2}{\alpha^2} \leq \frac{4\delta_1^2}{\|\boldsymbol{\zeta}\|_4^{12} \left[1 - \frac{\theta}{4(1-\theta)C_\star^2}\right]^2} \|\boldsymbol{\zeta}\|_4^4 \overset{(F.83)}{\leq} 4\delta_1^2 \left(\frac{800}{7}\right)^3 \|\boldsymbol{\zeta}\|_4^4. \qquad (F.86)$$

This is a contradiction whenever

$$\delta_1 \leq \frac{1}{2}\left(\frac{7}{800}\right)^{-3/2},$$

which is the case if $\delta_1 \leq 10^{-4}$. ∎

### F.4.5 Proof of Lemma F.10

**Proof.** Let $q$ be any critical point $q \in R'_1(C_\star)$ with $C_\star \geq 1/5$ and write $\zeta = A^T q$. Suppose there exists $l, m \in [r]$ with $l \neq m$ such that

$$|\zeta_l| > \sqrt{\alpha} - \frac{2|\beta|}{\alpha}, \qquad |\zeta_m| > \sqrt{\alpha} - \frac{2|\beta|}{\alpha}.$$

We prove there exist $v$ such that.

$$v^T \operatorname{Hess} F(q)v \leq v^T \operatorname{Hess} f(q)v + v^T \left[\operatorname{Hess} F(q) - \operatorname{Hess} f(q)\right] v < 0. \tag{F.87}$$

Pick any vector $v \in \mathbb{S}^{p-1}$ such that $v \perp q$ and $v$ lies in the span of $\{a_l, a_m\}$, that is, $v = c_l a_l + c_m a_m$ for some $c_l^2 + c_m^2 = 1$. Recall from (F.9) that

$$v^T \operatorname{Hess} f(q)v = v^T \operatorname{Hess}_{\ell_4} f(q)v + v^T \operatorname{Hess}_{\ell_2} f(q)v.$$

By (F.10), we first have

$$v^T \operatorname{Hess}_{\ell_4} f(q)v$$

$$= (1-\theta) \left[ -3\left(a_l^T v\right)^2 \zeta_l^2 - 3\left(a_m^T v\right)^2 \zeta_m^2 - 3 \sum_{k \neq l, k \neq m} \left(A^T v\right)_k^2 \zeta_k^2 + \|\zeta\|_4^4 \right]$$

$$\leq (1-\theta) \left\{ -3 \left[ \left(a_l^T v\right)^2 + \left(a_m^T v\right)^2 \right] \min\{\zeta_l^2, \zeta_m^2\} + \|\zeta\|_4^4 \right\}$$

$$\leq (1-\theta) \left\{ -3 \left[ c_l^2 + c_m^2 + (c_1^2 + c_2^2)\left(a_l^T a_m\right)^2 + 4c_l c_m \left(a_l^T a_m\right) \right] \min\{\zeta_l^2, \zeta_m^2\} + \|\zeta\|_4^4 \right\}$$

$$= (1-\theta) \left\{ -3 \min\left\{\zeta_l^2, \zeta_m^2\right\} + \|\zeta\|_4^4 \right\}. \tag{F.88}$$

Here $\|v\|_2^2 = c_l^2 + c_m^2 = 1$ and $a_l^T a_m = 0$ are used in last step derivation. Note that

$$\zeta_l^2 \geq \left(\sqrt{\alpha} - \frac{2|\beta|}{\alpha}\right)^2 \geq \alpha - \frac{4|\beta|}{\sqrt{\alpha}}. \tag{F.89}$$

Lemma F.5 gives

$$\alpha \geq \|\zeta\|_4^4 - \frac{\theta}{1-\theta}\left[\|\zeta\|_2^2 - \|\zeta\|_2^4\right].$$

Also by (F.82), we obtain

$$\frac{4|\beta|}{\sqrt{\alpha}} \leq \frac{4\delta_1}{\|\zeta\|_4^2\left[1 - \frac{\theta}{4(1-\theta)C_\star^2}\right]^{\frac{1}{2}}} \leq 4\delta_1\left(\frac{800}{7}\right)^{1/2} \leq \frac{4\delta_1}{C_\star^2}\left(\frac{800}{7}\right)^{1/2}\|\zeta\|_4^4 \tag{F.90}$$

where the second inequality is due to (F.83) and the last one uses $\|\zeta\|_4^4 \geq \|\zeta\|_\infty^4 \geq C_\star^2$. By writing

$$\eta \doteq \frac{4\delta_1}{C_\star^2}\left(\frac{800}{7}\right)^{1/2}, \tag{F.91}$$

it follows that

$$\zeta_l^2 \geq \|\zeta\|_4^4 (1-\eta) - \frac{\theta}{(1-\theta)}\left[\|\zeta\|_2^2 - \|\zeta\|_2^4\right]. \tag{F.92}$$

This lower bound also holds for $\min\{\zeta_l^2, \zeta_m^2\}$. Plugging it in (F.88) yields

$$v^T \operatorname{Hess}_{\ell_4} f(q)v \leq (1-\theta)\left\{-3\left\{\|\zeta\|_4^4(1-\eta) - \frac{\theta}{(1-\theta)}\left[\|\zeta\|_2^2 - \|\zeta\|_2^4\right]\right\} + \|\zeta\|_4^4\right\}$$

$$= (1-\theta)\left\{(-2 + 3\eta)\|\zeta\|_4^4 + \frac{3\theta}{1-\theta}\left[\|\zeta\|_2^2 - \|\zeta\|_2^4\right]\right\}. \tag{F.93}$$

On the other hand, from (F.10), we have

$$\boldsymbol{v}^T \operatorname{Hess}_{\ell_2} f(\boldsymbol{q})\boldsymbol{v} = \theta \left[ -2 \left( \boldsymbol{v}^T \boldsymbol{A}\boldsymbol{\zeta} \right)^2 - \|\boldsymbol{\zeta}\|_2^2 \left\| \boldsymbol{A}^T \boldsymbol{v} \right\|_2^2 + \|\boldsymbol{\zeta}\|_2^4 \right]$$

$$\leq \theta \left[ - \|\boldsymbol{\zeta}\|_2^2 \left\| \boldsymbol{A}^T \boldsymbol{v} \right\|_2^2 + \|\boldsymbol{\zeta}\|_2^4 \right]$$

$$= \theta \left[ - \|\boldsymbol{\zeta}\|_2^2 + \|\boldsymbol{\zeta}\|_2^4 \right]. \tag{F.94}$$

Third inequality uses $\left\| \boldsymbol{A}^T \boldsymbol{v} \right\|_2^2 = \sum_{j=1}^{r}(\boldsymbol{v}^T \boldsymbol{a}_j)^2 = c_l^2 + c_m^2 = \|\boldsymbol{v}\|_2^2 = 1$. Combine (F.94) and (F.93) to obtain

$$\boldsymbol{v}^T \operatorname{Hess} f(\boldsymbol{q})\boldsymbol{v} \leq (1 - \theta) \left\{ (-2 + 3\eta) \|\boldsymbol{\zeta}\|_4^4 + \frac{3\theta}{1 - \theta} \left[ \|\boldsymbol{\zeta}\|_2^2 - \|\boldsymbol{\zeta}\|_2^4 \right] \right\} + \theta \left[ - \|\boldsymbol{\zeta}\|_2^2 + \|\boldsymbol{\zeta}\|_2^4 \right] \tag{F.95}$$

$$\leq (1 - \theta) (-2 + 3\eta) \|\boldsymbol{\zeta}\|_4^4 + 2\theta \left[ \|\boldsymbol{\zeta}\|_2^2 - \|\boldsymbol{\zeta}\|_2^4 \right] \tag{F.96}$$

$$\leq (1 - \theta) (-2 + 3\eta) \|\boldsymbol{\zeta}\|_4^4 + \frac{\theta}{2}. \tag{F.97}$$

Here $\|\boldsymbol{\zeta}\|_2^2 - \|\boldsymbol{\zeta}\|_2^4 \leq 1/4$ is used in last step. We thus conclude that

$$\boldsymbol{v}^T \operatorname{Hess} F(\boldsymbol{q})\boldsymbol{v} \leq \boldsymbol{v}^T \operatorname{Hess}_f(\boldsymbol{q})\boldsymbol{v} + \|\operatorname{Hess} F(\boldsymbol{q}) - \operatorname{Hess} f(\boldsymbol{q})\|_{\operatorname{op}} \tag{F.98}$$

$$\leq (1 - \theta) (-2 + 3\eta) \|\boldsymbol{\zeta}\|_4^4 + \frac{\theta}{2} + \delta_2 \tag{F.99}$$

on the event $\mathcal{E}$. Note that $-2 + 3\eta \leq 0$ from $C_\star \geq 1/5$ and $\delta_1 < 10^{-4}$. By using $\|\boldsymbol{\zeta}\|_4^4 \geq \|\boldsymbol{\zeta}\|_\infty^4 \geq C_\star^2$, we obtain

$$\boldsymbol{v}^T \operatorname{Hess} F(\boldsymbol{q})\boldsymbol{v} \leq (1 - \theta) (-2 + 3\eta) C_\star^2 + \frac{\theta}{2} + \delta_2 \tag{F.100}$$

$$= (1 - \theta) \left( -2C_\star^2 + 12\delta_1 \left( \frac{800}{7} \right)^{1/2} \right) + \frac{\theta}{2} + \delta_2 \quad \text{by } (F.91). \tag{F.101}$$

Recalling that

$$\theta \leq 1/9, \quad C_\star \geq 1/5, \quad \delta_1 \leq 10^{-4}, \quad \delta_2 \leq 10^{-3},$$

we further have

$$\boldsymbol{v}^T \operatorname{Hess} F(\boldsymbol{q})\boldsymbol{v} \leq -0.00315 \|P_{\boldsymbol{q}}^\perp \boldsymbol{v}\|_2^2 = -0.00315 < 0. \tag{F.102}$$

This completes the proof. ∎

### F.4.6  Proof of Lemma F.9

**Proof.**  This proof contains two parts: the first part shows that any critical $\boldsymbol{q} \in R_1'$ is close to the ground truth vector $\boldsymbol{a}_l$ for some $l \in [m]$, and the second part proves the second order optimality for this $\boldsymbol{q}$.

**Closeness to the target ground-truth vector:**  Pick any critical point $\boldsymbol{q} \in R_1'$ and suppose that, for some $l \in [m]$,

$$\zeta_l \geq \sqrt{\alpha} - \frac{2|\beta|}{\alpha}, \quad \zeta_j \leq \frac{2|\beta|}{\alpha}, \quad \forall j \neq \ell.$$

On the one hand, we bound $\zeta_l^4$ from below as

$$\zeta_l^4 = \|\boldsymbol{\zeta}\|_4^4 - \sum_{k \neq l} \zeta_k^4 \geq \|\boldsymbol{\zeta}\|_4^4 - \sum_{k \neq l} \zeta_k^2 \max_{k \neq l} \zeta_k^2$$

$$\geq \|\boldsymbol{\zeta}\|_4^4 - \|\boldsymbol{\zeta}\|_2^2 \frac{4\beta^2}{\alpha^2}$$

$$\geq \|\boldsymbol{\zeta}\|_4^4 - 4\delta_1^2 \left(\frac{800}{7}\right)^3 \|\boldsymbol{\zeta}\|_4^4 \qquad \text{by (F.86).} \tag{F.103}$$

The last inequality also uses $\sum_{k\neq l} \zeta_k^2 \leq \|\boldsymbol{\zeta}\|_2^2 \leq 1$. On the other hand, the upper bound of $\zeta_l^2$ follows that

$$\zeta_l^2 \leq \left(\sqrt{\alpha} + \frac{2|\beta|}{\alpha}\right)^2 = \alpha + \frac{4\beta^2}{\alpha^2} + \frac{4|\beta|}{\sqrt{\alpha}}$$

$$\leq \|\boldsymbol{\zeta}\|_4^4 + 4\delta_1^2 \left(\frac{800}{7}\right)^3 \|\boldsymbol{\zeta}\|_4^4 + \eta \|\boldsymbol{\zeta}\|_4^4 \tag{F.104}$$

where we also use Lemma F.5, (F.86) and (F.90). Write

$$\xi \doteq 4\delta_1^2 \left(\frac{800}{7}\right)^3 \tag{F.105}$$

and also recall that

$$\eta \doteq \frac{4\delta_1}{C_\star^2} \left(\frac{800}{7}\right)^{1/2}.$$

Combine (F.103) and (F.104) to obtain

$$\zeta_l^2 = \frac{\zeta_l^4}{\zeta_l^2} \geq \frac{1-\xi}{1+\xi+\eta} = 1 - \frac{2\xi+\eta}{1+\xi+\eta} \tag{F.106}$$

which implies

$$1 - |\zeta_l| \leq \frac{2\xi+\eta}{1+\xi+\eta}.$$

Consequently, assuming $\zeta_l = \boldsymbol{a}_l^T \boldsymbol{q} > 0$ without loss of generality,

$$\|\boldsymbol{a}_l - \boldsymbol{q}\|_2^2 = \|\boldsymbol{a}_l\|_2^2 + \|\boldsymbol{q}\|_2^2 - 2\zeta_l = 2(1 - |\zeta_l|) \leq \frac{2(2\xi+\eta)}{1+\xi+\eta}. \tag{F.107}$$

**Second order optimality:** We show that

$$\boldsymbol{v}^T \operatorname{Hess} F(\boldsymbol{q}) \boldsymbol{v} \geq 0. \tag{F.108}$$

for any $\boldsymbol{v} \in \mathbb{S}^{p-1}$ that $P_{\boldsymbol{q}}^\perp \boldsymbol{v} \neq 0$. Pick any $\boldsymbol{v} \in \mathbb{S}^{p-1}$ such that $\boldsymbol{v} \perp \boldsymbol{q}$. We have

$$\boldsymbol{v}^T \operatorname{Hess} F(\boldsymbol{q}) \boldsymbol{v} = \boldsymbol{v}^T \operatorname{Hess} f(\boldsymbol{q}) \boldsymbol{v} + \boldsymbol{v}^T [\operatorname{Hess} F(\boldsymbol{q}) - \operatorname{Hess} f(\boldsymbol{q})] \boldsymbol{v} \tag{F.109}$$

$$\geq \boldsymbol{v}^T \operatorname{Hess} f(\boldsymbol{q}) \boldsymbol{v} - \|\operatorname{Hess} F(\boldsymbol{q}) - \operatorname{Hess} f(\boldsymbol{q})\|_{\mathrm{op}} \tag{F.110}$$

$$= \boldsymbol{v}^T [\operatorname{Hess}_{\ell_4} f(\boldsymbol{q}) + \operatorname{Hess}_{\ell_2} f(\boldsymbol{q})] \boldsymbol{v} - \|\operatorname{Hess} F(\boldsymbol{q}) - \operatorname{Hess} f(\boldsymbol{q})\|_{\mathrm{op}}. \tag{F.111}$$

We bound from below $\boldsymbol{v}^T \operatorname{Hess}_{\ell_4} f(\boldsymbol{q}) \boldsymbol{v}$ and $\boldsymbol{v}^T \operatorname{Hess}_{\ell_2} f(\boldsymbol{q}) \boldsymbol{v}$ respectively.

Recall from (F.10) that

$$\operatorname{Hess}_{\ell_4} f(\boldsymbol{q}) = -(1-\theta) P_{\boldsymbol{q}^\perp} \left[ 3 \sum_{j=1}^r \boldsymbol{a}_j \boldsymbol{a}_j^T (\boldsymbol{q}^T \boldsymbol{a}_j)^2 - \|\boldsymbol{q}^T \boldsymbol{A}\|_4^4 \boldsymbol{I} \right] P_{\boldsymbol{q}^\perp}.$$

Also recall that $\|\boldsymbol{\zeta}\|_\infty = |\zeta_l|$. We have

$$\boldsymbol{v}^T \operatorname{Hess}_{\ell_4} f(\boldsymbol{q}) \boldsymbol{v} = (1-\theta) \left[ -3\boldsymbol{v}^T \boldsymbol{A} \boldsymbol{D}_{\boldsymbol{\zeta}}^{\circ 2} \boldsymbol{A} \boldsymbol{v} + \|\boldsymbol{\zeta}\|_4^4 \right]$$

$$= (1-\theta) \left[ -3 \sum_{k=1}^r \left(\boldsymbol{A}^T \boldsymbol{v}\right)_k^2 \zeta_k^2 + \|\boldsymbol{\zeta}\|_4^4 \right]$$

$$= (1-\theta) \left\{ -3 \left[ \left( \boldsymbol{A}^T \boldsymbol{v} \right)_l^2 \zeta_l^2 + \sum_{k \neq l} \left( \boldsymbol{A}^T \boldsymbol{v} \right)_k^2 \zeta_k^2 \right] + \|\boldsymbol{\zeta}\|_4^4 \right\}. \tag{F.112}$$

Note from (F.107) that

$$\left( \boldsymbol{A}^T \boldsymbol{v} \right)_l^2 = |\langle \boldsymbol{a}_l, \boldsymbol{v} \rangle|^2 = |\langle \boldsymbol{a}_l - \boldsymbol{q}, \boldsymbol{v} \rangle|^2 \leq \|\boldsymbol{a}_l - \boldsymbol{q}\|_2^2 = 2(1 - |\zeta_l|) \leq 2 \left( 1 - \zeta_l^2 \right). \tag{F.113}$$

Also note that

$$\sum_{k \neq l} \left( \boldsymbol{A}^T \boldsymbol{v} \right)_k^2 \zeta_k^2 \leq \frac{4\beta^2}{\alpha^2} \|\boldsymbol{A}^T \boldsymbol{v}\|_2^2 \leq \frac{4\beta^2}{\alpha^2} \overset{(F.86)}{\leq} \xi \cdot \|\boldsymbol{\zeta}\|_4^4$$

with $\xi$ defined in (F.105). We thus have

$$\boldsymbol{v}^T \operatorname{Hess}_{\ell_4}(\boldsymbol{q}) \boldsymbol{v} \geq (1-\theta) \left[ -6 \left( 1 - \zeta_l^2 \right) \zeta_l^2 - 3\xi \|\boldsymbol{\zeta}\|_4^4 + \|\boldsymbol{\zeta}\|_4^4 \right]. \tag{F.114}$$

Since (F.104) ensures

$$\zeta_l^2 \leq (1 + \xi + \eta) \|\boldsymbol{\zeta}\|_4^4, \tag{F.115}$$

we further obtain

$$\begin{aligned} \boldsymbol{v}^T \operatorname{Hess}_{\ell_4}(\boldsymbol{q}) \boldsymbol{v} &\geq (1-\theta) \|\boldsymbol{\zeta}\|_4^4 \left[ -6(1 + \xi + \eta) \left( 1 - \zeta_l^2 \right) - 3\xi + 1 \right] \\ &\geq (1-\theta) \|\boldsymbol{\zeta}\|_4^4 \left[ -6(2\xi + \eta) - 3\xi + 1 \right] \\ &= (1-\theta) \left( 1 - 15\xi - 6\eta \right) \|\boldsymbol{\zeta}\|_4^4. \end{aligned} \tag{F.116}$$

The last step uses (F.106) again.

To bound from below $\boldsymbol{v}^T \operatorname{Hess}_{\ell_2} f(\boldsymbol{q}) \boldsymbol{v}$, by (F.10), we have

$$\boldsymbol{v}^T \operatorname{Hess}_{\ell_2}(\boldsymbol{q}) \boldsymbol{v} = \theta \left[ -2 \left( \boldsymbol{v}^T \boldsymbol{A} \boldsymbol{\zeta} \right)^2 - \|\boldsymbol{\zeta}\|_2^2 \|\boldsymbol{A} \boldsymbol{v}\|_2^2 + \|\boldsymbol{\zeta}\|_2^4 \right]. \tag{F.117}$$

To upper bound $\left( \boldsymbol{v}^T \boldsymbol{A} \boldsymbol{\zeta} \right)^2$, observe that

$$\begin{aligned} \left( \boldsymbol{v}^T \boldsymbol{A} \boldsymbol{\zeta} \right)^2 = \left[ \boldsymbol{a}_l^T \boldsymbol{v} \zeta_l + \sum_{k \neq l} \boldsymbol{a}_k^T \boldsymbol{v} \zeta_k \right]^2 &\leq 2 \left[ \left( \boldsymbol{v}^T \boldsymbol{a}_l \right)^2 \zeta_l^2 + \left( \sum_{k \neq l} \boldsymbol{a}_k^T \boldsymbol{v} \zeta_k \right)^2 \right] \\ &\leq 2 \left[ 2(1 - \zeta_l^2) \zeta_l^2 + \sum_{k \neq l} (\boldsymbol{a}_k^T \boldsymbol{v})^2 \sum_{k \neq l} \zeta_k^2 \right] \\ &\leq 2 \left[ 2(1 - \zeta_l^2) \zeta_l^2 + \|\boldsymbol{\zeta}\|_2^2 - \zeta_l^2 \right] \end{aligned} \tag{F.118}$$

where we used (F.113) and Cauchy-Schwarz inequality in the second line, and the fact that $\sum_{k \neq \ell} (\boldsymbol{A}^T \boldsymbol{v})_k^2 \leq \|\boldsymbol{A}^T \boldsymbol{v}\|_2^2 \leq 1$ in the third line. By observing that

$$\|\boldsymbol{\zeta}\|_2^2 = \frac{\|\boldsymbol{\zeta}\|_2^2}{\|\boldsymbol{\zeta}\|_4^4} \|\boldsymbol{\zeta}\|_4^4 \leq \frac{\|\boldsymbol{\zeta}\|_2^2}{\zeta_l^4} \|\boldsymbol{\zeta}\|_4^4 \leq \frac{\|\boldsymbol{\zeta}\|_4^4}{\zeta_l^4} \tag{F.119}$$

and

$$\zeta_l^2 \leq \frac{\|\boldsymbol{\zeta}\|_4^4}{\zeta_l^4} \zeta_l^2,$$

we have

$$\|\boldsymbol{\zeta}\|_2^2 - \zeta_l^2 \leq \frac{\|\boldsymbol{\zeta}\|_4^4}{\zeta_l^4} (1 - \zeta_l^2)$$

which further yields

$$\left( \boldsymbol{v}^T \boldsymbol{A} \boldsymbol{\zeta} \right)^2 \leq 2 \left[ 2(1 - \zeta_l^2) \zeta_l^2 + \frac{\|\boldsymbol{\zeta}\|_4^4}{\zeta_l^4} (1 - \zeta_l^2) \right] \tag{F.120}$$

$$\leq \frac{2(2\xi + \eta)}{1 + \xi + \eta}\left[2\zeta_l^2 + \frac{\|\zeta\|_4^4}{\zeta_l^4}\right] \qquad \text{by (F.106)} \tag{F.121}$$

$$\leq \frac{2(2\xi + \eta)}{1 + \xi + \eta}\left[2(1 + \xi + \eta) + \frac{1}{\left(1 - \frac{2\xi+\eta}{1+\xi+\eta}\right)^2}\right]\|\zeta\|_4^4 \quad \text{by (F.106) and (F.115)} \tag{F.122}$$

$$\leq 2(2\xi + \eta)\left[2 + \frac{1}{1 - 3\xi - \eta}\right]\|\zeta\|_4^4. \tag{F.123}$$

On the other hand, we have

$$\|\zeta\|_2^2\|Av\|_2^2 \leq \|\zeta\|_2^2 \leq \frac{1}{\left(1 - \frac{2\xi+\eta}{1+\xi+\eta}\right)^2}\|\zeta\|_4^4$$

by (F.119) and (F.106), and $\|\zeta\|_2^4 \geq \|\zeta\|_4^4$. It then follows that

$$v^T \operatorname{Hess}_{\ell_2}(q)v \geq \theta \|\zeta\|_4^4\left[-4(2\xi + \eta)\left[2 + \frac{1}{1 - 3\xi - \eta}\right] - \frac{1}{\left(1 - \frac{2\xi+\eta}{1+\xi+\eta}\right)^2} + 1\right]$$

$$\geq -16.5\theta\|\zeta\|_4^4(2\xi + \eta) \tag{F.124}$$

where the last line follows from $\delta_1 \leq 5 \times 10^{-5}$ and $C_\star \geq 1/5$ together with some simple algebra. Combine (F.116) and (F.124) to obtain

$$v^T \operatorname{Hess}_{\ell_4} f(q)v \geq \|\zeta\|_4^4\left[(1 - \theta)(1 - 15\xi - 6\eta) - 16.5\theta(2\xi + \eta)\right] \tag{F.125}$$

whence, on the event $\mathcal{E}$,

$$v^T \operatorname{Hess} F(q)v \geq \|\zeta\|_4^4\left[(1 - \theta)(1 - 15\xi - 6\eta) - 16.5\theta(2\xi + \eta)\right] - \delta_2 \tag{F.126}$$

$$\geq C_\star^2\left[(1 - \theta)(1 - 15\xi - 6\eta) - 16.5\theta(2\xi + \eta)\right] - \delta_2 \tag{F.127}$$

$$> 0 \tag{F.128}$$

by using $\delta_1 \leq 5 \times 10^{-5}$ and $\delta_2 < 10^{-3}$. The proof is complete. ∎

### F.5 Proof of Proposition 3.5

**Proof.** Write the eigenvalue decomposition of $YY^T = U\Lambda U^T$ with $U = [u_1, \ldots, u_r]$ and $\Lambda$ contains the first $r$ eigenvalues (in non-increasing order). By the definition of the Moore-Penrose inverse, we have

$$D = U\Lambda^{-1/2}U^T$$

such that

$$\bar{Y} = DY = U\Lambda^{-1/2}U^TAX.$$

Here $\Lambda^{-1/2}$ is the diagonal matrix with diagonal elements equal to the reciprocals of the square root of those of $\Lambda$. Further write the SVD of $A$ as $A = U_A D_A V_A^T$ with $U_A^T U_A = I_r$ and $D_A$ being diagonal and containing non-increasing singular values. Since $U_A = UQ$ for some orthogonal matrix $Q \in \mathbb{R}^{r \times r}$, we obtain

$$D = U_A Q^T \Lambda^{-1/2} Q U_A = U_A \frac{D_A^{-1}}{\sqrt{n\sigma^2\theta}}U_A^T + U_A\left(Q^T\Lambda^{-1/2}Q - \frac{D_A^{-1}}{\sqrt{n\sigma^2\theta}}\right)U_A^T. \tag{F.129}$$

It then follows that

$$\bar{Y} = U_A \frac{D_A^{-1}}{\sqrt{n\sigma^2\theta}}U_A^TAX + U_A\left(Q^T\Lambda^{-1/2}Q - \frac{D_A^{-1}}{\sqrt{n\sigma^2\theta}}\right)U_A^TAX \tag{F.130}$$

$$= U_A V_A^T \bar{X} + U_A \left( Q^T \Lambda^{-1/2} Q - \frac{D_A^{-1}}{\sqrt{n\sigma^2\theta}} \right) D_A V_A^T X \tag{F.131}$$

$$= U_A V_A^T \bar{X} + U_A V_A^T V_A \left( \sqrt{n\sigma^2\theta} Q^T \Lambda^{-1/2} Q D_A - I_r \right) V_A^T \bar{X} \tag{F.132}$$

$$= \bar{A}\bar{X} + \bar{A} V_A \left( \sqrt{n\sigma^2\theta} Q^T \Lambda^{-1/2} Q D_A - I_r \right) V_A^T \bar{X} \tag{F.133}$$

$$= \bar{A}\bar{X} + \bar{A}\Delta\bar{X} \tag{F.134}$$

where we used $\bar{X} = X/\sqrt{n\sigma^2\theta}$ and $\bar{A} = U_A V_A^T$ and write

$$\Delta = V_A \left[ \sqrt{\theta n\sigma^2} Q^T \Lambda^{-1/2} Q D_A - I_r \right] V_A^T$$

and it remains to bound from above $\|\Delta\|_{\mathrm{op}}$. Note that

$$U \Lambda U^T = Y Y^T = A \left( n\sigma^2\theta I_r + X X^T - n\sigma^2\theta I_r \right) A^T \tag{F.135}$$

$$= U_A D_A V_A^T \left( n\sigma^2\theta I_r + X X^T - n\sigma^2\theta I_r \right) V_A D_A U_A^T. \tag{F.136}$$

It then follows by using $U_A = U Q$ that

$$Q^T \Lambda Q - n\sigma^2\theta D_A^2 = D_A V_A^T \left( X X^T - n\sigma^2\theta I_r \right) V_A D_A,$$

hence

$$\frac{1}{\theta n\sigma^2} D_A^{-1} Q^T \Lambda Q D_A^{-1} - I_r = V_A^T \left( \frac{1}{\theta n\sigma^2} X X^T - I_r \right) V_A.$$

Let $\lambda_k$ denote the largest $k$th eigenvalue of the left hand side of the above equation, for $1 \leq k \leq r$. Then Weyl's inequality guarantees

$$\max_k |\lambda_k - 1| \leq \left\| \frac{1}{\theta n\sigma^2} X X^T - I_r \right\|_{\mathrm{op}}.$$

Clearly,

$$\|\Delta\|_{\mathrm{op}} = \left\| \sqrt{\theta n\sigma^2} Q^T \Lambda^{-1/2} Q D_A - I_r \right\|_{\mathrm{op}} \tag{F.137}$$

$$= \max_k \left| \frac{1}{\sqrt{\lambda_k}} - 1 \right| \tag{F.138}$$

$$= \max_k \frac{|1 - \lambda_k|}{\sqrt{\lambda_k}(1 + \sqrt{\lambda_k})} \tag{F.139}$$

$$\leq \max_k \frac{|1 - \lambda_k|}{\sqrt{\lambda_k}}. \tag{F.140}$$

It remains to bound from above the operator norm of $(\theta n\sigma^2)^{-1} X X^T - I_r$. It is easy to see that

$$\mathbb{E}\left[ \frac{1}{\theta n\sigma^2} X X^T \right] = I_r.$$

Since $X_{it}$ for $1 \leq i \leq r$ and $1 \leq t \leq n$ are i.i.d. sub-Gaussian random variables with sub-Gaussian constant no greater than 1, classical deviation inequality of the operator norm of the sample covariance matrices for i.i.d. sub-Gaussian entries [41, Remark 5.40] gives

$$\left\| \frac{1}{n\sigma^2} X X^T - \theta I_r \right\|_{\mathrm{op}} \leq c \left( \sqrt{\frac{r}{n}} + \frac{r}{n} \right) \tag{F.141}$$

with probability $1 - 2e^{-c'r}$ for some constants $c, c' > 0$. Using

$$\frac{1}{\theta}\sqrt{\frac{r}{n}} \leq c''$$

for some small constant $c'' > 0$ concludes

$$\left\| \sqrt{\theta n\sigma^2} Q^T \Lambda^{-1/2} Q D_A - I_r \right\|_{\mathrm{op}} \leq c''' \frac{1}{\theta}\sqrt{\frac{r}{n}} \tag{F.142}$$

with probability $1 - 2e^{-c'r}$. This completes the proof. $\blacksquare$

## F.6 Proof of Theorem 3.6

In this section we provide the proof of Theorem 3.6. Our proof is similar to Section F.4. Recall that $\bar{A} = U_A V_A^T$. We define a new partition of $\mathbb{S}^{p-1}$ as

$$R_0'' \doteq R_0''(c_\star) = \left\{ q \in \mathbb{S}^{p-1} : \left\| \bar{A}^T q \right\|_\infty^2 \leq c_\star \right\}, \tag{F.143}$$

$$R_1'' \doteq R_1(C_\star) = \left\{ q \in \mathbb{S}^{p-1} : \left\| \bar{A}^T q \right\|_\infty^2 \geq C_\star \right\},$$

$$R_2'' = \mathbb{S}^{p-1} \setminus \left( R_0'' \cup R_1'' \right).$$

Here $c_\star$ and $C_\star$ are positive constants satisfying $0 \leq c_\star \leq C_\star < 1$. Further define

$$\bar{f}_g(q) := \mathbb{E}\left[ -\frac{1}{12\theta\sigma^4 n} \left\| q^T \bar{A} X \right\|_4^4 \right] = -\frac{1}{4} \left[ (1-\theta) \left\| \bar{A}^T q \right\|_4^4 + \theta \left\| \bar{A}^T q \right\|_2^4 \right]. \tag{F.144}$$

The equality uses Lemma 2.5. Let $\delta_1$ and $\delta_2$ be some positive sequences and define the random event

$$\mathcal{E} = \left\{ \sup_{q \in \mathbb{S}^{p-1}} \left\| \operatorname{grad} \bar{f}_g(q) - \operatorname{grad} F_g(q) \right\|_2 \leq \delta_1, \quad \sup_{q \in \mathbb{S}^{p-1}} \left\| \operatorname{Hess} \bar{f}_g(q) - \operatorname{Hess} F_g(q) \right\|_{\mathrm{op}} \leq \delta_2 \right\}. \tag{F.145}$$

Here $\operatorname{grad} \bar{f}_g(q)$ and $\operatorname{grad} F_g(q)$ are the gradients of (F.144) and (2.8), respectively, at any point $q \in \mathbb{S}^{p-1}$. Similarly, $\operatorname{Hess} \bar{f}_g(q)$ and $\operatorname{Hess} F_g(q)$ are the corresponding Hessian matrices.

We observe that Lemmas F.6 and F.7 continue to hold by replacing $F(q)$, $f(q)$ and $A$ by $F_g(q)$, $\bar{f}_g(q)$ and $\bar{A}$, respectively, and by using $R_1''$ and $R_2''$ in lieu of $R_1'$ and $R_2'$. The proof is then completed by verifying that $\delta_2 \leq c_\star/25$, $\delta_2 \leq 10^{-3}$ and $\delta_1 \leq 5 \times 10^{-5}$. These are guaranteed by invoking Lemmas F.18 and F.19 and using condition (3.13).

## F.7 Proof of Lemma 4.1

**Proof.** It suffices to prove

$$\left\| \bar{A}^T q^{(0)} \right\|_\infty^2 \geq c_\star$$

for some $c_\star$ such that (3.13) holds. To this end, we work on the event where Proposition 3.5 holds such that

$$\bar{Y} = \bar{A}(I_r + \Delta)\bar{X}.$$

It then follows that

$$\left\| \bar{A}^T q^{(0)} \right\|_\infty^2 \geq \frac{1}{r} \left\| \bar{A}^T q^{(0)} \right\|_2^2 = \frac{1}{r} \left\| \frac{\bar{A}^T \bar{Y} \mathbf{1}_n}{\|\bar{Y} \mathbf{1}_n\|_2} \right\|_2^2 = \frac{1}{r} \left\| \frac{(I_r + \Delta)\bar{X} \mathbf{1}_n}{\|\bar{A}(I_r + \Delta)\bar{X} \mathbf{1}_n\|_2} \right\|_2^2 = \frac{1}{r} \tag{F.146}$$

by using $\bar{A}^T \bar{A} = I_r$, provided that $\|\bar{Y} \mathbf{1}_n\|_2 \neq 0$ which holds only one a set with zero measure. The proof is completed by invoking condition (4.2) to ensure (3.13) holds for $c_\star = 1/(2r)$. ∎

## F.8 Proof of Lemma 4.2

**Proof.** Recall that $F_g(q)$ and $\bar{f}_g(q)$ are defined in (2.8) and (F.144), respectively. We work on the event

$$\mathcal{E}_g := \left\{ \sup_{q \in \mathbb{S}^{p-1}} \left| F_g(q) - \bar{f}_g(q) \right| \lesssim \delta_n \right\}, \tag{F.147}$$

with

$$\delta_n = \left( \sqrt{r\theta} + \sqrt{\log n} \right) \sqrt{\frac{r}{\theta^2 \sqrt{\theta} n}} + \left( \theta r^2 + \frac{\log^2 n}{\theta} \right) \frac{r \log n}{n}.$$

According to Lemma F.16, $\mathcal{E}_g$ holds with probability at least $1 - cn^{-c'} - 2e^{-c''r}$. We aim to prove

$$\left\| \bar{A}^T q^{(k)} \right\|_\infty^2 \geq c_\star = \frac{1}{2r}, \qquad \forall k \geq 1.$$

Pick any $k \geq 1$. On the event $\mathcal{E}_g$, we have

$$F_g\left(q^{(k)}\right) \geq \bar{f}_g\left(q^{(k)}\right) - \delta_n.$$

For any $q \in \mathbb{S}^{p-1}$, we write $\bar{\zeta} = \bar{A}^T q$. Since

$$\begin{aligned}
\bar{f}_g(q) &= -\frac{1}{4}\left[ (1-\theta) \left\| \bar{\zeta} \right\|_4^4 + \theta \left\| \bar{\zeta} \right\|_2^4 \right] \\
&\geq -\frac{1}{4}\left[ (1-\theta) \left\| \bar{\zeta} \right\|_\infty^2 + \theta r \left\| \bar{\zeta} \right\|_\infty^2 \right] \\
&= -\frac{1}{4}(1-\theta+\theta r) \left\| \bar{\zeta} \right\|_\infty^4,
\end{aligned} \tag{F.148}$$

where we used $\left\| \bar{\zeta} \right\|_4^4 \leq \left\| \bar{\zeta} \right\|_\infty^2 \left\| \bar{\zeta} \right\|_2^2 \leq \left\| \bar{\zeta} \right\|_\infty^2$, $\left\| \bar{\zeta} \right\|_2^2 \leq r \left\| \bar{\zeta} \right\|_\infty^2$ and $\left\| \bar{\zeta} \right\|_2^2 \leq 1$. It then follows that

$$F_g\left(q^{(k)}\right) \geq -\frac{1}{4}(1-\theta+\theta r) \left\| \bar{A}^T q^{(k)} \right\|_\infty^2 - \delta_n \tag{F.149}$$

on the event $\mathcal{E}_g$. On the other hand, any gradient descent algorithm ensures

$$F_g(q^{(k)}) \leq F_g(q^{(0)}).$$

We thus have

$$\frac{1}{4}(1-\theta+\theta r) \left\| \bar{A}^T q^{(k)} \right\|_\infty^2 \geq -F_g\left(q^{(0)}\right) - \delta_n \geq -\bar{f}_g\left(q^{(0)}\right) - 2\delta_n$$

by using $\mathcal{E}_g$ again in the last inequality. To bound from below $-\bar{f}_g(q^{(0)})$, recalling the definition of $\bar{f}_g$ from (F.144), we have

$$-\bar{f}_g(q^{(0)}) = \frac{1}{4}\left[ (1-\theta) \left\| \bar{A}^T q^{(0)} \right\|_4^4 + \theta \left\| \bar{A}^T q^{(0)} \right\|_2^4 \right] \tag{F.150}$$

$$\geq \frac{1}{4} \frac{(1-\delta_1)^4}{(1+\delta_1)^4}\left[ (1-\theta) \left\| \frac{\bar{x}}{\|\bar{x}\|_2} \right\|_4^4 + \theta \right] \tag{F.151}$$

$$\geq \frac{1}{4} \frac{(1-\delta_1)^4}{(1+\delta_1)^4}\left[ \frac{1-\theta}{\|\bar{x}\|_0} + \theta \right] \tag{F.152}$$

Since, on the event where Proposition 3.5 holds such that $\bar{Y} = \bar{A}(I_r + \Delta)\bar{X}$,

$$\left\| \bar{A}^T q^{(0)} \right\|_4^4 = \left\| \frac{\bar{A}^T \bar{Y} \mathbf{1}_n}{\|\bar{Y}\mathbf{1}_n\|_2} \right\|_4^4 = \frac{\|(I_r+\Delta)\bar{X}\mathbf{1}_n\|_4^4}{\|(I_r+\Delta)\bar{X}\mathbf{1}_n\|_2^4} \tag{F.153}$$

and, similarly,

$$\left\| \bar{A}^T q^{(0)} \right\|_2^4 = 1,$$

we conclude

$$-\bar{f}_g(q^{(0)}) \geq \frac{1}{4}\left[ (1-\theta) \frac{\|(I_r+\Delta)\bar{X}\mathbf{1}_n\|_4^4}{\|(I_r+\Delta)\bar{X}\mathbf{1}_n\|_2^4} + \theta \right] \geq \frac{1}{4}\left[ \frac{1-\theta}{r} + \theta \right]$$

with probability $1 - 2e^{-cr}$. Here we used the basic inequality $\|v\|_2^4 \leq r\|v\|_4^4$ for any $v \in \mathbb{R}^r$. With the same probability, we further have

$$\left\| \bar{A}^T q^{(k)} \right\|_\infty^2 \geq \frac{1}{r} - \frac{8\delta_n}{1-\theta+\theta r} \geq \frac{1}{2r}$$

provided that

$$\frac{8\delta_n}{1-\theta+\theta r} \leq \frac{1}{2r}.$$

This is guaranteed by condition (4.3). The proof is complete. ∎

### F.9  Concentration inequalities when $A$ is semi-orthonormal

In this section, we provide deviation inequalities for different quantities between the population-level problem $f(\boldsymbol{q})$ in (2.5) and its sample counterpart $F(\boldsymbol{q})$ in (2.3), including the objective function, the Riemannian gradient and the Riemannian Hessian matrix. Our analysis adapts some technical results in [44] and [35] to our setting.

#### F.9.1  Deviation inequalities of the objective value

Recall that

$$F(\boldsymbol{q}) = -\frac{1}{12\theta\sigma^4 n}\left\|\boldsymbol{q}^T\boldsymbol{A}\boldsymbol{X}\right\|_4^4, \tag{F.154}$$

$$f(\boldsymbol{q}) = -\frac{1}{4}\left[(1-\theta)\left\|\boldsymbol{A}^T\boldsymbol{q}\right\|_4^4 + \theta\left\|\boldsymbol{A}^T\boldsymbol{q}\right\|_2^4\right].$$

Define

$$M_n = C(n+r)\left(\theta r^2 + \frac{\log^2 n}{\theta}\right) \tag{F.155}$$

for some constant $C > 0$.

**Lemma F.13** *Under Assumptions 2.1 and 2.3, with probability greater than $1 - cn^{-c'}$ for some constants $c, c' > 0$, one has*

$$\sup_{\boldsymbol{q}\in\mathbb{S}^{p-1}}\left|F\left(\boldsymbol{q}\right) - f\left(\boldsymbol{q}\right)\right| \lesssim \sqrt{\frac{r\log(M_n)}{\theta n}} + \frac{M_n}{n}\frac{r\log(M_n)}{n}.$$

**Proof.** Pick any $\boldsymbol{q}\in\mathbb{S}^{p-1}$. Note that the result holds trivially if $\boldsymbol{A}^T\boldsymbol{q} = 0$. For $\boldsymbol{A}^T\boldsymbol{q}\neq 0$, we define

$$\widetilde{\boldsymbol{q}} = \frac{\boldsymbol{A}^T\boldsymbol{q}}{\|\boldsymbol{A}^T\boldsymbol{q}\|_2}, \qquad \text{with}\quad \widetilde{\boldsymbol{q}}\in\mathbb{S}^{r-1}.$$

Note that

$$\frac{F(\boldsymbol{q})}{\|\boldsymbol{A}^T\boldsymbol{q}\|_2^4} = \frac{1}{n}\sum_{k=1}^n F_{\widetilde{\boldsymbol{q}}}\left(\boldsymbol{x}_k\right), \qquad \text{with}\qquad F_{\widetilde{\boldsymbol{q}}}\left(\boldsymbol{x}_k\right) = -\frac{1}{12\theta\sigma^4}\left(\widetilde{\boldsymbol{q}}^T\boldsymbol{x}_k\right)^4. \tag{F.156}$$

The proof of Lemma 2.5 shows that

$$\mathbb{E}[F_{\widetilde{\boldsymbol{q}}}\left(\boldsymbol{x}_k\right)] = \frac{f(\boldsymbol{q})}{\|\boldsymbol{A}^T\boldsymbol{q}\|_2^4} = -\frac{1}{4}\left[(1-\theta)\left\|\widetilde{\boldsymbol{q}}\right\|_4^4 + \theta\left\|\widetilde{\boldsymbol{q}}\right\|_2^4\right] \doteq g(\widetilde{\boldsymbol{q}}), \quad \forall 1\leq k\leq n. \tag{F.157}$$

We thus aim to invoke Lemma G.6 with $n_1 = r$, $d_1 = 1$, $n = r$ and $p = n$ to bound from above

$$\sup_{\widetilde{\boldsymbol{q}}\in\mathbb{S}^{r-1}}\left|\frac{1}{n}\sum_{k=1}^n F_{\widetilde{\boldsymbol{q}}}\left(\boldsymbol{x}_k\right) - g\left(\widetilde{\boldsymbol{q}}\right)\right|.$$

Consequently, the result follows by noting that

$$\sup_{\boldsymbol{q}\in\mathbb{S}^{p-1}}\left|F\left(\boldsymbol{q}\right) - f\left(\boldsymbol{q}\right)\right| = \sup_{\boldsymbol{q}\in\mathbb{S}^{p-1}\setminus R_0}\left\|\boldsymbol{A}^T\boldsymbol{q}\right\|_2^4\cdot\left|\frac{1}{n}\sum_{k=1}^n F_{\widetilde{\boldsymbol{q}}}\left(\boldsymbol{x}_k\right) - g\left(\widetilde{\boldsymbol{q}}\right)\right|$$

and using $\|\boldsymbol{A}^T\boldsymbol{q}\|_2\leq 1$ which holds uniformly over $\boldsymbol{q}\in\mathbb{S}^{p-1}$.

Since the entries of $\boldsymbol{x}_k$ are i.i.d. Bernoulli-Gaussian random variables with parameter $(\theta, \sigma^2)$, each $\boldsymbol{x}_{ki}$, for $1\leq i\leq r$, is sub-Gaussian with the sub-Gaussian parameter equal to $\sigma^2$. It thus suffices to verify Conditions 1–2 in Lemma G.6. For simplicity, we write $\boldsymbol{x} = \boldsymbol{x}_k$.

**Verification of Condition 1:** Since $\mathbb{E}[F_{\widetilde{\boldsymbol{q}}}(\boldsymbol{x})] = g(\widetilde{\boldsymbol{q}})$, we observe

$$\left|\mathbb{E}[F_{\widetilde{\boldsymbol{q}}}(\boldsymbol{x})]\right| = \frac{1}{4}\left[(1-\theta)\left\|\widetilde{\boldsymbol{q}}\right\|_4^4 + \theta\left\|\widetilde{\boldsymbol{q}}\right\|_2^4\right] \leq \frac{1}{4}\left\|\widetilde{\boldsymbol{q}}\right\|_2^4 = \frac{1}{4} \tag{F.158}$$

where we used $\|\widetilde{q}\|_4^4 \leq \|\widetilde{q}\|_2^4$. Thus $B_f = 1/4$. For any $q_1 \neq q_2 \in \mathbb{S}^{r-1}$, we have

$$|\mathbb{E}[F_{q_1}(x)] - \mathbb{E}[F_{q_2}(x)]| \tag{F.159}$$

$$\leq \frac{1-\theta}{4}\left|\|q_1\|_4^4 - \|q_2\|_4^4\right| + \frac{\theta}{4}\left|\|q_1\|_2^4 - \|q_2\|_2^4\right| \tag{F.160}$$

$$= \frac{1-\theta}{4}\left|\|q_1\|_4 - \|q_2\|_4\right|(\|q_1\|_4 + \|q_2\|_4)\left(\|q_1\|_4^2 + \|q_2\|_4^2\right) \tag{F.161}$$

$$+ \frac{\theta}{4}\left|\|q_1\|_2 - \|q_2\|_2\right|(\|q_1\|_2 + \|q_2\|_2)\left(\|q_1\|_2^2 + \|q_2\|_2^2\right) \tag{F.162}$$

$$\leq (1-\theta)\|q_1 - q_2\|_4 + \theta\|q_1 - q_2\|_2 \qquad\qquad \text{by } \|q_1\|_4 \leq \|q_1\|_2 = 1 \tag{F.163}$$

$$\leq \|q_1 - q_2\|_2. \tag{F.164}$$

This gives $L_f = 1$.

**Verification of Condition 2:** We define

$$x = \bar{x} + \widetilde{x}$$

as (G.27) with $B = 2\sigma\sqrt{\log(nr)}$. For the similar fashion, we define $x_k = \bar{x}_k + \widetilde{x}_k$ for $1 \leq k \leq n$. We verify Condition 2 on the event

$$\mathcal{E}' := \bigcap_{k=1}^{n}\left\{\|x_k\|_2 \lesssim \sigma(\sqrt{r\theta} + \sqrt{\log n})\right\}. \tag{F.165}$$

Lemma G.3 ensures that $\mathbb{P}(\mathcal{E}') \geq 1 - 2n^{-c}$ for some $c > 0$. Note that on the event $\mathcal{E}'$,

$$\|\bar{x}\|_2 \leq \sigma(\sqrt{r\theta} + \sqrt{\log n}). \tag{F.166}$$

Pick any $q \in \mathbb{S}^{r-1}$, we have

$$\|F_q(\bar{x})\|_2 = \left\|\frac{1}{12\theta\sigma^4}\left(q^T\bar{x}\right)^4\right\|_2 \leq \frac{\|q\|_2^4\|\bar{x}\|_2^4}{12\theta\sigma^4} \leq C\left(\theta r^2 + \frac{\log^2 n}{\theta}\right) \tag{F.167}$$

for some constant $C > 0$. Thus,

$$R_1 = C\left(\theta r^2 + \frac{\log^2 n}{\theta}\right).$$

On the other hand, we have

$$\sup_{q \in \mathbb{S}^{r-1}}\mathbb{E}\left[\|F_q(\bar{x})\|_2^2\right] \leq \sup_{q \in \mathbb{S}^{r-1}}\mathbb{E}\left[\|F_q(x)\|_2^2\right] \leq c\theta^{-1} = R_2 \tag{F.168}$$

for some constant $c > 0$. Here Lemma G.4 is used in the last inequality.

On the other hand, pick $q_1 \neq q_2 \in \mathbb{S}^{r-1}$. We obtain

$$\|F_{q_1}(\bar{x}) - F_{q_2}(\bar{x})\|_2 \tag{F.169}$$

$$= \frac{1}{12\theta\sigma^4}\left\|\left(q_1^T\bar{x}\right)^4 - \left(q_2^T\bar{x}\right)^4\right\|_2 \tag{F.170}$$

$$= \frac{1}{12\theta\sigma^4}\left|\left(q_1^T\bar{x}\right) - \left(q_2^T\bar{x}\right)\right| \cdot \left|\left(q_1^T\bar{x}\right) + \left(q_2^T\bar{x}\right)\right|\left(\left(q_1^T\bar{x}\right)^2 + \left(q_2^T\bar{x}\right)^2\right) \tag{F.171}$$

$$\leq \frac{4}{12\theta\sigma^4}\|\bar{x}\|_2^4\|q_1 - q_2\|_2. \tag{F.172}$$

Combine with (F.166) to conclude

$$\|F_{q_1}(\bar{x}) - F_{q_2}(\bar{x})\|_2 \leq R_1\|q_1 - q_2\|_2, \tag{F.173}$$

hence $\bar{L}_f = R_1$.

Finally, invoke Lemma G.6 with $M = c'R_1n = M_n$ to obtain the desired result and complete the proof. ∎

### F.9.2 Deviation inequalities of the Riemannian gradient

In this part, we derive the deviation inequalities between the Riemannian gradient of $F(\boldsymbol{q})$ and that of function $f(\boldsymbol{q})$. From (F.8), note that, for any $\boldsymbol{q} \in \mathbb{S}^{p-1}$,

$$\operatorname{grad} F(\boldsymbol{q}) \doteq \operatorname{grad}_{\|\boldsymbol{q}\|_2 = 1} F(\boldsymbol{q}) = -\frac{1}{3\theta\sigma^4 n} \boldsymbol{P}_{\boldsymbol{q}\perp} \sum_{k=1}^{n} \left(\boldsymbol{q}^T \boldsymbol{A} \boldsymbol{x}_k\right)^3 \boldsymbol{A} \boldsymbol{x}_k, \tag{F.174}$$

$$\operatorname{grad} f(\boldsymbol{q}) \doteq \operatorname{grad}_{\|\boldsymbol{q}\|_2 = 1} f(\boldsymbol{q}) = -\boldsymbol{P}_{\boldsymbol{q}\perp} \left[ (1-\theta) \sum_{j=1}^{r} a_j (\boldsymbol{q}^T \boldsymbol{a}_j)^3 + \theta \|\boldsymbol{q}^T \boldsymbol{A}\|_2^2 \boldsymbol{A}\boldsymbol{A}^T \boldsymbol{q} \right]. \tag{F.175}$$

Direct calculation shows that

$$\mathbb{E}\left[\operatorname{grad} F(\boldsymbol{q})\right] = \operatorname{grad} f(\boldsymbol{q}). \tag{F.176}$$

The following lemma provides deviation inequalities between $F(\boldsymbol{q})$ and $f(\boldsymbol{q})$ by invoking Lemma G.6, stated in Appendix G. Recall that $M_n$ is defined in (F.155).

**Lemma F.14** *Under Assumptions 2.1 and 2.3, with probability greater than $1 - cn^{-c'}$ for some constants $c, c' > 0$, one has*

$$\sup_{\boldsymbol{q} \in \mathbb{S}^{p-1}} \|\operatorname{grad} F(\boldsymbol{q}) - \operatorname{grad} f(\boldsymbol{q})\|_2 \lesssim \sqrt{\frac{r^2 \log(M_n)}{\theta n}} + \frac{M_n}{n} \frac{r \log(M_n)}{n}.$$

**Proof.** Pick any $\boldsymbol{q} \in \mathbb{S}^{p-1}$. As the result trivially holds for $\boldsymbol{A}^T \boldsymbol{q} = 0$, we only focus on when $\boldsymbol{A}^T \boldsymbol{q} \neq 0$. Define

$$\widetilde{\boldsymbol{q}} = \frac{\boldsymbol{A}^T \boldsymbol{q}}{\|\boldsymbol{A}^T \boldsymbol{q}\|_2}, \qquad \text{with} \quad \widetilde{\boldsymbol{q}} \in \mathbb{S}^{r-1}.$$

By writing $\boldsymbol{\zeta} = \boldsymbol{A}^T \boldsymbol{q}$, observe that

$$\|\operatorname{grad} F(\boldsymbol{q}) - \operatorname{grad} f(\boldsymbol{q})\|_2 \tag{F.177}$$

$$= \left\| \boldsymbol{P}_{\boldsymbol{q}\perp} \left[ \frac{1}{3\theta\sigma^4 n} \sum_{k=1}^{n} \left(\boldsymbol{q}^T \boldsymbol{A} \boldsymbol{x}_k\right)^3 \boldsymbol{A} \boldsymbol{x}_k - \left[(1-\theta)\boldsymbol{A}(\boldsymbol{\zeta})^{\circ 3} + \theta\|\boldsymbol{\zeta}\|_2^2 \boldsymbol{A}\boldsymbol{\zeta}\right] \right] \right\|_2 \tag{F.178}$$

$$\leq \left\| \frac{1}{3\theta\sigma^4 n} \sum_{k=1}^{n} \left(\boldsymbol{q}^T \boldsymbol{A} \boldsymbol{x}_k\right)^3 \boldsymbol{A} \boldsymbol{x}_k - \left[(1-\theta)\boldsymbol{A}(\boldsymbol{\zeta})^{\circ 3} + \theta\|\boldsymbol{\zeta}\|_2^2 \boldsymbol{A}\boldsymbol{\zeta}\right] \right\|_2 \tag{F.179}$$

$$\leq \left\| \frac{1}{3\theta\sigma^4 n} \sum_{k=1}^{n} \left(\boldsymbol{q}^T \boldsymbol{A} \boldsymbol{x}_k\right)^3 \boldsymbol{x}_k - \left[(1-\theta)(\boldsymbol{\zeta})^{\circ 3} + \theta\|\boldsymbol{\zeta}\|_2^2 \boldsymbol{A}\boldsymbol{\zeta}\right] \right\|_2 \tag{F.180}$$

$$\leq \|\boldsymbol{A}^T \boldsymbol{q}\|_2^3 \left\| \frac{1}{3\theta\sigma^4 n} \sum_{k=1}^{n} \left(\widetilde{\boldsymbol{q}}^T \boldsymbol{x}_k\right)^3 \boldsymbol{x}_k - \left[(1-\theta)(\widetilde{\boldsymbol{q}})^{\circ 3} + \theta\|\widetilde{\boldsymbol{q}}\|_2^2 \widetilde{\boldsymbol{q}}\right] \right\|_2 \tag{F.181}$$

$$\leq \left\| \frac{1}{3\theta\sigma^4 n} \sum_{k=1}^{n} \left(\widetilde{\boldsymbol{q}}^T \boldsymbol{x}_k\right)^3 \boldsymbol{x}_k - \left[(1-\theta)(\widetilde{\boldsymbol{q}})^{\circ 3} + \theta\|\widetilde{\boldsymbol{q}}\|_2^2 \widetilde{\boldsymbol{q}}\right] \right\|_2, \tag{F.182}$$

where we have used $\|\boldsymbol{A}\|_{\mathrm{op}} \leq 1$ and $\|\boldsymbol{A}^T \boldsymbol{q}\|_2^3 \leq 1$ in the last two steps. Define

$$F_{\widetilde{\boldsymbol{q}}}(\boldsymbol{x}) := \frac{1}{3\theta\sigma^4} \left(\widetilde{\boldsymbol{q}}^T \boldsymbol{x}\right)^3 \boldsymbol{x}. \tag{F.183}$$

It is easy to verify that

$$\mathbb{E}\left[F_{\widetilde{\boldsymbol{q}}}(\boldsymbol{x})\right] = \left[(1-\theta)(\widetilde{\boldsymbol{q}})^{\circ 3} + \theta\|\widetilde{\boldsymbol{q}}\|_2^2 \widetilde{\boldsymbol{q}}\right] \doteq g(\widetilde{\boldsymbol{q}}). \tag{F.184}$$

We thus aim to invoke Lemma G.6 with $n_1 = r$, $d_1 = r$, $n = r$ and $p = n$, to bound from above

$$\sup_{\tilde{\boldsymbol{q}} \in \mathbb{S}^{r-1}} \left\| \frac{1}{n} \sum_{k=1}^{n} F_{\tilde{\boldsymbol{q}}}(\boldsymbol{x}_k) - g(\tilde{\boldsymbol{q}}) \right\|_2 .$$

Recall that $x_{ij}$ is sub-Gaussian with parameter $\sigma^2$, for $1 \le i \le n$ and $1 \le j \le r$.

**Verification of Condition 1:** By $\|\tilde{\boldsymbol{q}}\|_2 = 1$, notice that

$$\|g(\tilde{\boldsymbol{q}})\|_2 = \left\| (1-\theta)(\tilde{\boldsymbol{q}})^{\circ 3} + \theta \tilde{\boldsymbol{q}} \right\|_2 \le (1-\theta) \left\| \tilde{\boldsymbol{q}}^{\circ 3} \right\|_2 + \theta \|\tilde{\boldsymbol{q}}\|_2 \le \|\tilde{\boldsymbol{q}}\|_2 = 1. \qquad \text{(F.185)}$$

Further note that, for any $\tilde{\boldsymbol{q}}_1, \tilde{\boldsymbol{q}}_2 \in \mathbb{S}^{r-1}$,

$$\|g(\tilde{\boldsymbol{q}}_1) - g(\tilde{\boldsymbol{q}}_2)\|_2 \le (1-\theta) \left\| (\tilde{\boldsymbol{q}}_1)^{\circ 3} - (\tilde{\boldsymbol{q}}_2)^{\circ 3} \right\|_2 + \theta \|\tilde{\boldsymbol{q}}_1 - \tilde{\boldsymbol{q}}_2\|_2 \qquad \text{(F.186)}$$

$$\le 3(1-\theta) \|\tilde{\boldsymbol{q}}_1 - \tilde{\boldsymbol{q}}_2\|_2 + \theta \|\tilde{\boldsymbol{q}}_1 - \tilde{\boldsymbol{q}}_2\|_2 \qquad \text{(F.187)}$$

$$\le 3 \|\tilde{\boldsymbol{q}}_1 - \tilde{\boldsymbol{q}}_2\|_2 . \qquad \text{(F.188)}$$

Here $\|(\tilde{\boldsymbol{q}}_1)^{\circ 3} - (\tilde{\boldsymbol{q}}_2)^{\circ 3}\|_2 \le 3 \|\tilde{\boldsymbol{q}}_1 - \tilde{\boldsymbol{q}}_2\|_2$ is used in the second step. As a result, $B_f = 1$ and $L_f = 6$.

**Verification of Condition 2:** We still work on the event $\mathcal{E}'$ in (F.165) such that (F.166) holds for all $1 \le k \le n$. In this case,

$$\left\| F_{\tilde{\boldsymbol{q}}}(\bar{\boldsymbol{x}}_i) \right\|_2 = \left\| \frac{1}{3\theta\sigma^4} \left( \tilde{\boldsymbol{q}}^T \bar{\boldsymbol{x}}_i \right)^3 \bar{\boldsymbol{x}}_i \right\|_2 \le \frac{\|\bar{\boldsymbol{x}}_i\|_2^4}{3\theta\sigma^4} \le C \left( \theta r^2 + \frac{\log^2 n}{\theta} \right). \qquad \text{(F.189)}$$

Hence

$$R_1 = C \left( \theta r^2 + \frac{\log^2 n}{\theta} \right). \qquad \text{(F.190)}$$

Also from Lemma G.8 with some straightforward modifications, we know

$$\sup_{\tilde{\boldsymbol{q}} \in \mathbb{S}^{r-1}} \mathbb{E} \left[ \left\| F_{\tilde{\boldsymbol{q}}}(\bar{\boldsymbol{x}}_i) \right\|_2^2 \right] \le \sup_{\tilde{\boldsymbol{q}} \in \mathbb{S}^{r-1}} \mathbb{E} \left[ \left\| F_{\tilde{\boldsymbol{q}}}(\boldsymbol{x}_i) \right\|_2^2 \right] \le c\theta^{-1} r, \qquad \text{(F.191)}$$

for some constant $c > 0$. We thus have $R_2 = c\theta^{-1} r$.

To calculate $\bar{L}_f$, we have

$$\left\| F_{\tilde{\boldsymbol{q}}_1}(\bar{\boldsymbol{x}}_i) - F_{\tilde{\boldsymbol{q}}_2}(\bar{\boldsymbol{x}}_i) \right\|_2 = \frac{1}{3\theta\sigma^4} \left\| \left( \tilde{\boldsymbol{q}}_1^T \bar{\boldsymbol{x}}_i \right)^3 \bar{\boldsymbol{x}}_i - \left( \tilde{\boldsymbol{q}}_2^T \bar{\boldsymbol{x}}_i \right)^3 \bar{\boldsymbol{x}}_i \right\|_2 \qquad \text{(F.192)}$$

$$\le \frac{1}{3\theta\sigma^4} \left\| (\tilde{\boldsymbol{q}}_1)^{\circ 3} - (\tilde{\boldsymbol{q}}_2)^{\circ 3} \right\|_2 \|\bar{\boldsymbol{x}}\|_2^4 \qquad \text{(F.193)}$$

$$\le \frac{1}{\theta\sigma^4} \|\tilde{\boldsymbol{q}}_1 - \tilde{\boldsymbol{q}}_2\|_2 \|\bar{\boldsymbol{x}}\|_2^4 . \qquad \text{(F.194)}$$

Here $\|\tilde{\boldsymbol{q}}_1^{\circ 3} - \tilde{\boldsymbol{q}}_2^{\circ 3}\|_2 \le 3 \|\tilde{\boldsymbol{q}}_1 - \tilde{\boldsymbol{q}}_2\|_2$ is used in the last step. We thus conclude

$$\left\| F_{\boldsymbol{q}_1}(\bar{\boldsymbol{x}}) - F_{\boldsymbol{q}_2}(\bar{\boldsymbol{x}}) \right\|_2 \le R_1 \|\boldsymbol{q}_1 - \boldsymbol{q}_2\|_2 \qquad \text{(F.195)}$$

hence $\bar{L}_f = R_1$.

Finally invoke Lemma G.6 with $M = C'(n+r)R_1$ to complete the proof. ∎

### F.9.3 Deviation inequalities of the Riemannian Hessian

In this part we will show that the Hessian of $F(\boldsymbol{q})$ concentrates around that of $f(\boldsymbol{q})$. Notice that, for any $\boldsymbol{q} \in \mathbb{S}^{p-1}$ with $\boldsymbol{\zeta} = \boldsymbol{A}^T \boldsymbol{q}$,

$$\text{Hess}\, F(\boldsymbol{q}) = -\frac{1}{3\theta\sigma^4 n} \sum_{k=1}^{n} P_{\boldsymbol{q}\perp} \left[ 3 \left( \boldsymbol{\zeta}^T \boldsymbol{x}_k \right)^2 \boldsymbol{A} \boldsymbol{x}_k \left( \boldsymbol{A} \boldsymbol{x}_k \right)^T - \left( \boldsymbol{\zeta}^T \boldsymbol{x}_k \right)^4 \boldsymbol{I}_p \right] P_{\boldsymbol{q}\perp}, \qquad \text{(F.196)}$$

$$\text{Hess}\, f\,(\boldsymbol{q}) = -\left\{(1-\theta)\,P_{\boldsymbol{q}\perp}\left[3\boldsymbol{A}\,\text{diag}(\boldsymbol{\zeta}^{\circ 2})\boldsymbol{A}^T - \|\boldsymbol{\zeta}\|_4^4\,\boldsymbol{I}\right]P_{\boldsymbol{q}\perp}\right.$$
$$\left.+\theta P_{\boldsymbol{q}\perp}\left[\|\boldsymbol{\zeta}\|_2^2\,\boldsymbol{A}\boldsymbol{A}^T + 2\boldsymbol{A}\boldsymbol{\zeta}\boldsymbol{\zeta}^T\boldsymbol{A}^T - \|\boldsymbol{\zeta}\|_2^4\,\boldsymbol{I}\right]P_{\boldsymbol{q}\perp}\right\}. \tag{F.197}$$

Straightforward calculation shows that

$$\mathbb{E}[\text{Hess}\, F(\boldsymbol{q})] = \text{Hess}\, f(\boldsymbol{q}).$$

The following lemma provides the deviation inequalities between $\text{Hess}\, F\,(\boldsymbol{q})$ and $\text{Hess}\, f(\boldsymbol{q})$ via an application of Lemma G.5, stated in Appendix G. Recall that $M_n$ is defined in (F.155).

**Lemma F.15** *Under Assumptions 2.1 and 2.3, with probability greater than $1 - cn^{-c'}$ for some constants $c, c' > 0$, one has*

$$\sup_{\boldsymbol{q}\in\mathbb{S}^{p-1}} \|\text{Hess}\, F\,(\boldsymbol{q}) - \text{Hess}\, f\,(\boldsymbol{q})\|_{\text{op}} \lesssim \sqrt{\frac{r^3\log(M_n)}{\theta n}} + \frac{M_n}{n}\frac{r\log(M_n)}{n}.$$

**Proof.** Pick any $\boldsymbol{q}\in\mathbb{S}^{p-1}$ and consider $\boldsymbol{A}^T\boldsymbol{q}\neq 0$. Recall that

$$\tilde{\boldsymbol{q}} = \frac{\boldsymbol{A}^T\boldsymbol{q}}{\|\boldsymbol{A}^T\boldsymbol{q}\|_2}, \qquad \text{with}\quad \tilde{\boldsymbol{q}}\in\mathbb{S}^{r-1}.$$

Observe that

$$\|\text{Hess}\, F\,(\boldsymbol{q}) - \text{Hess}\, f\,(\boldsymbol{q})\|_{\text{op}} \tag{F.198}$$

$$= \left\|\frac{1}{3\theta\sigma^4 n}\sum_{k=1}^{n} P_{\boldsymbol{q}\perp}\left[3\left(\boldsymbol{\zeta}^T\boldsymbol{x}_k\right)^2 \boldsymbol{A}\boldsymbol{x}_k\left(\boldsymbol{A}\boldsymbol{x}_k\right)^T - \left(\boldsymbol{\zeta}^T\boldsymbol{x}_k\right)^4 \boldsymbol{I}_p\right]P_{\boldsymbol{q}\perp}\right. \tag{F.199}$$

$$\left. - (1-\theta)\,P_{\boldsymbol{q}\perp}\left[3\boldsymbol{A}\,\text{diag}(\boldsymbol{\zeta}^{\circ 2})\boldsymbol{A}^T - \|\boldsymbol{\zeta}\|_4^4\,\boldsymbol{I}\right]P_{\boldsymbol{q}\perp} - \theta P_{\boldsymbol{q}\perp}\left[\|\boldsymbol{\zeta}\|_2^2\,\boldsymbol{A}\boldsymbol{A}^T + 2\boldsymbol{A}\boldsymbol{\zeta}\boldsymbol{\zeta}^T\boldsymbol{A}^T - \|\boldsymbol{\zeta}\|_2^4\,\boldsymbol{I}_p\right]P_{\boldsymbol{q}\perp}\right\|_{\text{op}} \tag{F.200}$$

$$\leq \left\|\frac{1}{3\theta\sigma^4 n}\sum_{k=1}^{n}\left[3\left(\boldsymbol{\zeta}^T\boldsymbol{x}_k\right)^2 \boldsymbol{A}\boldsymbol{x}_k\left(\boldsymbol{A}\boldsymbol{x}_k\right)^T - \left(\boldsymbol{\zeta}^T\boldsymbol{x}_k\right)^4 \boldsymbol{I}_p\right]\right. \tag{F.201}$$

$$\left. - (1-\theta)\left[3\boldsymbol{A}\,\text{diag}(\boldsymbol{\zeta}^{\circ 2})\boldsymbol{A}^T - \|\boldsymbol{\zeta}\|_4^4\,\boldsymbol{I}_p\right] - \theta\left[\|\boldsymbol{\zeta}\|_2^2\,\boldsymbol{A}\boldsymbol{A}^T + 2\boldsymbol{A}\boldsymbol{\zeta}\boldsymbol{\zeta}^T\boldsymbol{A}^T - \|\boldsymbol{\zeta}\|_2^4\,\boldsymbol{I}_p\right]\right\|_{\text{op}}$$

$$\leq \left\|\frac{1}{3\theta\sigma^4 n}\sum_{k=1}^{n}\left[3\left(\boldsymbol{\zeta}^T\boldsymbol{x}_k\right)^2 \boldsymbol{A}\boldsymbol{x}_k\boldsymbol{x}_k^T\boldsymbol{A}^T\right] - \left[3\,(1-\theta)\,\boldsymbol{A}\,\text{diag}(\boldsymbol{\zeta}^{\circ 2})\boldsymbol{A}^T + \theta\left(\|\boldsymbol{\zeta}\|_2^2\,\boldsymbol{A}\boldsymbol{A}^T + 2\boldsymbol{A}\boldsymbol{\zeta}\boldsymbol{\zeta}^T\boldsymbol{A}^T\right)\right]\right\|_{\text{op}} \tag{F.202}$$

$$+ \left\|\frac{1}{3\theta\sigma^4 n}\sum_{k=1}^{n}\left(\boldsymbol{\zeta}^T\boldsymbol{x}_k\right)^4 \boldsymbol{I}_p - \left[\theta\|\boldsymbol{\zeta}\|_2^4 + (1-\theta)\|\boldsymbol{\zeta}\|_4^4\right]\boldsymbol{I}_p\right\|_{\text{op}} \tag{F.203}$$

$$\leq \left\|\frac{1}{3\theta\sigma^4 n}\sum_{k=1}^{n}\left[3\left(\boldsymbol{\zeta}^T\boldsymbol{x}_k\right)^2 \boldsymbol{x}_k\boldsymbol{x}_k^T\right] - \left[3\,(1-\theta)\,\text{diag}(\boldsymbol{\zeta}^{\circ 2}) + \theta\left(\|\boldsymbol{\zeta}\|_2^2\,\boldsymbol{I}_p + 2\boldsymbol{\zeta}\boldsymbol{\zeta}^T\right)\right]\right\|_{\text{op}} \tag{F.204}$$

$$+ \left|\frac{1}{3\theta\sigma^4 n}\sum_{k=1}^{n}\left(\boldsymbol{\zeta}^T\boldsymbol{x}_k\right)^4 - \theta\|\boldsymbol{\zeta}\|_2^4 - (1-\theta)\|\boldsymbol{\zeta}\|_4^4\right| \tag{F.205}$$

$$= \left\|\frac{1}{\theta\sigma^4 n}\|\boldsymbol{A}^T\boldsymbol{q}\|_2^2\sum_{k=1}^{n}\left[\left(\tilde{\boldsymbol{q}}^T\boldsymbol{x}_k\right)^2 \boldsymbol{x}_k\boldsymbol{x}_k^T\right] - \|\boldsymbol{A}^T\boldsymbol{q}\|_2^2\left[3\,(1-\theta)\,\text{diag}(\tilde{\boldsymbol{q}}^{\circ 2}) + \theta\left(\|\tilde{\boldsymbol{q}}\|_2^2\,\boldsymbol{I}_p + 2\tilde{\boldsymbol{q}}\tilde{\boldsymbol{q}}^T\right)\right]\right\|_{\text{op}} \tag{F.206}$$

$$+ \left|\|\boldsymbol{A}^T\boldsymbol{q}\|_2^4\frac{1}{3\theta\sigma^4 n}\sum_{k=1}^{n}\left(\tilde{\boldsymbol{q}}^T\boldsymbol{x}_k\right)^4 - \theta\|\boldsymbol{A}^T\boldsymbol{q}\|_2^4\|\tilde{\boldsymbol{q}}\|_2^4 - (1-\theta)\|\boldsymbol{A}^T\boldsymbol{q}\|_2^4\|\tilde{\boldsymbol{q}}\|_4^4\right|. \tag{F.207}$$

Define

$$F_{\tilde{q}}^{L_2}(x) \doteq \frac{1}{\theta\sigma^3}\left(\tilde{q}^T x\right)^2 xx^T, \qquad F_{\tilde{q}}^{L_4}(x) \doteq \frac{1}{3\theta\sigma^3}\left(\tilde{q}^T x\right)^4 \tag{F.208}$$

and

$$g^{L_2}(\tilde{q}) \doteq 3\left(1-\theta\right)\operatorname{diag}(\tilde{q}^{\circ 2}) + \theta\left(\|\tilde{q}\|_2^2 I_r + 2\tilde{q}\tilde{q}^T\right), \qquad g^{L_4}(\tilde{q}) \doteq \theta\|\tilde{q}\|_2^4 + (1-\theta)\|\tilde{q}\|_4^4 \tag{F.209}$$

such that

$$\mathbb{E}\left[F_{\tilde{q}}^{L_2}(x)\right] = g^{L_2}(\tilde{q}), \qquad \mathbb{E}\left[F_{\tilde{q}}^{L_4}(x)\right] = g^{L_4}(\tilde{q}) \tag{F.210}$$

from Lemma 2.5. Using $\zeta = \|\zeta\|_2\,\tilde{q}$ and $\|\zeta\|_2 \le 1$ further yields

$$\|\operatorname{Hess} F(q) - \operatorname{Hess} f(q)\|_{\mathrm{op}} \le \left\|\frac{1}{n}\sum_{k=1}^n F_{\tilde{q}}^{L_2}(x_k) - g^{L_2}(\tilde{q})\right\|_{\mathrm{op}} + \left|\frac{1}{3\theta\sigma^4 n}\sum_{k=1}^n F_{\tilde{q}}^{L_4}(x_k) - g^{L_4}(\tilde{q})\right|. \tag{F.211}$$

Notice that the second term has been studied in Appendix F.9.1. It suffices to invoke Lemma G.5 with $n_1 = d_1 = d_2 = r$, $n_2 = 1$ and $p = n$ to bound from above

$$\sup_{\tilde{q}\in\mathbb{S}^{r-1}}\left\|\frac{1}{n}\sum_{k=1}^n F_{\tilde{q}}^{L_2}(x_k) - g^{L_2}(\tilde{q})\right\|_{\mathrm{op}}.$$

We note that $x_{ki}$ is sub-Gaussian for all $1 \le k \le n$ and $1 \le i \le r$. W.L.O.G., we assume $\sigma^2 = 1$.

**Verification of Condition 1:** By $\|\tilde{q}\|_2 = 1$, notice that

$$\left\|\mathbb{E}\left[F_{\tilde{q}}^{L_2}(x)\right]\right\|_{\mathrm{op}} = \left\|3\left(1-\theta\right)\operatorname{diag}(\tilde{q}^{\circ 2}) + \theta\left(\|\tilde{q}\|_2^2 I_r + 2\tilde{q}\tilde{q}^T\right)\right\|_{\mathrm{op}} \tag{F.212}$$

$$\le 3\left(1-\theta\right)\|\tilde{q}\|_2^2 + 3\theta \tag{F.213}$$

$$= 3. \tag{F.214}$$

For any $\tilde{q}_1, \tilde{q}_2 \in \mathbb{S}^{r-1}$,

$$\left\|\mathbb{E}\left[F_{\tilde{q}_1}^{L_2}(x)\right] - \mathbb{E}\left[F_{\tilde{q}_2}^{L_2}(x)\right]\right\|_{\mathrm{op}} \tag{F.215}$$

$$= \left\|3\left(1-\theta\right)\operatorname{diag}(\tilde{q}_1^{\circ 2}) + 2\theta\tilde{q}_1\tilde{q}_1^T - 3\left(1-\theta\right)\operatorname{diag}(\tilde{q}_2^{\circ 2}) - 2\theta\tilde{q}_2\tilde{q}_2^T + \theta\|\tilde{q}_1\|_2^2 I - \theta\|\tilde{q}_2\|_2^2 I\right\|_{\mathrm{op}} \tag{F.216}$$

$$\le 3\left(1-\theta\right)\left\|\operatorname{diag}(\tilde{q}_1^{\circ 2}) - \operatorname{diag}(\tilde{q}_2^{\circ 2})\right\|_{\mathrm{op}} + 2\theta\left\|\tilde{q}_1\tilde{q}_1^T - \tilde{q}_2\tilde{q}_2^T\right\|_2 + \theta\left|\|\tilde{q}_1\|_2^2 - \|\tilde{q}_2\|_2^2\right| \tag{F.217}$$

$$\le 6\left(1-\theta\right)\|\tilde{q}_2 - \tilde{q}_1\|_\infty + 4\theta\|\tilde{q}_2 - \tilde{q}_1\|_{\mathrm{op}} + 2\theta\|\tilde{q}_2 - \tilde{q}_1\|_{\mathrm{op}} \tag{F.218}$$

$$\le 6\|\tilde{q}_2 - \tilde{q}_1\|_2. \tag{F.219}$$

We thus have $L_f = 6$ and $B_f = 3$.

**Verification of Condition 2:** We again work on the event $\mathcal{E}'$ in (F.165) such that, for each $i \in [n]$,

$$\left\|F_{\tilde{q}}^{L_2}(\bar{x}_i)\right\|_{\mathrm{op}} = \frac{1}{\theta\sigma^3}\left\|\left(\tilde{q}^T\bar{x}\right)^2 \bar{x}_i\bar{x}_i^T\right\|_{\mathrm{op}} \le \theta^{-1}\|\tilde{q}\|_2^2\|\bar{x}_i\|_2^4 \tag{F.220}$$

$$\le C\left(\theta r^2 + \frac{\log^2 n}{\theta}\right). \tag{F.221}$$

Lemma G.9 in Appendix G with some straightforward modifications ensures

$$\sup_{q\in\mathbb{S}^{r-1}}\left\|\mathbb{E}\left[F_{\tilde{q}}^{L_2}(\bar{x}_i)\left(F_{\tilde{q}}^{L_2}(\bar{x}_i)\right)^T\right]\right\|_{\mathrm{op}} \le \sup_{q\in\mathbb{S}^{r-1}}\left\|\mathbb{E}\left[F_{\tilde{q}}^{L_2}(x_i)\left(F_{\tilde{q}}^{L_2}(x_i)\right)^T\right]\right\|_{\mathrm{op}} \le c\theta^{-1}r^2 \tag{F.222}$$

for some constant $c > 0$. Therefore, we have

$$R_1 = C\left(\theta r^2 + \frac{\log^2 n}{\theta}\right), \qquad R_2 = c\theta^{-1}r^2. \tag{F.223}$$

On the other hand, for any $\tilde{\boldsymbol{q}}_1, \tilde{\boldsymbol{q}}_2 \in \mathbb{S}^{r-1}$,

$$\left\| F_{\tilde{\boldsymbol{q}}_1}^{L_2}\left(\bar{\boldsymbol{x}}_i\right) - F_{\tilde{\boldsymbol{q}}_2}^{L_2}\left(\bar{\boldsymbol{x}}_i\right) \right\|_{\mathrm{op}} \le \frac{1}{\theta\sigma^3}\left| \left(\tilde{\boldsymbol{q}}_1^T\bar{\boldsymbol{x}}_i\right)^2 - \left(\tilde{\boldsymbol{q}}_2^T\bar{\boldsymbol{x}}_i\right)^2 \right| \left\| \bar{\boldsymbol{x}}_i\bar{\boldsymbol{x}}_i^T \right\|_{\mathrm{op}} \tag{F.224}$$

$$\le \frac{2}{\theta}\left\| \bar{\boldsymbol{x}}_i \right\|_2^4 \left\| \tilde{\boldsymbol{q}}_1 - \tilde{\boldsymbol{q}}_2 \right\|_2 \tag{F.225}$$

$$\le 2R_1\left\| \tilde{\boldsymbol{q}}_1 - \tilde{\boldsymbol{q}}_2 \right\|_2 \tag{F.226}$$

on the event $\mathcal{E}'$, which implies $\bar{L}_f = 2R_1$.

Finally invoke Lemma G.5 with $M = C'R_1(n+r)$ to conclude the proof. ∎

## F.10 Concentration inequalities when $A$ is full column rank

In this section we provide deviation bounds for the objective values, Riemannian gradients and Hessian matrices between $F_g(\boldsymbol{q})$ and $\bar{f}_g(\boldsymbol{q})$ defined as

$$F_g(\boldsymbol{q}) := -\frac{\theta n}{12}\left\| \bar{\boldsymbol{Y}}^T\boldsymbol{q} \right\|_4^4, \tag{F.227}$$

$$\bar{f}_g(\boldsymbol{q}) := -\frac{1}{4}\left[ (1-\theta)\left\| \bar{\boldsymbol{A}}^T\boldsymbol{q} \right\|_4^4 + \theta\left\| \bar{\boldsymbol{A}}^T\boldsymbol{q} \right\|_2^4 \right] \tag{F.228}$$

where

$$\bar{\boldsymbol{Y}} = \left( \left(\boldsymbol{Y}\boldsymbol{Y}^T\right)^+ \right)^{\frac{1}{2}}\boldsymbol{Y} = \boldsymbol{D}\boldsymbol{Y},$$

$$\bar{\boldsymbol{A}} = \left( \left(\boldsymbol{A}\boldsymbol{A}^T\right)^+ \right)^{\frac{1}{2}}\boldsymbol{A} = \boldsymbol{U}_A\boldsymbol{V}_A^T. \tag{F.229}$$

### F.10.1 Deviation inequalities between the function values

Recall that $M_n$ is defined in (F.155).

**Lemma F.16** *Under Assumptions 2.1 and 2.2, assume*

$$n \ge C\max\left\{ \frac{r\log^3 n}{\theta}, \ \theta r^3\log n, \ \frac{r^2}{\theta\sqrt{\theta}}, \ \frac{r\log n}{\theta^2\sqrt{\theta}} \right\} \tag{F.230}$$

*for some constant $C > 0$. With probability greater than $1 - cn^{-c'} - 2e^{-c''r}$,*

$$\sup_{\boldsymbol{q}\in\mathbb{S}^{p-1}}\left| F_g\left(\boldsymbol{q}\right) - \bar{f}_g\left(\boldsymbol{q}\right) \right| \lesssim \left( \sqrt{r\theta} + \sqrt{\log n} \right)\sqrt{\frac{r}{\theta^2 n\sqrt{\theta n}}} + \left( \theta r^2 + \frac{\log^2 n}{\theta} \right)\frac{r\log n}{n}.$$

**Proof.** First we introduce

$$f_g(\boldsymbol{q}) := -\frac{1}{12\sigma^4\theta n}\left\| \boldsymbol{q}^T\bar{\boldsymbol{A}}\boldsymbol{X} \right\|_4^4. \tag{F.231}$$

The proof of Lemma 2.5 yields

$$\mathbb{E}\left[ f_g(\boldsymbol{q}) \right] = \bar{f}_g(\boldsymbol{q}). \tag{F.232}$$

Triangle inequality gives

$$\sup_{\boldsymbol{q}\in\mathbb{S}^{p-1}}\left| F_g\left(\boldsymbol{q}\right) - \bar{f}_g\left(\boldsymbol{q}\right) \right| \le \underbrace{\sup_{\boldsymbol{q}\in\mathbb{S}^{p-1}}\left| F_g\left(\boldsymbol{q}\right) - f_g\left(\boldsymbol{q}\right) \right|}_{\Gamma_1} + \underbrace{\sup_{\boldsymbol{q}\in\mathbb{S}^{p-1}}\left| f_g\left(\boldsymbol{q}\right) - \bar{f}_g\left(\boldsymbol{q}\right) \right|}_{\Gamma_2}. \tag{F.233}$$

**Controlling $\Gamma_1$:** Define

$$\boldsymbol{v}_0 := \sqrt{\theta n \sigma^2} \bar{\boldsymbol{Y}}^T \boldsymbol{q} \qquad \text{and} \qquad \boldsymbol{v}_1 := \left(\bar{\boldsymbol{A}}\boldsymbol{X}\right)^T \boldsymbol{q} \tag{F.234}$$

We have

$$\begin{aligned}
\Gamma_1 &= \frac{1}{12\sigma^4\theta n} \sup_{\boldsymbol{q}\in\mathbb{S}^{p-1}} \left| \theta^2 n^2 \sigma^4 \left\| \boldsymbol{q}^T\bar{\boldsymbol{Y}}\right\|_4^4 - \left\| \boldsymbol{q}^T\bar{\boldsymbol{A}}\boldsymbol{X}\right\|_4^4 \right| \\
&= \sup_{\boldsymbol{q}\in\mathbb{S}^{p-1}} \frac{1}{12\sigma^4\theta n} \left| \left( \|\boldsymbol{v}_0\|_4 - \|\boldsymbol{v}_1\|_4 \right) \left( \|\boldsymbol{v}_0\|_4 + \|\boldsymbol{v}_1\|_4 \right) \left( \|\boldsymbol{v}_0\|_4^2 + \|\boldsymbol{v}_1\|_4^2 \right) \right| \\
&\lesssim \sup_{\boldsymbol{q}\in\mathbb{S}^{p-1}} \frac{1}{\sigma^4\theta n} \|\boldsymbol{v}_0 - \boldsymbol{v}_1\|_4 \left( \|\boldsymbol{v}_0\|_4^3 + \|\boldsymbol{v}_1\|_4^3 \right).
\end{aligned} \tag{F.235}$$

Invoking Lemma F.17 gives

$$\mathbb{P}\left\{ \Gamma_1 \lesssim \left( \sqrt{r\theta} + \sqrt{\log n} \right) \frac{1}{\theta} \sqrt{\frac{r}{n\sqrt{\theta n}}} \right\} \geq 1 - 2e^{-cr} - c'n^{-c''r}. \tag{F.236}$$

**Controlling $\Gamma_2$:** Notice that that $\bar{\boldsymbol{A}}^T\bar{\boldsymbol{A}} = \boldsymbol{I}_r$. We can thus apply Lemma F.13 by replacing $F(\boldsymbol{q})$ and $f(\boldsymbol{q})$ with $f_g(\boldsymbol{q})$ and $\bar{f}_g(\boldsymbol{q})$, respectively, to obtain

$$\mathbb{P}\left\{ \Gamma_2 \lesssim \sqrt{\frac{r\log(M_n)}{\theta n}} + \frac{M_n}{n}\frac{r\log(M_n)}{n} \right\} \geq 1 - cn^{-c'}. $$

Combining the bounds of $\Gamma_1$ and $\Gamma_2$ and using (F.230) to simplify the expressions complete the proof. ∎

Recall that, for any $\boldsymbol{q} \in \mathbb{S}^{p-1}$,

$$\boldsymbol{v}_0 := \sqrt{\theta n \sigma^2} \bar{\boldsymbol{Y}}^T \boldsymbol{q} \qquad \text{and} \qquad \boldsymbol{v}_1 := \left(\bar{\boldsymbol{A}}\boldsymbol{X}\right)^T \boldsymbol{q}$$

with $\bar{\boldsymbol{A}} = \boldsymbol{U}_A \boldsymbol{V}_A^T$ and $\bar{\boldsymbol{Y}} = \boldsymbol{D}\boldsymbol{Y}$.

**Lemma F.17** *Assume $n \geq Cr/\theta^2$ for some constant $C > 0$. With probability $1 - 2e^{-cr} - 2n^{-c'}$ for some constant $c, c' > 0$, one has*

$$\sup_{\boldsymbol{q}\in\mathbb{S}^{p-1}} \|\boldsymbol{v}_0 - \boldsymbol{v}_1\|_\infty \lesssim \sigma \left( \sqrt{r\theta} + \sqrt{\log n} \right) \frac{1}{\theta}\sqrt{\frac{r}{n}}. \tag{F.237}$$

*Furthermore, if additionally (F.230) holds, then with probability $1 - 2e^{-cr} - c'n^{-c''r}$,*

$$\sup_{\boldsymbol{q}\in\mathbb{S}^{p-1}} \|\boldsymbol{v}_1\|_4 \lesssim (\theta n \sigma^4)^{1/4}, \qquad \sup_{\boldsymbol{q}\in\mathbb{S}^{p-1}} \|\boldsymbol{v}_0\|_4 \lesssim (\theta n \sigma^4)^{1/4}. \tag{F.238}$$

**Proof.** We work on the event $\mathcal{E}'$, defined in (F.165), intersecting with

$$\mathcal{E}'' := \left\{ \left\| \sqrt{\theta n \sigma^2}\boldsymbol{D}\boldsymbol{A} - \bar{\boldsymbol{A}} \right\|_{\mathrm{op}} \lesssim \frac{1}{\theta}\sqrt{\frac{r}{n}} \right\}, \tag{F.239}$$

which, according to Lemmas G.3 and G.1, holds with probability $1 - 2e^{-cr} - 2n^{-c'}$. Recall $\bar{\boldsymbol{Y}} = \boldsymbol{D}\boldsymbol{Y} = \boldsymbol{D}\boldsymbol{A}\boldsymbol{X}$. By definition,

$$\begin{aligned}
\|\boldsymbol{v}_0 - \boldsymbol{v}_1\|_\infty &= \max_{t\in[n]} \left| \boldsymbol{q}^T \left( \bar{\boldsymbol{A}} - \sqrt{\theta n \sigma^2}\boldsymbol{D}\boldsymbol{A} \right) \boldsymbol{x}_t \right| \tag{F.240} \\
&\leq \max_{t\in[n]} \|\boldsymbol{x}_t\|_2 \|\boldsymbol{q}\|_2 \left\| \sqrt{\theta n \sigma^2}\boldsymbol{D}\boldsymbol{A} - \bar{\boldsymbol{A}} \right\|_{\mathrm{op}} \tag{F.241} \\
&\lesssim \sigma \left( \sqrt{r\theta} + \sqrt{\log n} \right) \frac{1}{\theta}\sqrt{\frac{r}{n}} \qquad \text{(by } \mathcal{E}' \cap \mathcal{E}'' \text{).} \tag{F.242}
\end{aligned}$$

To bound from above $\|\boldsymbol{v}_1\|_4$, by recalling $F(\boldsymbol{q})$ and $f(\boldsymbol{q})$ from (F.154) with $\bar{\boldsymbol{A}}$ in lieu of $\boldsymbol{A}$, we observe that

$$\|\boldsymbol{v}_1\|_4^4 = \left\|\boldsymbol{q}^T \bar{\boldsymbol{A}} \boldsymbol{X}\right\|_4^4 = 12\theta n \sigma^4 |F(\boldsymbol{q})| \le 12\theta n \sigma^4 \left(|F(\boldsymbol{q}) - f(\boldsymbol{q})| + |f(\boldsymbol{q})|\right). \tag{F.243}$$

By Lemma F.13 and $|f(\boldsymbol{q})| \le 1$ from its proof, we obtain

$$\sup_{\boldsymbol{q} \in \mathbb{S}^{p-1}} \|\boldsymbol{v}_1\|_4^4 \le 12\theta n \sigma^4 \left(1 + \sqrt{\frac{r \log(M_n)}{\theta n}} + \frac{M_n}{n} \frac{r \log(M_n)}{n}\right) \tag{F.244}$$

with probability at least $1 - (nr)^{-2} - c M_n^{-c'r}$ for some constants $c, c' > 0$. Here $M_n$ is defined in (F.155). The result then follows by invoking condition (F.230) and noting that $\log M_n \lesssim \log n$.

Finally, since

$$\|\boldsymbol{v}_0\|_4 \le \|\boldsymbol{v}_1\|_4 + \|\boldsymbol{v}_0 - \boldsymbol{v}_1\|_4 \le \|\boldsymbol{v}_1\|_4 + n^{1/4}\|\boldsymbol{v}_0 - \boldsymbol{v}_1\|_\infty,$$

the last result follows by combining the previous two results. $\blacksquare$

### F.10.2 Deviation inequalities between the Riemannian gradients

In this section, we derive the deviation inequalities between the Riemannian gradient of $F_g(\boldsymbol{q})$ and that of $\bar{f}_g(\boldsymbol{q})$. Note that, for any $\boldsymbol{q} \in \mathbb{S}^{p-1}$,

$$\operatorname{grad} F_g(\boldsymbol{q}) \doteq \operatorname{grad}_{\|\boldsymbol{q}\|_2 = 1} F_g(\boldsymbol{q}) = -\frac{\theta n}{3} \boldsymbol{P}_{\boldsymbol{q}\perp} \sum_{k=1}^{n} \left(\boldsymbol{q}^T \bar{\boldsymbol{Y}}_k\right)^3 \bar{\boldsymbol{Y}}_k, \tag{F.245}$$

$$\operatorname{grad} \bar{f}_g(\boldsymbol{q}) \doteq \operatorname{grad}_{\|\boldsymbol{q}\|_2 = 1} \bar{f}_g(\boldsymbol{q}) = -\boldsymbol{P}_{\boldsymbol{q}\perp} \left[(1-\theta) \sum_{j=1}^{r} \bar{\boldsymbol{A}}_j (\boldsymbol{q}^T \bar{\boldsymbol{A}}_j)^3 + \theta \|\boldsymbol{q}^T \bar{\boldsymbol{A}}\|_2^2 \bar{\boldsymbol{A}} \bar{\boldsymbol{A}}^T \boldsymbol{q}\right]. \tag{F.246}$$

Here $\bar{\boldsymbol{Y}}$ and $\bar{\boldsymbol{A}}$ are defined in (F.229).

**Lemma F.18** *Under Assumptions 2.1 and 2.2, assume*

$$n \ge C \max\left\{\frac{r \log^3 n}{\theta}, \; \theta r^3 \log n, \; \frac{r^2}{\theta\sqrt{\theta}}, \; \frac{r^2 \log n}{\theta}, \; \frac{r \log n}{\theta^2 \sqrt{\theta}}\right\} \tag{F.247}$$

*for some constant $C > 0$. With probability greater than $1 - cn^{-c'} - 2e^{-c''r}$,*

$$\sup_{\boldsymbol{q} \in \mathbb{S}^{p-1}} \left\|\operatorname{grad} F_g(\boldsymbol{q}) - \operatorname{grad} \bar{f}_g(\boldsymbol{q})\right\|_2 \lesssim \sqrt{\frac{r \log n}{\theta^2 n}} + \sqrt{\frac{r^2 \log n}{\theta n}} + \left(\theta r^2 + \frac{\log^2 n}{\theta}\right) \frac{r \log n}{n}$$

**Proof.** Recall $f_g(\boldsymbol{q})$ from (F.231). Its Riemannian gradient is

$$\operatorname{grad} f_g(\boldsymbol{q}) := \operatorname{grad}_{\|\boldsymbol{q}\|_2 = 1} f_g(\boldsymbol{q}) = -\frac{1}{3\theta\sigma^4 n} \boldsymbol{P}_{\boldsymbol{q}\perp} \sum_{k=1}^{n} \left(\boldsymbol{q}^T \bar{\boldsymbol{A}} \boldsymbol{x}_k\right)^3 \bar{\boldsymbol{A}} \boldsymbol{x}_k. \tag{F.248}$$

We have

$$\sup_{\boldsymbol{q} \in \mathbb{S}^{p-1}} \|\operatorname{grad} F_g(\boldsymbol{q}) - \operatorname{grad} \bar{f}_g(\boldsymbol{q})\|_2 \tag{F.249}$$

$$\le \underbrace{\sup_{\boldsymbol{q} \in \mathbb{S}^{p-1}} \|\operatorname{grad} F_g(\boldsymbol{q}) - \operatorname{grad} f_g(\boldsymbol{q})\|_2}_{\Gamma_1} + \underbrace{\sup_{\boldsymbol{q} \in \mathbb{S}^{p-1}} \|\operatorname{grad} f_g(\boldsymbol{q}) - \operatorname{grad} \bar{f}_g(\boldsymbol{q})\|_2}_{\Gamma_2}. \tag{F.250}$$

**Controlling $\Gamma_1$**   Recall that $\boldsymbol{v}_0$ and $\boldsymbol{v}_1$ are defined in (F.234). We have:

$$\Gamma_1 = \sup_{\boldsymbol{q} \in \mathbb{S}^{p-1}} \|\operatorname{grad} F_g(\boldsymbol{q}) - \operatorname{grad} f_g(\boldsymbol{q})\|_2$$

$$= \sup_{\boldsymbol{q} \in \mathbb{S}^{p-1}} \frac{1}{3\theta\sigma^4 n} \left\| \theta^2 n^2 \sigma^4 \boldsymbol{P}_{\boldsymbol{q}\perp} \sum_{k=1}^{n} \left(\boldsymbol{q}^T \bar{\boldsymbol{Y}}_k\right)^3 \bar{\boldsymbol{Y}}_k - \boldsymbol{P}_{\boldsymbol{q}\perp} \sum_{k=1}^{n} \left(\boldsymbol{q}^T \bar{\boldsymbol{A}} \boldsymbol{x}_k\right)^3 \bar{\boldsymbol{A}} \boldsymbol{x}_k \right\|_2$$

$$\leq \sup_{\boldsymbol{q} \in \mathbb{S}^{p-1}} \frac{1}{3\theta\sigma^4 n} \left\| \sum_{k=1}^{n} \left( \sqrt{\theta n \sigma^2} \bar{\boldsymbol{Y}}_k \boldsymbol{v}_{0k}^3 - \bar{\boldsymbol{A}} \boldsymbol{x}_k \boldsymbol{v}_{1k}^3 \right) \right\|_2$$

$$\leq \underbrace{\sup_{\boldsymbol{q} \in \mathbb{S}^{p-1}} \frac{1}{3\theta\sigma^4 n} \left\| \sum_{k=1}^{n} \left( \sqrt{\theta n \sigma^2} \bar{\boldsymbol{Y}}_k - \bar{\boldsymbol{A}} \boldsymbol{x}_k \right) \boldsymbol{v}_{0k}^3 \right\|_2}_{\Gamma_{11}}$$

$$+ \underbrace{\sup_{\boldsymbol{q} \in \mathbb{S}^{p-1}} \frac{1}{3\theta\sigma^4 n} \left\| \sum_{k=1}^{n} \bar{\boldsymbol{A}} \boldsymbol{x}_k (\boldsymbol{v}_{0k}^3 - \boldsymbol{v}_{1k}^3) \right\|_2}_{\Gamma_{12}}.$$

For $\Gamma_{11}$, we obtain

$$\Gamma_{11} = \sup_{\boldsymbol{q} \in \mathbb{S}^{p-1}} \frac{1}{3\theta\sigma^4 n} \left\| \left( \sqrt{\theta n \sigma^2} \boldsymbol{D} \boldsymbol{A} - \bar{\boldsymbol{A}} \right) \sum_{k=1}^{n} \boldsymbol{x}_k \boldsymbol{v}_{0k}^3 \right\|_2 \tag{F.251}$$

$$\leq \sup_{\boldsymbol{q} \in \mathbb{S}^{p-1}} \frac{1}{3\theta\sigma^4 n} \left\| \sum_{k=1}^{n} \boldsymbol{x}_k \boldsymbol{v}_{0k}^3 \right\|_2 \left\| \sqrt{\theta n \sigma^2} \boldsymbol{D} \boldsymbol{A} - \bar{\boldsymbol{A}} \right\|_{\mathrm{op}} \tag{F.252}$$

$$\leq \sup_{\boldsymbol{q} \in \mathbb{S}^{p-1}} \frac{1}{3\theta\sigma^4 n} \left( \left\| \sum_{k=1}^{n} \boldsymbol{x}_k \boldsymbol{v}_{1k}^3 \right\|_2 + \left\| \sum_{k=1}^{n} \boldsymbol{x}_k (\boldsymbol{v}_{0k}^3 - \boldsymbol{v}_{1k}^3) \right\|_2 \right) \left\| \sqrt{\theta n \sigma^2} \boldsymbol{D} \boldsymbol{A} - \bar{\boldsymbol{A}} \right\|_{\mathrm{op}}. \tag{F.253}$$

Observing

$$\sum_{k=1}^{n} \boldsymbol{x}_k \boldsymbol{v}_{1k}^3 = 3\theta n \sigma^4 \frac{1}{n} \left\| \bar{\boldsymbol{A}}^T \boldsymbol{q} \right\|_2^3 \sum_{k=1}^{n} F_{\tilde{\boldsymbol{q}}}(\boldsymbol{x}_k)$$

with $F_{\tilde{\boldsymbol{q}}}(\boldsymbol{x}_k)$ defined in (F.183) and $\tilde{\boldsymbol{q}} = \frac{\bar{\boldsymbol{A}}^T \boldsymbol{q}}{\|\bar{\boldsymbol{A}}^T \boldsymbol{q}\|_2} \in \mathbb{S}^{r-1}$. We also have

$$\mathbb{E}[F_{\tilde{\boldsymbol{q}}}(\boldsymbol{x}_k)] = g(\tilde{\boldsymbol{q}}) \stackrel{(F.184)}{=} \left[ (1 - \theta)(\tilde{\boldsymbol{q}})^{\circ 3} + \theta \|\tilde{\boldsymbol{q}}\|_2^2 \tilde{\boldsymbol{q}} \right]$$

with

$$\sup_{\tilde{\boldsymbol{q}} \in \mathbb{S}^{r-1}} \|g(\tilde{\boldsymbol{q}})\|_2 \leq 1.$$

Lemma F.14 and its proof guarantee that

$$\sup_{\boldsymbol{q} \in \mathbb{S}^{p-1}} \frac{1}{3\theta n \sigma^4} \left\| \sum_{k=1}^{n} \boldsymbol{x}_k \boldsymbol{v}_{1t}^3 \right\|_2 \leq \left\| \bar{\boldsymbol{A}}^T \boldsymbol{q} \right\|_2^3 \sup_{\tilde{\boldsymbol{q}} \in \mathbb{S}^{r-1}} \left\| \frac{1}{n} \sum_{k=1}^{n} F_{\tilde{\boldsymbol{q}}}(\boldsymbol{x}_k) - g(\tilde{\boldsymbol{q}}) \right\|_2 + \left\| \bar{\boldsymbol{A}}^T \boldsymbol{q} \right\|_2^3 \sup_{\tilde{\boldsymbol{q}} \in \mathbb{S}^{r-1}} \|g(\tilde{\boldsymbol{q}})\|_2 \tag{F.254}$$

$$\lesssim 1 + \sqrt{\frac{r^2 \log(M_n)}{\theta n}} + \frac{M_n}{n} \frac{r \log(M_n)}{n} \tag{F.255}$$

$$\lesssim 1 \tag{F.256}$$

with probability at least $1 - cn^{-c'}$, where $M_n$ is defined in (F.155). We used condition (F.247) to simplify the expressions in the last step and $\left\| \bar{\boldsymbol{A}}^T \boldsymbol{q} \right\|_2^3 \leq 1$ in the second step. Invoke $\mathcal{E}''$ in (F.239) to conclude

$$\Gamma_{11} \lesssim \frac{1}{\theta} \sqrt{\frac{r}{n}} (1 + \Gamma_{12}) \tag{F.257}$$

with probability at least $1 - cn^{-c'} - 2e^{-c''r}$.

To control $\Gamma_{12}$, we have

$$\Gamma_{12} = \sup_{\boldsymbol{q} \in \mathbb{S}^{p-1}} \frac{1}{3\theta\sigma^4 n} \left\| \sum_{k=1}^{n} \boldsymbol{x}_k (\boldsymbol{v}_{0k}^3 - \boldsymbol{v}_{1k}^3) \right\|_2 \tag{F.258}$$

$$\leq \sup_{\boldsymbol{q} \in \mathbb{S}^{p-1}} \frac{1}{3\theta\sigma^4 n} \left( \left\| \sum_{k=1}^{n} \boldsymbol{x}_k (\boldsymbol{v}_{0k} - \boldsymbol{v}_{1k}) \boldsymbol{v}_{1k}^2 \right\|_2 + \left\| \sum_{k=1}^{n} \boldsymbol{x}_k \boldsymbol{v}_{0k} (\boldsymbol{v}_{0k} - \boldsymbol{v}_{1k})(\boldsymbol{v}_{0k} + \boldsymbol{v}_{1k}) \right\|_2 \right) \tag{F.259}$$

$$\lesssim \frac{1}{\theta n\sigma^4} \|\boldsymbol{X}\|_{\mathrm{op}} \sup_{\boldsymbol{q} \in \mathbb{S}^{p-1}} \left( \left\| \boldsymbol{v}_1^{\circ 2} \circ (\boldsymbol{v}_0 - \boldsymbol{v}_1) \right\|_2 + \left\| \boldsymbol{v}_0^{\circ 2} \circ (\boldsymbol{v}_0 - \boldsymbol{v}_1) \right\|_2 \right) \tag{F.260}$$

$$\leq \frac{1}{\theta n\sigma^4} \|\boldsymbol{X}\|_{\mathrm{op}} \sup_{\boldsymbol{q} \in \mathbb{S}^{p-1}} \left( \|\boldsymbol{v}_0\|_4^2 + \|\boldsymbol{v}_1\|_4^2 \right) \|\boldsymbol{v}_0 - \boldsymbol{v}_1\|_\infty \tag{F.261}$$

where in the penultimate step we used

$$\|\boldsymbol{v}^{\circ 2} \circ \boldsymbol{v}'\|_2^2 = \sum_i \boldsymbol{v}_i^4 (\boldsymbol{v}_i')^2 \leq \|\boldsymbol{v}'\|_\infty^2 \|\boldsymbol{v}\|_4^4.$$

Invoking Lemma G.2 and Lemma F.17 yields

$$\Gamma_{12} \lesssim \left( \sqrt{r\theta} + \sqrt{\log n} \right) \frac{1}{\theta} \sqrt{\frac{r}{n}} \tag{F.262}$$

with probability at least $1 - cn^{-c'} - 2e^{-c''r}$.

**Controlling $\Gamma_2$:** Since $\bar{\boldsymbol{A}}^T \bar{\boldsymbol{A}} = \boldsymbol{I}_r$ and direct calculation gives

$$\mathbb{E}\left[ \operatorname{grad} f_g(\boldsymbol{q}) \right] = \operatorname{grad} \bar{f}_g(\boldsymbol{q}). \tag{F.263}$$

Applying lemma F.14 with $F(\boldsymbol{q})$ and $f(\boldsymbol{q})$ replaced by $f_g(\boldsymbol{q})$ and $\bar{f}_g(\boldsymbol{q})$, respectively, gives

$$\mathbb{P}\left\{ \Gamma_2 \lesssim \sqrt{\frac{r^2 \log(M_n)}{\theta n}} + \frac{M_n}{n} \frac{r \log(M_n)}{n} \right\} \geq 1 - cn^{-c'}. \tag{F.264}$$

Finally collecting (F.257), (F.262) and (F.264) and using (F.247) to simplify the expression finish the proof. ∎

### F.10.3 Deviation inequalities of the Riemannian Hessian

In this part we will show that the Hessian of $F_g(\boldsymbol{q})$ concentrates around that of $\bar{f}_g(\boldsymbol{q})$. Notice that, for any $\boldsymbol{q} \in \mathbb{S}^{-1}$,

$$\operatorname{Hess} F_g(\boldsymbol{q}) = -\frac{\theta n}{3} \sum_{k=1}^{n} P_{\boldsymbol{q}\perp} \left[ 3 \left( \boldsymbol{q}^T \bar{\boldsymbol{Y}}_k \right)^2 \bar{\boldsymbol{Y}}_k \left( \bar{\boldsymbol{Y}}_k \right)^T - \left( \boldsymbol{q}^T \bar{\boldsymbol{Y}}_k \right)^4 \boldsymbol{I}_p \right] P_{\boldsymbol{q}\perp}, \tag{F.265}$$

$$\operatorname{Hess} \bar{f}_g(\boldsymbol{q}) = - \left\{ (1-\theta) P_{\boldsymbol{q}\perp} \left[ 3\bar{\boldsymbol{A}} \operatorname{diag}((\bar{\boldsymbol{A}}^T \boldsymbol{q})^{\circ 2}) \bar{\boldsymbol{A}}^T - \|\bar{\boldsymbol{A}}^T \boldsymbol{q}\|_4^4 \boldsymbol{I}_p \right] P_{\boldsymbol{q}\perp} \right.$$

$$\left. + \theta P_{\boldsymbol{q}\perp} \left[ \|\boldsymbol{q}^T \bar{\boldsymbol{A}}\|_2^2 \bar{\boldsymbol{A}} \bar{\boldsymbol{A}}^T + 2\bar{\boldsymbol{A}} \bar{\boldsymbol{A}}^T \boldsymbol{q} \boldsymbol{q}^T \bar{\boldsymbol{A}} \bar{\boldsymbol{A}}^T - \|\boldsymbol{q}^T \bar{\boldsymbol{A}}\|_2^4 \boldsymbol{I}_p \right] P_{\boldsymbol{q}\perp} \right\}. \tag{F.266}$$

Here $\bar{\boldsymbol{Y}}$ and $\bar{\boldsymbol{A}}$ are defined in (F.229).

**Lemma F.19** *Under Assumptions 2.1 and 2.2, assume*

$$n \geq C \max \left\{ \frac{r \log^3 n}{\theta}, \ \frac{r \log n}{\theta^2 \sqrt{\theta}}, \ \frac{r^2}{\theta\sqrt{\theta}}, \ \frac{r^3 \log n}{\theta} \right\} \tag{F.267}$$

*for some constant $C > 0$. With probability greater than $1 - cn^{-c'} - 4e^{-c''r}$,*

$$\sup_{\boldsymbol{q} \in \mathbb{S}^{p-1}} \left\| \operatorname{Hess} F_g(\boldsymbol{q}) - \operatorname{Hess} \bar{f}_g(\boldsymbol{q}) \right\|_{op} \tag{F.268}$$

$$\lesssim \left( \sqrt{r\sqrt{\theta}} + \sqrt{\frac{\log n}{\sqrt{\theta}}} + \log n \right) \sqrt{\frac{r}{\theta^2 n}} + \sqrt{\frac{r^3 \log n}{\theta n}} + \left( \theta r^2 + \frac{\log^2 n}{\theta} \right) \frac{r \log n}{n}. \tag{F.269}$$

**Proof.** Recall $f_g(\boldsymbol{q})$ from (F.231). Notice that

$$\sup_{\boldsymbol{q} \in \mathbb{S}^{p-1}} \left\| \operatorname{Hess} F_g(\boldsymbol{q}) - \operatorname{Hess} \bar{f}_g(\boldsymbol{q}) \right\|_{op} \leq \underbrace{\sup_{\boldsymbol{q} \in \mathbb{S}^{p-1}} \left\| \operatorname{Hess} F_g(\boldsymbol{q}) - \operatorname{Hess} f_g(\boldsymbol{q}) \right\|_{op}}_{\Gamma_1}$$

$$+ \underbrace{\sup_{\boldsymbol{q} \in \mathbb{S}^{p-1}} \left\| \operatorname{Hess} f_g(\boldsymbol{q}) - \operatorname{Hess} \bar{f}_g(\boldsymbol{q}) \right\|_{op}}_{\Gamma_2}. \tag{F.270}$$

Straightforward calculation gives

$$\operatorname{Hess} f_g(\boldsymbol{q}) = -\frac{1}{3\theta\sigma^4 n} \sum_{k=1}^{n} P_{\boldsymbol{q}\perp} \left[ 3 \left( \boldsymbol{q}^T \bar{\boldsymbol{A}} \boldsymbol{X}_k \right)^2 \bar{\boldsymbol{A}} \boldsymbol{X}_k \left( \bar{\boldsymbol{A}} \boldsymbol{X}_k \right)^T - \left( \boldsymbol{q}^T \bar{\boldsymbol{A}} \boldsymbol{X}_k \right)^4 \boldsymbol{I}_p \right] P_{\boldsymbol{q}\perp} \tag{F.271}$$

It remains to bound from above $\Gamma_1$ and $\Gamma_2$ respectively.

**Controlling $\Gamma_1$:** Using the definition of $\boldsymbol{v}_0$ and $\boldsymbol{v}_1$ in (F.234), we have:

$$\Gamma_1 = \sup_{\boldsymbol{q} \in \mathbb{S}^{p-1}} \left\| \operatorname{Hess} F_g(\boldsymbol{q}) - \operatorname{Hess} f_g(\boldsymbol{q}) \right\|_{op} \tag{F.272}$$

$$\leq \frac{1}{3\theta\sigma^4 n} \sup_{\boldsymbol{q} \in \mathbb{S}^{p-1}} \left\| \theta^2 n^2 \sigma^4 \left[ 3 \bar{\boldsymbol{Y}} \operatorname{diag}((\bar{\boldsymbol{Y}}^T \boldsymbol{q})^{\circ 2}) \bar{\boldsymbol{Y}}^T - \left\| \bar{\boldsymbol{Y}}^T \boldsymbol{q} \right\|_4^4 \boldsymbol{I} \right] \right.$$

$$\left. - \left[ 3 \bar{\boldsymbol{A}} \boldsymbol{X} \operatorname{diag}(((\bar{\boldsymbol{A}} \boldsymbol{X})^T \boldsymbol{q})^{\circ 2}) \left( \bar{\boldsymbol{A}} \boldsymbol{X} \right)^T - \left\| (\bar{\boldsymbol{A}} \boldsymbol{X})^T \boldsymbol{q} \right\|_4^4 \boldsymbol{I} \right] \right\|_{op} \tag{F.273}$$

$$\leq \underbrace{\frac{1}{\theta\sigma^4 n} \sup_{\boldsymbol{q} \in \mathbb{S}^{p-1}} \left\| \theta n \sigma^2 \bar{\boldsymbol{Y}} \operatorname{diag}\left( \boldsymbol{v}_0^{\circ 2} \right) \bar{\boldsymbol{Y}}^T - \bar{\boldsymbol{A}} \boldsymbol{X} \operatorname{diag}\left( \boldsymbol{v}_1^{\circ 2} \right) \left( \bar{\boldsymbol{A}} \boldsymbol{X} \right)^T \right\|_{op}}_{\beta_1} \tag{F.274}$$

$$+ \underbrace{\frac{1}{3\theta\sigma^4 n} \sup_{\boldsymbol{q} \in \mathbb{S}^{p-1}} \left| \left\| \boldsymbol{v}_0 \right\|_4^4 - \left\| \boldsymbol{v}_1 \right\|_4^4 \right|}_{\beta_2}. \tag{F.275}$$

**Upper bound for $\beta_1$:** By adding and subtracting terms, we have

$$\beta_1 = \underbrace{\frac{1}{\theta\sigma^4 n} \sup_{\boldsymbol{q} \in \mathbb{S}^{p-1}} \left\| (\sqrt{\theta n \sigma^4} \bar{\boldsymbol{Y}} - \bar{\boldsymbol{A}} \boldsymbol{X}) \operatorname{diag}(\boldsymbol{v}_1^{\circ 2}) \boldsymbol{X}^T \right\|_{op}}_{\beta_{11}} \tag{F.276}$$

$$+ \underbrace{\frac{1}{\theta\sigma^4 n} \sup_{\boldsymbol{q} \in \mathbb{S}^{p-1}} \left\| \sqrt{\theta n \sigma^4} \bar{\boldsymbol{Y}} \operatorname{diag}(\boldsymbol{v}_1^{\circ 2})(\sqrt{\theta n \sigma^4} \bar{\boldsymbol{Y}} - \bar{\boldsymbol{A}} \boldsymbol{X})^T \right\|_{op}}_{\beta_{12}} \tag{F.277}$$

$$+ \underbrace{\frac{1}{\theta\sigma^4 n} \sup_{\boldsymbol{q} \in \mathbb{S}^{p-1}} \left\| \sqrt{\theta n \sigma^4} \bar{\boldsymbol{Y}} \operatorname{diag}(\boldsymbol{v}_1^{\circ 2} - \boldsymbol{v}_0^{\circ 2}) \sqrt{\theta n \sigma^4} \bar{\boldsymbol{Y}}^T \right\|_{op}}_{\beta_{13}}. \tag{F.278}$$

For $\beta_{11}$, by recalling that (F.229), we have

$$\beta_{11} = \frac{1}{\theta\sigma^4 n} \sup_{\boldsymbol{q} \in \mathbb{S}^{p-1}} \left\| (\sqrt{\theta n \sigma^4} \boldsymbol{D} \boldsymbol{A} - \bar{\boldsymbol{A}}) \boldsymbol{X} \operatorname{diag}(\boldsymbol{v}_1^{\circ 2}) \boldsymbol{X}^T \right\|_{op} \tag{F.279}$$

$$\leq \frac{1}{\theta\sigma^4 n} \sup_{\boldsymbol{q}\in\mathbb{S}^{p-1}} \left\| \sum_{t=1}^{n} \boldsymbol{v}_{1t}^2 \boldsymbol{x}_t \boldsymbol{x}_t^T \right\|_{\mathrm{op}} \left\| \sqrt{\theta n\sigma^4}\boldsymbol{D}\boldsymbol{A} - \bar{\boldsymbol{A}} \right\|_{\mathrm{op}}. \tag{F.280}$$

Recalling (F.208) and (F.209), we have

$$\frac{1}{\theta\sigma^4 n} \left\| \sum_{t=1}^{n} \boldsymbol{v}_{1t}^2 \boldsymbol{x}_t \boldsymbol{x}_t^T \right\|_{\mathrm{op}} \leq \left\| \bar{\boldsymbol{A}}^T \boldsymbol{q} \right\|_2^2 \left\| g^{L_2}(\tilde{\boldsymbol{q}}) \right\|_{\mathrm{op}} + \left\| \bar{\boldsymbol{A}}^T \boldsymbol{q} \right\|_2^2 \left\| \frac{1}{n}\sum_{t=1}^{n} F_{\tilde{\boldsymbol{q}}}^{L_2}(\boldsymbol{x}_t) - g^{L_2}(\tilde{\boldsymbol{q}}) \right\|_{\mathrm{op}} \tag{F.281}$$

$$\leq \left\| g^{L_2}(\tilde{\boldsymbol{q}}) \right\|_{\mathrm{op}} + \left\| \frac{1}{n}\sum_{t=1}^{n} F_{\tilde{\boldsymbol{q}}}^{L_2}(\boldsymbol{x}_t) - g^{L_2}(\tilde{\boldsymbol{q}}) \right\|_{\mathrm{op}} \tag{F.282}$$

with $\tilde{\boldsymbol{q}} = \bar{\boldsymbol{A}}^T\boldsymbol{q}/\left\|\bar{\boldsymbol{A}}^T\boldsymbol{q}\right\|_2 \in \mathbb{S}^{r-1}$ and $\|g^{L_2}(\tilde{\boldsymbol{q}})\|_{\mathrm{op}}\leq 3$. Hence, according to the proof of Lemma F.15, invoke Lemma F.15 and $\mathcal{E}''$ in (F.239) together with (F.267) to conclude that

$$\mathbb{P}\left\{ \beta_{11} \lesssim \frac{1}{\theta}\sqrt{\frac{r}{n}} \right\} \geq 1 - cn^{-c'} - 2e^{-c''r}. \tag{F.283}$$

By similar arguments and $\bar{\boldsymbol{Y}} = \boldsymbol{D}\boldsymbol{A}\boldsymbol{X}$, we have

$$\beta_{12} \leq \frac{1}{\theta\sigma^4 n} \sup_{\boldsymbol{q}\in\mathbb{S}^{p-1}} \left\| \sum_{t=1}^{n} \boldsymbol{v}_{1t}^2 \boldsymbol{x}_t \boldsymbol{x}_t^T \right\|_{\mathrm{op}} \left\| \sqrt{\theta n\sigma^4}\boldsymbol{D}\boldsymbol{A} - \bar{\boldsymbol{A}} \right\|_{\mathrm{op}} \left\| \sqrt{\theta n\sigma^4}\boldsymbol{D}\boldsymbol{A} \right\|_{\mathrm{op}}. \tag{F.284}$$

Since on the event $\mathcal{E}''$, condition (F.267) ensures

$$\left\| \sqrt{\theta n\sigma^4}\boldsymbol{D}\boldsymbol{A} \right\|_{\mathrm{op}} \leq 1 + \left\| \sqrt{\theta n\sigma^4}\boldsymbol{D}\boldsymbol{A} - \bar{\boldsymbol{A}} \right\|_{\mathrm{op}} \lesssim 1. \tag{F.285}$$

We obtain

$$\mathbb{P}\left\{ \beta_{12} \lesssim \frac{1}{\theta}\sqrt{\frac{r}{n}} \right\} \geq 1 - cn^{-c'} - 2e^{-c''r}. \tag{F.286}$$

Finally, on the event $\mathcal{E}''$,

$$\beta_{13} \leq \frac{1}{\theta\sigma^4 n} \sup_{\boldsymbol{q}\in\mathbb{S}^{p-1}} \left\| \sum_{t=1}^{n} (\boldsymbol{v}_{1t} - \boldsymbol{v}_{0t})(\boldsymbol{v}_{1t} + \boldsymbol{v}_{0t})\boldsymbol{x}_t\boldsymbol{x}_t^T \right\|_{\mathrm{op}} \left\| \sqrt{\theta n\sigma^4}\boldsymbol{D}\boldsymbol{A} \right\|_{\mathrm{op}}^2 \tag{F.287}$$

$$\lesssim \frac{1}{\theta\sigma^4 n} \sup_{\boldsymbol{q}\in\mathbb{S}^{p-1}} \left\| \sum_{t=1}^{n} (\boldsymbol{v}_{1t} - \boldsymbol{v}_{0t})(\boldsymbol{v}_{1t} + \boldsymbol{v}_{0t})\boldsymbol{x}_t\boldsymbol{x}_t^T \right\|_{\mathrm{op}} \tag{F.288}$$

$$\lesssim \frac{1}{\theta\sigma^4 n} \sup_{\boldsymbol{q}\in\mathbb{S}^{p-1}} \|\boldsymbol{X}\|_{\mathrm{op}}^2 \|\boldsymbol{v}_0 - \boldsymbol{v}_1\|_\infty (\|\boldsymbol{v}_0\|_\infty + \|\boldsymbol{v}_1\|_\infty). \tag{F.289}$$

Since on the event $\mathcal{E}'$ in (F.165),

$$\|\boldsymbol{v}_1\|_\infty = \max_{t\in[n]}\|\boldsymbol{q}^T\bar{\boldsymbol{A}}\boldsymbol{x}_t\|_\infty \lesssim \sigma(\sqrt{r\theta} + \sqrt{\log n}),$$

and

$$\|\boldsymbol{v}_0\|_\infty \leq \|\boldsymbol{v}_1\|_\infty + \|\boldsymbol{v}_0 - \boldsymbol{v}_1\|_\infty,$$

invoke Lemma F.17 and Lemma G.2 to obtain

$$\beta_{13} \lesssim (r\theta + \log n)\frac{1}{\theta}\sqrt{\frac{r}{n}} = \sqrt{\frac{r^3}{n}} + \sqrt{\frac{r\log^2 n}{\theta^2 n}} \tag{F.290}$$

with probability at least $1 - 4e^{-cr} - c'n^{-c''}$.

**Upper bound for $\beta_2$**  Notice that

$$\beta_2 = \frac{1}{4} \sup_{\boldsymbol{q}\in\mathbb{S}^{p-1}} |F_g(\boldsymbol{q}) - f_g(\boldsymbol{q})|. \tag{F.291}$$

Display (F.236) yields

$$\mathbb{P}\left\{ \beta_2 \lesssim \left(\sqrt{r\theta} + \sqrt{\log n}\right)\frac{1}{\theta}\sqrt{\frac{r}{\sqrt{\theta}n}} \right\} \geq 1 - 2e^{-cr} - c'n^{-c''r}. \tag{F.292}$$

**Controlling $\Gamma_2$**   Notice that $\bar{A}^T \bar{A} = I_r$ and simple calculation gives

$$\mathbb{E}\left[\text{Hess}\, f_g\left(q\right)\right] = \text{Hess}\,\bar{f}_g\left(q\right). \tag{F.293}$$

Apply Lemma F.15 with $F(q)$ and $f(q)$ replaced by $f_g\left(q\right)$ and $\bar{f}_g\left(q\right)$, respectively, to obtain

$$\mathbb{P}\left\{\Gamma_2 \lesssim \sqrt{\frac{r^3 \log(M_n)}{\theta n}} + \frac{M_n}{n}\frac{r\log(M_n)}{n}\right\} \geq 1 - cn^{-c'}. \tag{F.294}$$

Finally, collecting (F.283), (F.286), (F.290), (F.292) and (F.294) and using condition (F.267) to simplify expressions complete the proof. ∎

# G Auxiliary lemmas

Recall that the SVD of $A$ is $U_A D_A V_A^T$ and $D$ is defined in (2.6).

**Lemma G.1** *Under Assumptions 2.1 and 2.2, assume $n \geq Cr/\theta^2$ for some constant $C > 0$. With probability at least $1 - 2e^{-cr}$ for some constant $c > 0$, we have:*

$$\left\| \sqrt{\theta n \sigma^2} D A - U_A V_A^T \right\|_{op} \lesssim \frac{1}{\theta} \sqrt{\frac{r}{n}}. \tag{G.1}$$

**Proof.** From (F.129), we have

$$\sqrt{\theta n \sigma^2} D = U_A D_A^{-1} U_A^T + U_A \left( \sqrt{\theta n \sigma^2} Q^T \Lambda^{-1/2} Q - D_A^{-1} \right) U_A^T.$$

where $U$ contains the left $r$ singular vectors of $Y$ and $U_A = UQ$. It then follows

$$\left\| \sqrt{\theta n \sigma^2} D A - U_A V_A^T \right\|_{op} = \left\| U_A \left( \sqrt{\theta n \sigma^2} Q^T \Lambda^{-1/2} Q D_A - I_r \right) V_A^T \right\|_{op} \tag{G.2}$$

$$= \left\| \sqrt{\theta n \sigma^2} Q^T \Lambda^{-1/2} Q D_A - I_r \right\|_{op}. \tag{G.3}$$

The result follows by invoking (F.142). ∎

**Lemma G.2** *Under Assumption 2.1, assume $n \geq Cr/\theta^2$ for some constant $C > 0$. One has*

$$\| X \|_{op} \lesssim \sqrt{\theta n \sigma^2} \tag{G.4}$$

*with probability at least $1 - 2e^{-cr}$. Here $c$ is some positive constant.*

**Proof.** Assume $\sigma^2 = 1$ without loss of generality. Recall from (F.141) that

$$\left\| \frac{1}{n\sigma^2} X X^T - \theta I \right\|_{op} \leq c \left( \sqrt{\frac{r}{n}} + \frac{r}{n} \right) \tag{G.5}$$

with probability at least $1 - 2e^{-c'r}$ for some constants $c, c' > 0$. With the same probability, it follows immediately

$$\frac{1}{n\sigma^2} \| X X^T \|_{op} \leq \theta + c \left( \sqrt{\frac{r}{n}} + \frac{r}{n} \right) \leq c'' \theta$$

provided that

$$n \geq Cr/\theta^2$$

for some constant $C > 0$. ∎

Recall that $X = (x_1, \ldots, x_n) \in \mathbb{R}^{r \times n}$. The following lemma provides the upper bound of $\max_{i \in [n]} \| x_i \|_2$.

**Lemma G.3** *Under Assumption 2.1, we have*

$$\max_{i \in [n]} \| x_i \|_2 \lesssim \sigma \left( \sqrt{r\theta} + \sqrt{\log(n)} \right) \tag{G.6}$$

*with probability at least $1 - 2n^{-c}$. Here $c$ is some positive constant.*

**Proof.** Pick any $i \in [n]$ and assume $\sigma^2 = 1$ without loss of generality. By [16, Theorem 7.30], we have

$$\mathbb{P} \left\{ \left| \| x_i \|_2^2 - r\theta \right| \geq t \right\} \leq 2 \exp \left( -\frac{t^2}{4r\theta + 4t} \right). \tag{G.7}$$

Take $t = c \log n$ to obtain

$$\mathbb{P} \left\{ \left| \| x_i \|_2^2 - r\theta \right| \leq c'(\sqrt{r\theta \log n} + \log n) \right\} \geq 1 - 2n^{-c''}$$

for some constants $c, c', c'' > 0$. Take the union bounds over $1 \leq i \leq n$ to complete the proof. ∎

**Lemma G.4** *Suppose $x \in \mathbb{R}^r$ has i.i.d Bernoulli-Gaussian entries. Let $F_q(x)$ be defined as equation ([F.156](#)). We have*

$$\sup_{q \in \mathbb{S}^{r-1}} \mathbb{E}\left[\|F_q(x)\|_2^2\right] \leq C\theta^{-1} \tag{G.8}$$

*for some constant $C > 0$.*

**Proof.** Note that $x = b \circ g$ where $b \overset{i.i.d.}{\sim} \text{Ber}(\theta)$ and $g \overset{i.i.d.}{\sim} \mathcal{N}(0, \sigma^2)$. We assume $\sigma = 1$ for simplicity. Define $\mathcal{I}$ as the nonzero support of $x$ such that we could write $x = \mathcal{P}_{\mathcal{I}}(g)$. We have

$$\mathbb{E}\left[\|F_q(x)\|_2^2\right] = \left(\frac{1}{12\theta}\right)^2 \mathbb{E}\left[(q^T x)^8\right] = \left(\frac{1}{12\theta}\right)^2 \mathbb{E}\left[\langle \mathcal{P}_{\mathcal{I}}(q), g \rangle^8\right]. \tag{G.9}$$

Since

$$\mathbb{E}\left[\langle \mathcal{P}_{\mathcal{I}}(q), g \rangle^8\right] = (7!!)\,\mathbb{E}_{\mathcal{I}}\left[\|\mathcal{P}_{\mathcal{I}}(q)\|_2^8\right], \tag{G.10}$$

we further obtain

$$\mathbb{E}_{\mathcal{I}}\left[\|\mathcal{P}_{\mathcal{I}}(q)\|_2^8\right] = \sum_{k_1,k_2,k_3,k_4} q_{k_1}^2 \mathbb{1}_{k_1 \in \mathcal{I}} q_{k_2}^2 \mathbb{1}_{k_2 \in \mathcal{I}} q_{k_3}^2 \mathbb{1}_{k_3 \in \mathcal{I}} q_{k_4}^2 \mathbb{1}_{k_4 \in \mathcal{I}}. \tag{G.11}$$

We consider four scenarios.

- Only one index among $k_1, k_2, k_3, k_4$ in $\mathcal{I}$. In this case we have

$$\mathbb{E}_{\mathcal{I}}\left[\|\mathcal{P}_{\mathcal{I}}(q)\|_2^8\right] = \theta \sum_{k_1} q_{k_1}^8 \leq \theta \|q\|_2^8 \tag{G.12}$$

- Two index among $k_1, k_2, k_3, k_4$ in $\mathcal{I}$. In this case we have

$$\mathbb{E}_{\mathcal{I}}\left[\|\mathcal{P}_{\mathcal{I}}(q)\|_2^8\right] = \theta^2 \sum_{k_1,k_2} \left[q_{k_1}^2 q_{k_2}^6 + q_{k_1}^4 q_{k_2}^4 + q_{k_1}^6 q_{k_2}^2\right] \leq 3\theta^2 \|q\|_2^8 \tag{G.13}$$

- Three index among $k_1, k_2, k_3, k_4$ in $\mathcal{I}$. In this case we have

$$\mathbb{E}_{\mathcal{I}}\left[\|\mathcal{P}_{\mathcal{I}}(q)\|_2^8\right] = \theta^3 \sum_{k_1,k_2,k_3} \left[q_{k_1}^2 q_{k_2}^2 q_{k_3}^4\right] \leq \theta^3 \|q\|_2^8 \tag{G.14}$$

- All four index among $k_1, k_2, k_3, k_4$ in $\mathcal{I}$. In this case we have

$$\mathbb{E}_{\mathcal{I}}\left[\|\mathcal{P}_{\mathcal{I}}(q)\|_2^8\right] = \theta^4 \sum_{k_1,k_2,k_3,k_4} \left[q_{k_1}^2 q_{k_2}^2 q_{k_3}^2 q_{k_4}^2\right] \leq \theta^4 \|q\|_2^8 \tag{G.15}$$

Use $\|q\|_2 = 1$ and collect the above four results to obtain

$$\mathbb{E}_{\mathcal{I}}\left[\|\mathcal{P}_{\mathcal{I}}(q)\|_2^8\right] = \theta + 3\theta^2 + \theta^3 + \theta^4 \leq c_1 \theta \tag{G.16}$$

Here $c_1 > 6$. Plugging back into ([G.10](#)) yields

$$\mathbb{E}\left[\|f_q(x)\|_2^2\right] \leq (7!!)\,\frac{\theta}{144\theta^2} \leq C\theta^{-1} \tag{G.17}$$

and finishes our proof. ∎

The following results provide deviation inequalities of the average of i.i.d. functionals of a sub-Gaussian random vector / matrix. They are proved in [44] and we offer a modified version here.

**Lemma G.5 (Theorem** *F*.1**, [35])** *Let* $\boldsymbol{Z}_1, \boldsymbol{Z}_2, \dots, \boldsymbol{Z}_p$ *be i.i.d realizations of a random matrix* $\boldsymbol{Z} \in \mathbb{R}^{n_1 \times n_2}$ *satisfying*

$$\mathbb{E}[\boldsymbol{Z}] = \boldsymbol{0}, \qquad \mathbb{P}(|Z_{ij}| > t) \le 2\exp\left(-\frac{t^2}{2\sigma^2}\right), \quad \forall 1 \le i \le n_1, 1 \le j \le n_2. \qquad \text{(G.18)}$$

*For any fixed* $\boldsymbol{q} \in \mathbb{S}^{n-1}$*, define a function* $f_{\boldsymbol{q}} : \mathbb{R}^{n_1 \times n_2} \to \mathbb{R}^{d_1 \times d_2}$ *such that the following conditions hold.*

- **Condition 1.** *There exists some positive numbers* $B_f$ *and* $L_f$ *such that*

$$\|\mathbb{E}[f_{\boldsymbol{q}}(\boldsymbol{Z})]\|_{\mathrm{op}} \le B_f, \qquad\qquad\qquad\qquad\qquad \text{(G.19)}$$
$$\|\mathbb{E}[f_{\boldsymbol{q}_1}(\boldsymbol{Z})] - \mathbb{E}[f_{\boldsymbol{q}_2}(\boldsymbol{Z})]\|_{\mathrm{op}} \le L_f \|\boldsymbol{q}_1 - \boldsymbol{q}_2\|_2, \quad \forall \boldsymbol{q}_1, \boldsymbol{q}_2 \in \mathbb{S}^{n-1}. \qquad \text{(G.20)}$$

- **Condition 2.** *Define* $\bar{\boldsymbol{Z}}$ *as a truncated random matrix of* $\boldsymbol{Z}$*, such that*

$$\boldsymbol{Z} = \bar{\boldsymbol{Z}} + \tilde{\boldsymbol{Z}}, \qquad \bar{Z}_{ij} = \begin{cases} Z_{ij}, & \text{if } |Z_{ij}| < B \\ 0, & \text{otherwise} \end{cases} \qquad \text{(G.21)}$$

*with* $B = 2\sigma\sqrt{\log(pn_1n_2)}$*. There exists some positive quantities* $R_1, R_2$ *and* $\bar{L}_f$ *such that*

$$\|f_{\boldsymbol{q}}(\bar{\boldsymbol{Z}})\|_{\mathrm{op}} \le R_1, \quad \max\left\{\left\|\mathbb{E}[f_{\boldsymbol{q}}(\bar{\boldsymbol{Z}})(f_{\boldsymbol{q}}(\bar{\boldsymbol{Z}}))^T]\right\|_{\mathrm{op}}, \left\|\mathbb{E}[(f_{\boldsymbol{q}}(\bar{\boldsymbol{Z}}))^T f_{\boldsymbol{q}}(\bar{\boldsymbol{Z}})]\right\|_{\mathrm{op}}\right\} \le R_2, \tag{G.22}$$

$$\|f_{\boldsymbol{q}_1}(\bar{\boldsymbol{Z}}) - f_{\boldsymbol{q}_2}(\bar{\boldsymbol{Z}})\|_{\mathrm{op}} \le \bar{L}_f \|\boldsymbol{q}_1 - \boldsymbol{q}_2\|_2, \qquad \forall \boldsymbol{q}_1, \boldsymbol{q}_2 \in \mathbb{S}^{n-1}. \tag{G.23}$$

*Then with probability greater than* $1 - (n_1 n_2 p)^{-2} - cM^{-c'n}$ *for some constants* $c, c' > 0$ *and* $M = (\bar{L}_f + L_f)(p + d_1 + d_2)$*, one has*

$$\sup_{\boldsymbol{q} \in \mathbb{S}^{n-1}} \left\| \frac{1}{p} \sum_{i=1}^p f_{\boldsymbol{q}}(\boldsymbol{Z}_i) - \mathbb{E}[f_{\boldsymbol{q}}(\boldsymbol{Z})] \right\|_{\mathrm{op}} \lesssim \frac{(d_1 \wedge d_2)B_f}{n_1 n_2 p} + \sqrt{\frac{R_2 n \log(M)}{p}} + \frac{R_1 n \log(M)}{p}.$$

**Lemma G.6 (Corollary** *F*.2**, [35])** *Let* $\boldsymbol{Z}_1, \boldsymbol{Z}_2, \dots, \boldsymbol{Z}_p$ *be i.i.d realizations of a sub-Gaussian random vector* $\boldsymbol{Z} \in \mathbb{R}^{n_1}$ *satisfying*

$$\mathbb{E}[\boldsymbol{Z}] = 0, \qquad \mathbb{P}(|Z_j| > t) \le 2\exp\left(-\frac{t^2}{2\sigma^2}\right), \qquad \forall 1 \le j \le n_1. \qquad \text{(G.24)}$$

*For any fixed* $\boldsymbol{q} \in \mathbb{S}^{n-1}$*, define a function* $f_{\boldsymbol{q}} : \mathbb{R}^{n_1} \to \mathbb{R}^{d_1}$ *satisfying the following conditions.*

- **Condition 1.** *For some positive numbers* $B_f, L_f > 0$*,*

$$\|\mathbb{E}[f_{\boldsymbol{q}}(\boldsymbol{Z})]\|_2 \le B_f, \qquad\qquad\qquad\qquad\qquad \text{(G.25)}$$
$$\|\mathbb{E}[f_{\boldsymbol{q}_1}(\boldsymbol{Z})] - \mathbb{E}[f_{\boldsymbol{q}_2}(\boldsymbol{Z})]\|_2 \le L_f \|\boldsymbol{q}_1 - \boldsymbol{q}_2\|_2, \qquad \forall \boldsymbol{q}_1, \boldsymbol{q}_2 \in \mathbb{S}^{n-1}. \qquad \text{(G.26)}$$

- **Condition 2.** *Let* $\bar{\boldsymbol{Z}}$ *be the truncated random vector of* $\boldsymbol{Z}$*, such that*

$$\boldsymbol{Z} = \bar{\boldsymbol{Z}} + \tilde{\boldsymbol{Z}}, \qquad \bar{Z}_j = \begin{cases} Z_j, & \text{if } |Z_j| < B \\ 0, & \text{otherwise} \end{cases} \qquad \text{(G.27)}$$

*with* $B = 2\sigma\sqrt{\log(pn_1)}$*. There exists some positive numbers* $R_1, R_2$ *and* $\bar{L}_f$ *such that*

$$\|f_{\boldsymbol{q}}(\bar{\boldsymbol{Z}})\|_2 \le R_1, \qquad \mathbb{E}\left[\|f_{\boldsymbol{q}}(\bar{\boldsymbol{Z}})\|_2^2\right] \le R_2, \qquad\qquad \text{(G.28)}$$
$$\|f_{\boldsymbol{q}_1}(\bar{\boldsymbol{Z}}) - f_{\boldsymbol{q}_2}(\bar{\boldsymbol{Z}})\|_2 \le \bar{L}_f(\sigma) \|\boldsymbol{q}_1 - \boldsymbol{q}_2\|_2, \qquad \forall \boldsymbol{q}_1, \boldsymbol{q}_2 \in \mathbb{S}^{n-1}. \qquad \text{(G.29)}$$

*Then with probability greater than $1 - (n_1 p)^{-2} - cM^{-c'n}$ for some constants $c, c' > 0$ and $M = (\bar{L}_f + L_f)(p + d_1)$, one has*

$$\sup_{\boldsymbol{q} \in \mathbb{S}^{n-1}} \left\| \frac{1}{p} \sum_{i=1}^{p} f_{\boldsymbol{q}} (\boldsymbol{Z}_i) - \mathbb{E} \left[ f_{\boldsymbol{q}} (\boldsymbol{Z}) \right] \right\|_{\text{op}} \lesssim \frac{B_f}{n_1 p} + \sqrt{\frac{R_2 n \log(M)}{p}} + \frac{R_1 n \log(M)}{p}.$$

Our analysis also uses the following lemmas that have already been established in the existing literature.

**Lemma G.7** (**Lemma 4, [44]**) *Let $\boldsymbol{v} \in \mathbb{R}^n$ contains i.i.d. $\mathrm{Ber}(\theta)$ random variables. We have*

$$\mathbb{P} \left[ \|\boldsymbol{v}\|_0 \geq (1 + t) n\theta \right] \leq 2 \exp \left( -\frac{3t^2 n\theta}{2t + 6} \right) \tag{G.30}$$

**Lemma G.8** (**Corollary** F.5, **[35]**) *Suppose $\boldsymbol{x}$ is i.i.d Bernoulli-Gaussian random variables with parameter $(\theta, \sigma^2 = 1)$. For any $\boldsymbol{q} \in \mathbb{S}^{r-1}$, let*

$$f_{\boldsymbol{q}}(\boldsymbol{x}) = \frac{1}{3\theta} \left( \boldsymbol{q}^T \boldsymbol{x} \right)^3 \boldsymbol{x}.$$

*We have*

$$\sup_{\boldsymbol{q} \in \mathbb{S}^{r-1}} \mathbb{E} \left[ \| f_{\boldsymbol{q}} (\boldsymbol{x}) \|_2^2 \right] \leq C\theta^{-1} r, \tag{G.31}$$

*for some constant $C > 0$.*

**Lemma G.9** (**Corollary** F.7, **[35]**) *Suppose $\boldsymbol{x} \in \mathbb{R}^r$ contains i.i.d Bernoulli-Gaussian random variables with parameter $(\theta, \sigma^2 = 1)$. For any $\boldsymbol{q} \in \mathbb{S}^{r-1}$, let*

$$f_{\boldsymbol{q}}(\boldsymbol{x}) = \frac{1}{\theta} \left( \boldsymbol{q}^T \boldsymbol{x} \right)^2 \boldsymbol{x}\boldsymbol{x}^T.$$

*We have*

$$\sup_{\boldsymbol{q} \in \mathbb{S}^{r-1}} \left\| \mathbb{E} \left[ f_{\boldsymbol{q}} (\boldsymbol{x}) (f_{\boldsymbol{q}} (\boldsymbol{x}))^T \right] \right\|_{op} \leq C\theta^{-1} r^2, \tag{G.32}$$

*for some constant $C > 0$.*