# OpenReview forum: "Unique sparse decomposition of low rank matrices"
_NeurIPS.cc/2021/Conference — NeurIPS 2021 Poster_

### Official Review · Reviewer_zPkZ · 2021-07-03

**Rating:** 9
**Confidence:** 3

**Summary:**

The paper studies the problem of decomposing a low-rank p x n matrix Y into Y = AX, where A is p x r of full column rank and X is a sparse r x n matrix.

The paper first considers the case where A has orthonormal columns. It shows that by accessing Y only, one can recover approximately all columns of A up to the signs. It constructs a minimization programme involving the Schatten-4 norm, whose solution is an approximation to some column of A. The minimization problem can be solved by second-order methods. Then one can project A away from this solution and repeat the process. The paper gives quantitative error guarantees, provided that n is sufficiently large. The paper generalizes the result of the orthonormal case to a general A by preconditioning.


**Limitations And Societal Impact:**

NIL

**Main Review:**

The paper is very well written and smooth to read. I did not check the proofs for most parts except a few lemmata for the case where A has orthonormal columns. This special case is convincingly correct and I believe the generalization to the non-orthonormal case should work out.

The paper can be regarded as a follow-up paper of [34], which considers the case where A has full row rank. In this paper, A has full column rank, which creates many difficulties in the analysis, since A has nontrivial null space and the solver for the minimization problem thus has to avoid moving into the null space. Much more sophisticated analysis is therefore needed.

The weak part of the paper is the experiment section. It would be a better paper if it considers the runtime of the algorithm and comparisons with some existing methods in dictionary learning. Despite the drawback in experiments, this paper makes a solid contribution to the matrix decomposition problems and I recommend its acceptance.

Minor comments:
- Between Line 187 and 188: min (q) should be min F(q)
- Between Line 193 and 193: min (q) should be min f(q)


**Time Spent Reviewing:**

5

---

> ### Author Response · Authors · 2021-08-10
> **We thank the reviewer for appreciating our work.**
>
> Since the low rank setting is rarely used in dictionary learning problem, we therefore compare our problem with Sparse PCA instead of dictionary learning experimentally. We will include more experiments on dictionary learning for audience's interests.

---

### Official Review · Reviewer_cZxs · 2021-07-09

**Rating:** 7
**Confidence:** 5

**Summary:**

This paper studies the unique decomposition of a low-rank matrix into sparse and full column-rank components. Explicit conditions under which such decomposition is possible (up to a signed permutation) are derived. Moreover, an initialization scheme is proposed that, together a generic second order method, guarantees the exact and unique recovery of the sparse and full column-rank components. The authors first study the setting where the full column-rank component has orthogonal columns. Then they extend their results to the general setting where this component is no longer column-orthogonal. To the best of my knowledge, the technical contributions of the paper are sound and correct.

**Limitations And Societal Impact:**

The authors are encouraged to address the above-mentioned comments.

**Main Review:**

This paper has a strong technical component. The ideas are clear and novel, and the paper is very well-written. Below are my comments:

1. While the technical contributions of the paper is significant, the paper lacks a proper motivation. In particular, the authors have not adequately explained the main applications of the considered problem. In my opinion, the proposed framework can readily be applied to various applications of sparse PCA. The authors are encouraged to provide a better motivation of their work.

2. The authors claim that “The $\ell_4$ norm objective function and its variants have been adopted as a sparsity regularizer in a line of recent works.” This is not correct, since the $\ell_4$ norm does not impose sparsity; it rather helps recovering the full-rank component, from which the sparse component can be derived. The authors should make this clear in the revised manuscript.
3. The authors have nicely compared their proposed technique with the sparse dictionary recovery. However, they should provide real-world and concrete case studies where the techniques based on sparse dictionary recovery fail (specifically, the full row-rank assumption), and their proposed problem must be solved instead.

4. The authors must explicitly define the notion of descent algorithm in Section 4. Based on the proof of 4.2, it seems that the only requirement for the algorithm to avoid $R_0(c^*)$ is $f(x_0)\leq f(x_1)\leq \dots\leq f(x_t)$. This should be explicitly mentioned in the main body of the paper (in fact, it is one of the strengths of the proposed initialization technique).

5. It is known that several first order methods, such as perturbed gradient descent also escape saddle points, but they are not guaranteed to decrease the objective function at every iteration (as required in Lemma 4.2). In light of this, do the provided guarantees only hold with second-order methods (such as trust region and cubic regularization)? The authors have not mentioned which algorithm they use in their simulations.

6. My main comment is about the simulations. The authors only consider synthetically generated and relatively small scale instances to evaluate the performance of their algorithm. Moreover, in the main body of the paper, they only consider column-orthogonal A and only recover one column of A (as opposed to the full matrix). They are encouraged to add a realistic case study, showcasing the applicability of their proposed method in real life. Moreover, after carefully reading the appendix, I realized that the proposed algorithm has significantly inferior performance for a general choice of matrix A. In particular, even in the very low-rank instances (r=10), the best recovery error is around 20%, which is quite high. It is crucial that the authors explain why their simulations do match their theoretical guarantee in the general setting. I will certainly increase my overall rating if the authors successfully justify this gap.

========================================

After rebuttal: The authors have addressed my comments in the rebuttal. I strongly encourage the authors to include the simulations on general choices of matrix A in the main body of the paper.

**Time Spent Reviewing:**

4

---

> ### Author Response · Authors · 2021-08-10
> **Answer for questions from cZxs**
>
> Thank you for your valuable suggestions and we address them as the following.
>
> **Answer to Q1**
>
> We thank the reviewer for pointing out the potential application of the proposed framework, which can be applied to several SPCA applications, and numbers of other intrinsic low-rank models. One major distinction is the presence of nontrivial noise in sparse PCA problems. To fully adapt to the sparse PCA problem, we need to extend the proposed procedure in noisy setting, which is a direction we are pursuing right now. We will include more discussions of the motivation in revisions of this paper.
>
> **Answer to Q2**
>
> We do agree with the reviewer that recovery of sparse vector $ A^Tq$ could be derived from recovering full rank component of $ A$. At the same time, we would love to note, although not being a classic sparsity penalty, L4 norm can be shown effective to approximately encourage sparsity in many recent works [29,42,41,34].
>
> **Answer to Q3**
>
>   Compared to sparse dictionary learning, our proposed model further imposes the low rankness of the factorization. In terms of data compression, we expect our model to render sparser and more concise factorization. We will explore real world examples illustrating this.
>
>
> **Answer to Q4&Q5**
>
> We really appreciate the reviewer for the insightful suggestions and comments on numerical methods, and we will include more explanations in the revised version. Our current analysis only guarantees the success of descent algorithms (or second-order algorithms for nonconvex problems), however, we conjecture that it should be straightforward to extend to perturbed first-order methods by a slight modification of regions $R_0$ and $R_1$. The crucial technical bottleneck would be to show that the slight perturbation of the first-order method will not move the iterates far away from $R_1$.
>
> Empirically, we found that any method that is able to escape the saddle point efficiently works for our problem.  In the experiment part, we use power methods [34, 41], which achieve similar performance compared to Riemannian gradient descent with backtracking line search step.
>
> **Answer to Q6**
>
> In Section E.1 of the appendix, our experiment on recovering the general full rank matrix $A$ uses the averaged estimation error as our metric, which is defined as the normalized Frobenius norm of $A_{est} -A P$ with $P$ being some permutation matrix (see, (5.2)). Hence, the estimation error for a fixed sample size $n$ (approximately $0.2$) seems reasonable and could further decrease as the sample size increases. Moreover, if we normalize the metric by the scale of the matrix $A$ itself, since $\mathbb{E}_{i}\|A_i\|_2=\sqrt{p}=10$ with entries of $A$ being i.i.d. $N(0,1)$, the relative error of recovering $A$ is only approximately $2$%.
>
>  In our revised version, we will also include the experiment on the performance of our proposed algorithm to recover one column of general full column rank $ A$ by using the successful recovery metric, defined via (5.1).
>
> On the other hand, it is worth mentioning that, a performance gap between recovering a general $ A$ (assumption $2.2$) and an orthonormal $ A$ (assumption $2.3$) is expected, as a general $ A$ requires preconditioning and hence requires more samples to obtain comparable results. This can be seen from our theoretical results (Theorems 3.4 \& 3.6).

---

> > ### Comment · Reviewer_cZxs · 2021-08-24
> > **Review update**
> >
> > I thank the authors for addressing my comments. I will raise my score accordingly.

---

### Official Review · Reviewer_romD · 2021-07-14

**Rating:** 5
**Confidence:** 3

**Summary:**

The paper studies the problem of computing a low rank decomposition of a matrix. Specifically, let $A$ be a $p \times r$ matrix and $B$ be a $X \times n$ matrix. Given the matrix $Y = AX$, the authors study the problem of computing the matrices $A$, $X$. Without further assumptions, the problem is ill-defined as for any decomposition $A$, $X$ and any invertible matrix $Q$, $AQ$ and $Q^{-1}X$ is another decomposition of the matrix $Y$.

The authors make an assumption here that the matrix $X$ is sparse in particular the matrix $X$ has iid entries where each entry is independently nonzero with probability $\theta$ and conditioned on an entry being nonzero it has gaussian distribution. Assume that $A$ has orthonormal columns. The authors argue that solving the maximization problem $\max_{q: \|q\|_2 = 1}\|q^T Y\|_4^4$ recovers a column of the matrix $A$ upto a sign with some error additive error. Then they show that for the general case of $A$, they can use a preconditioner computed using $Y$ that would reduce the problem to the case  orthonormal matrix and additive noise matrix of small norm.

They argue that any 2nd order optimization algorithm can be used by arguing that all critical points are either close to recovering the optimal column or that they are saddle points and that there is a direction in which the hessian is negative.

They show some experiments by generating a random matrix $A$ with orthonormal columns and generating a matrix $X$ from the above described Gaussian-Bernoulli distribution. They show that as the matrix $X$ gets sparser, the probability of recovery of $A$ increases as predicted by their theorems and also that as $n$, the number of columns of the matrix $X$ increases, the probability of recovery of $A$ increases.

**Main Review:**

I think the assumption of the matrix $X$ being generated from the distribution as in the paper is very weak evident by the fact that their algorithm works just using concentration of measure. Is there any motivation for considering such a matrix $X$ and where such a setting is plausible. You mention that other references consider the same distribution but nothing more than that.

Even the population case Theorem~3.1 isn't very interesting and not very insightful. For this case, the objective authors solve is $\max_q [(1-\theta)\|A^Tq\|_4^4 + \theta\|A^Tq\|_2^2]$. As the assumption is that $A$ is orthonormal, if the vector $q$ isn't in the column span of $A$, then one can just perform a descent along the component of $q$ that is orthogonal to column space of $A$ to reduce the cost. For $q$ that lies in the column space, the second  term is equal to theta and so the only relevant term is the first and perturbing $q$ in the direction of $a_i$ that has the maximum $|\langle a_i, q\rangle|$ should increase the objective value. The proofs in the paper also seem to essentially use these facts. Under a lot of notation, the proofs obscure the simple ideas because of which they work. Clarify if my understanding is wrong and if I am missing anything here.

Strengths:
- Main part of the paper is written well and easy to follow
- The objective used to recover the decomposition is not complicated and simple to implement

Typos/Suggestions:
- Try to motivate why the distribution of $X$ you consider is interesting and if possible extend the algorithm to more types of matrices $X$. I believe the paper could be much stronger than the current version if you solve the problem for a more general $X$.
- The paper doesn't conform to rules of the conference. Use Times font as in the template however bad it may be :)
- Equations 2.3 and 2.5 are missing $F$ and $f$
- Line 185 "accent" -> "ascent"
- Line 281 "approximates recover" -> "approximately recovers"

**Time Spent Reviewing:**

3 hours

---

> ### Author Response · Authors · 2021-08-10
> **Clarification of major contribution and assumptions**
>
> We thank the reviewer for the thoughtful comments. We will address the main concerns and incorporate other detailed comments in the revision.
>
> **Distribution of $X$**:
>
> Our paper chooses Gaussian distribution for non-zero elements in $X$ and Bernoulli distribution to control the proportion of non-zero elements (sparsity). Here the Gaussianity is a convenience for mathematical exposition and our results can be extended to zero-mean symmetric distribution easily. The entries of $X$ are further assumed to be independent. As discussed in [18], the independence assumption plays a much more important role in the analysis for both finite sample and population case in related problems. And such Bernoulli-Gaussian assumption has been widely adopted in modeling sparse random matrix [2,1,37]. We agree that the model without the i.i.d. assumption of $X$ is more applicable to real world applications,  and we plan to defer it to future study.
>
> **Main Contribution**:
>
> We agree with the reviewer that once the objective function is decomposed into the convex combination of $\ell_4$ and$\ell_2$ term, then the intuition is clear and convincing. We are actually excited that an over-the-shelf algorithm would work by just considering the measure concentration. The main bottleneck in this paper actually lies in the nonconvexity of the problem, which usually involves numerous local solutions and saddle points. We view our contribution as proving that with this natural and intuitive objective, all the local solutions in R1 are the desired solutions and all the saddle points in R1 have negative second-order curvature. This geometric property is exactly what guarantees the success of simple algorithms, despite the notorious nonconvexity of the problem.
>
> We are also disappointed that this simple yet beautiful idea was dimmed due to the heavy calculation, mostly because of the spherical constraint and heavy tailed $\ell_4$ objective, which are (unfortunately) necessary for the success of this problem. Actually, the heavy technical derivations has been common to geometry characterization for nonconvex problems, and we would be rather delighted if better techniques could be developed for nonconvex problems.

---

### Official Review · Reviewer_wCTm · 2021-07-16

**Rating:** 6
**Confidence:** 3

**Summary:**

The authors propose a new method and analysis to decompose a given matrix $Y$ as $Y=AX$ with $X$ sparse and $A$ full column rank. This is similar but different to sparse PCA and dictionary learning, and has interesting applications.

The proposed algorithm essentially recovers $A$ one column at a time (after deflation) and maximizes the $\ell_4$ norm of the overlap of a unit-norm vector with $Y$. As the authors also state, using the $\ell_4$ norm is not entirely new but it gives a benign optimization landscape (in particular, strict local minimizers that are close to the desired solutions).

The best results (theoretically and experimentally) are obtained when $A$ is an orthonormal matrix and $X$ is a random matrix from a Bernouilli-Gaussian distribution. The authors propose a procedure (called preconditioning) that generalizes their results (again theoretically and experimentally) to any $A$ but it is difficult to see how good the results are. In any case, the provided experiment clearly show that we do no longer have exact recovery in this case.

While the theoretical results seem -- I did not check the proofs in the 40(!) page appendix -- nontrivial and interesting, they only apply when recovering one vector and for a random $X$ from the Bernouilli-Gaussian distribution. No results are given for general $X$. In addition, the proposed deflation strategy is a typical technique to extend to multiple vector but typically performs quite poorly when $A$ is ill-conditioned (near linear depednent columns). The authors should therefore restate their contributions in the abstract and introduction to better reflect their main contributions.

**Main Review:**

Overall impression: The main text is very well written with sufficient motivation and literature overview. The proposed method with its theoretical results is also well explained and, from a high level, look believable. However, I did not check the proofs of the claimed results in the Lemmas and Theorems since the whole appendix is 40 pages. I therefore believe this paper is more suited for a traditional journal where proofs get their deserved full attention.

Questions:

1) Abstract: "We prove that this low rank, sparse decomposition of Y can be uniquely identified, up to some intrinsic signed permutation" and "Our geometric analysis for its nonconvex optimization landscape shows that any strict local solution is close to the ground truth solution, and can be recovered by a simple data-driven initialization followed with any second order descent algorithm." Only the one vector case is proven. Please reformulate (and also at other places throughout the paper).

2) Assumption 2: If $\|A\|_{op}=1$ is wlog, why not remove it from the assumption? Such claims are sometimes made incorrectly and here it seems it can easily be avoided.

3) "semi-orthogonal matrix" is not really standard (despite that is appears on Wikipedia). Matrix with orthonormal columns is clearer.

4) The paper (mostly) deals with the one vector case and only random $X$ from a specific distribution. The deflation idea is sensible but not elaborated much (no theoretical results and only one experiments). The random $X$ is typical but the results are clearly not applicable to general sparse $X$. I strongly recommend restating the contributions in the abstract and introduction to reflect this better.

5) The theoretical guarantees seem to have a quite bad dependence on $r$ (even $r^5$). How does this compare to sparse PCA or other sparse low-rank results?

6) The experimental setup is for a very nice $A$: orthonormal and incoherent columns. I would prefer a more challenging examples in the main text or appendix since the theory allows for any $A$ of full column rank.

7) How close is the theory to predict the phase transitions in the experiments?

8) Line 246: "Lemma 3.2 shows that any critical point $\bar q \in R_1$ is either a strict saddle point or ..." Please explain since Lemma 3.2 only deals with a local solution $\bar q \in R_1$.

9) Thm 3.1 (and others): What is meant by "local solution"? A strict local minimizer?

10) Remark 2.6: Why would (2.3) approach the expectation in (2.4) for large $n$ when I only solve for a single $Y$?

11) Typo: footnote page 5: processes -> pocesses

12) Lemma 2.5: Missing definition of $f$

13) The intuition on lines 175+ for Lemma 2.4 does not give much insight why one would use the $l_4$ norm instead of the $l_2$ norm. In particular, it does not explain why I would recover a column of $A$ with the $l_4$ norm.



**Time Spent Reviewing:**

4h

---

> ### Author Response · Authors · 2021-08-10
> **Thanks for your insightful comments**
>
> We respond to the major concerns below and will address other minor issues in a later version of the paper.
>
> **Answers to Q1**
>
> Thanks for pointing it out, and we will correct related descriptions throughout the paper.
>
> **Answers to Q2**
>
> We imposed this assumption for simple exposition of the proof. We agree with the reviewer and will remove it from the assumption statement to the discussion afterward.
>
> **Answers to Q3**
>
> Thanks for raising this issue and we will address it in the revised version.
>
> **Answers to Q5**
>
>  We thank the referee for raising this important point. First, in our setting, we clarify that the dependence of the sample size $(n)$ on $r^5$ originates from the requirement that our initialization point does not lie in the null region. Indeed, provided a good initialization, the requirement of the sample size can be reduced to $r^3$ (see, Theorem 3.6). Therefore, the requirement of $r^5$ could be relaxed if a better initialization is available. On the other hand, in the many applications of our model, for instance, applications of the sparse PCA and low-rank models, the rank $r$ is oftentimes small and is assumed to be a fixed constant, independent of the sample size $n$.
>
> Since our model is not the same as the model for the sparse PCA (see, Section 1.1), our result is not directly comparable to that of the sparse PCA. Specifically, for the sparse PCA, existing results mainly focus on the error rate of estimating the space spanned by the columns of $A$ in the presence of some additional additive error $E$ (see, Birnbaum et al, 2012 and Ma, 2012, for fixed $r$, and Cai, Ma and Wu, 2014 for growing $r$). By contrast, there is no additive error and the matrix $A$ itself can be uniquely recovered under our model.
>
> **Answers to Q6**
>
> At section $E.1$ in the appendix, we demonstrate the performance of recovering whole general iid gaussian matrix **$A_i$** under difference scenes, notice that here the recovery error can be seen as the average of different $i$ such that $\mathbb{E}_{i}\|A_i- P\bar{A}_i\|_2$ for iid gaussian matrix $A$, estimated matrix $\bar{A}$ and properly chosen permutation matrix $P$. Also we will include additional results of our proposed algorithm to recover one column of general full column rank $A$ in our revised version and demonstrates that our algorithm ensures successful recovery for low rank($r=10$) and relatively small sparsity case($\theta\leq 0.45$)
>
> **Answers to Q7**
>
> Thanks for suggesting this promising direction. We are also intrigued to see if this problem has a sharp phase transition, and what is the best achievable bound / complexity. These are all interesting questions we aim to address in the future revision of this paper.
>
> **Answers to Q8**
>
> We thank the reviewer for pointing out the awkwardness here and we will modify the sentence accordingly. Indeed, we proved Lemma 3.2 by studying all the critical points in $R_1$, which can be categorized into two cases: (1) a strict saddle point and (2) a local solution with positive second-order curvature.
>
> **Answers to Q9**
>
> We refer local solution as a solution whose neighborhood does not include a better solution. Any local solution described in Theorem 3.1 is a strict local minimizer. For the proposed algorithm, we only need to demonstrate the desired solution is a local solution, although further characterization of second-order curvature (or strictness) would potentially contribute to the establishment of the convergence rate.
>
> **Answers to Q10**
>
> Recall that $ Y=AX \in R^{p\times n}$ and $ X\in R^{r\times n}$ is the random sparse matrix with i.i.d. columns. The matrix $Y$ thus contains $n$ i.i.d. columns as well. Since the objective in (2.4) can be written as the sum of $n$ i.i.d. terms, it concentrates around its expectation, with respect to the randomness in $ X$. We refer to Appendix B.9 for the precise concentration results.
>
> **Answers to Q13**
>
> Since inequality $\|A^{T} q\|_4^4\leq \|A^{T} q\|_\infty^2\|A^{T} q\|_2^2$ always holds, the equality can be obtained when $A^{T}q$ has only one non-zero entry, or $ q$ equals to one column of orthonormal $A$ up to sign permutation. In contrast, any $q\in span(A)$ renders the unit $\ell_2$ norm of $A^Tq$ for orthonormal $A$.
>
>  We thank the reviewer for suggesting all other improvements and will incorporate them in the future version of this manuscript.

---

> > ### Comment · Reviewer_wCTm · 2021-08-24
> > **Question 4**
> >
> > Could you answer my Q4?

---

> > > ### Author Response · Authors · 2021-08-24
> > > **Assumptions**
> > >
> > > We agree with the reviewer that our current analysis only addresses vector case and algorithmically requires a deflation procedure to recover the whole matrix factorization, which is a common approach in global nonconvex sparse matrix / tensor recovery literature [17,34,45,38,37]. We expect the deflation procedure can be proved relatively easily, although the more efficient and practical algorithmic approach would be to directly optimize over the orthogonal matrix, where only local guarantees are known to our knowledge [41]. We thank the reviewer for pointing out the gap between the theory and practice of nonconvex problems, and this is definitely a very interesting direction we are aiming to pursue in the long term.
> > >
> > > Our paper chooses Gaussian distribution for non-zero elements in $X$ and Bernoulli distribution to control the proportion of non-zero elements (sparsity). Here the Gaussianity is a convenience for mathematical exposition and our results can be extended to zero-mean symmetric distribution easily. The entries of  $X$ are further assumed to be independent. As discussed in [18], the independence assumption plays a much more important role in the analysis for both finite sample and population case in related problems. And such Bernoulli-Gaussian assumption has been widely adopted in modeling sparse random matrix [2,1,37]. We agree that the model without the i.i.d. assumption of $X$ is more applicable to real world applications, and we plan to defer it to future study.
> > >
> > > We thank the reviewer for pointing out above issues and will restate the contribution precisely in the revision.

---

### Decision · Program_Chairs · 2021-09-27

**Decision:**

Accept (Poster)

**Comment:**

Thank you for your submission to NeurIPS. There is broad consensus among the reviewers that the paper presents a significant step forward on a difficult problem. The reviewers found the paper well-written and easy to read.

Three of the four reviewers felt that the experimental aspects of the paper could have been improved. One reviewer raised concerns regarding the Bernoulli--Gaussian assumption on $X$. While the assumption is standard in the area, it nevertheless appears to be removed from practical applications. This reviewer suggested that more thorough simulations could have better covered this weakness.

Given the interesting and nontrivial nature of the theoretical contributions, the authors are encouraged to follow-up by submitting a long version of the paper to a traditional math journal.